# Mean-Field Langevin Dynamics for Signed Measures via a Bilevel Approach

**Guillaume Wang**[*1]     **Alireza Mousavi-Hosseini**[*2]     **Lénaïc Chizat**[1]

[1]École polytechnique fédérale de Lausanne
[2]University of Toronto and Vector Institute

guillaume.wang@epfl.ch,  mousavi@cs.toronto.edu,  lenaic.chizat@epfl.ch

## Abstract

Mean-field Langevin dynamics (MLFD) is a class of interacting particle methods that tackle convex optimization over probability measures on a manifold, which are scalable, versatile, and enjoy computational guarantees. However, some important problems – such as risk minimization for infinite width two-layer neural networks, or sparse deconvolution – are originally defined over the set of signed, rather than probability, measures. In this paper, we investigate how to extend the MFLD framework to convex optimization problems over signed measures. Among two known reductions from signed to probability measures – the *lifting* and the *bilevel* approaches – we show that the bilevel reduction leads to stronger guarantees and faster rates (at the price of a higher per-iteration complexity). In particular, we investigate the convergence rate of MFLD applied to the bilevel reduction in the low-noise regime and obtain two results. First, this dynamics is amenable to an annealing schedule, adapted from [SWON23], that results in improved convergence rates to a fixed multiplicative accuracy. Second, we investigate the problem of learning a single neuron with the bilevel approach and obtain local exponential convergence rates that depend polynomially on the dimension and noise level (to compare with the exponential dependence that would result from prior analyses).

## 1  Introduction

Let $\mathcal{M}(\mathcal{W})$ be the set of finite signed measures on a compact Riemannian manifold without boundaries $\mathcal{W}$ and let $G : \mathcal{M}(\mathcal{W}) \to \mathbb{R}$ be a convex function, assumed smooth in the sense of Assumption 1 below. In this paper, we investigate optimization methods to solve

$$\min_{\nu \in \mathcal{M}(\mathcal{W})} G_\lambda(\nu), \qquad\qquad G_\lambda(\nu) \coloneqq G(\nu) + \frac{\lambda}{2}\|\nu\|_{TV}^2, \qquad\qquad (1.1)$$

where $\|\cdot\|_{TV}$ is the total variation norm and $\lambda > 0$ the regularization level.[1] This covers for instance risk minimization for infinite-width 2-layer neural networks (2NN) [BRVDM05; Bac17] by taking $\mathcal{W} = \mathbb{S}^d$ the unit sphere in $\mathbb{R}^{d+1}$ or $\mathcal{W} = \mathbb{R}^{d+1}$ and

$$G(\nu) = \mathbb{E}_{(x,y)\sim\rho}\Big[\ell(h(\nu, x), y)\Big] \qquad \text{where} \qquad h(\nu, x) = \int_{\mathcal{W}} \varphi(\langle x, w\rangle)\mathrm{d}\nu(w). \quad (1.2)$$

Here $\varphi : \mathbb{R} \to \mathbb{R}$ is the activation function, $h(\nu, \cdot)$ is the predictor parameterized by $\nu$, $G$ is the (population or empirical) risk under the data distribution $\rho \in \mathcal{P}(\mathbb{R}^{d+1}\times\mathbb{R})$, and $\ell$ is smooth (uniformly

---

[*]Equal contributions, authors ordered randomly.

[1]The square exponent on $\|\cdot\|_{TV}$ might appear unusual, but it is convenient for our subsequent developments. We show in App. A that the regularization path is the same with or without the square.

38th Conference on Neural Information Processing Systems (NeurIPS 2024).

in $y$) and convex in its first argument. These 2NNs will be our guiding examples throughout, but note that the class of problems covered by Eq. (1.1) is more general and includes for instance sparse deconvolution via the Beurling-LASSO estimator [DG12] or optimal design [MZ04].

To tackle such problems, interacting particle methods use the parameterization $\nu = \sum_{i=1}^m r_i \delta_{w_i}$ and apply gradient methods in a well-chosen geometry [Chi22c; YWR23; GCM23]. They have recently gained traction thanks to their scalability and flexibility, and in the context of 2NNs, the usual gradient descent algorithm is an instance of such a method. On the downside, *global* convergence guarantees remain difficult to obtain due to the nonconvex nature of the reparameterized problem and existing positive results require either very specific settings [LMZ20], or modifications of the dynamics which often limit their scalability[2].

In a related, but slightly different context, mean-field Langevin dynamics (MFLD) solve entropy-regularized problems of the form

$$\min_{\mu \in \mathcal{P}(\mathcal{W}')} F_\beta(\mu), \qquad\qquad F_\beta(\mu) \coloneqq F(\mu) + \beta^{-1} H(\mu), \qquad\qquad (1.3)$$

where $\mathcal{P}(\mathcal{W}')$ is the space of probability measures on a manifold $\mathcal{W}'$ (typically $\mathbb{R}^d$), $F : \mathcal{P}(\mathcal{W}') \to \mathbb{R}$ is a (sufficiently regular) convex functional, $H(\mu) = \int \log(\mathrm{d}\mu/\mathrm{d\,vol})\mathrm{d}\mu$ is the negative differential entropy and $\beta > 0$. These dynamics are obtained as the mean-field limit of *noisy* interacting particles dynamics [MMN18; HRŠS21] and converge globally at an exponential rate [NWS22; Chi22b], under two key conditions on $F$: (i) a notion of regularity, which we refer to as *displacement smoothness* (see P1 below) and (ii) a *uniform log-Sobolev inequality (LSI)* condition (see P2 below). These mean-field, continuous-time guarantees have been further refined into computational guarantees for fully discrete algorithms [CRW22; SWN23]. The favorable properties of MFLD naturally lead to the following question:

*Can we efficiently solve problems of the form Eq. (1.1) using MFLD?*

At first, it is not obvious that MFLD can be applied at all since it is originally defined only for problems over probability measures. However, we can find in the literature two general recipes to reduce a problem over $\mathcal{M}(\mathcal{W})$ to a problem over $\mathcal{P}(\mathcal{W}')$, thus amenable to MFLD. The first one is a *lifting* reduction, that takes $\mathcal{W}' = \mathbb{R} \times \mathcal{W}$ where the extra dimension serves to encode the signed mass of particles [CB18, Section A.2] [Chi22c]. The second one, that takes $\mathcal{W}' = \mathcal{W}$, is a *bilevel* reduction [Bac21; TS24] that uses a variational representation of the regularizer $\|\cdot\|_{TV}^2$, common in the multiple kernel learning literature [LCBGJ04]. A first task is thus to compare the behavior of MFLD on these two approaches. Furthermore, MFLD involves an entropic regularization which is absent from Eq. (1.1). A second task is thus to analyze the behavior of MFLD in the large $\beta$ regime, when the regularization vanishes.

In this work, we tackle these two tasks and make the following contributions:

- In Sec. 3, we introduce the lifting and bilevel reductions and compare the "displacement smoothness" (P1) and "uniform LSI" (P2) properties of the resulting problems. These properties play a central role in the global convergence analysis of MFLD. Specifically, we consider a large class of lifting reductions and show that none satisfies simultaneously (P1) and (P2) unless $\lambda$ is large. In contrast, the bilevel reduction satisfies both under mild assumptions. So in the sequel we focus on MFLD applied to the bilevel reduction.

- In Sec. 4, we investigate what convergence rates can be obtained for the problem (1.1) by using MFLD on the bilevel formulation. While a classical simulated annealing technique yields convergence in $O(\log \log t / \log t)$, we show that the structure of the bilevel objective is in fact amenable to a more efficient annealing schedule, adapted from [SWON23], that reaches a fixed multiplicative accuracy, say $1.01 \inf G_\lambda$, in time $e^{O(\lambda^{-1} \log \lambda^{-1})}$ instead of $e^{O(\lambda^{-2})}$ for the classical schedule.

- In Sec. 5, to obtain a more complete picture, we investigate the problem of learning a single neuron. Here, using a Lyapunov type argument, we show that the *local* convergence rate of MFLD applied to the bilevel formulation scales polynomially in $\beta$ and $d$, at odds with all previous MFLD analyses which had exponential dependencies.

All proofs are deferred to the Appendix.

---

[2]Such as forcing the particles to remain close to their initial position [Chi22c], or adding new particles using a potentially hard linear minimization oracle [DDPS19].

## 1.1 Related work

**Particle methods and mean-field limits.** Interacting particle systems have been studied for decades in various fields, see e.g. [Szn91; CD13; Lac18]. Their more recent connection with the standard training of 2NNs [NS17; SS20; RV22; MMN18] has suggested new settings of analysis, where convexity of the functional plays a key role, and has led to many developments. In particular, the case of MFLD (under study here) quickly progressed from nonquantitative guarantees [MMN18; HRŠS21], to mean-field convergence rates [NWS22; Chi22b] and fully discrete computational guarantees [CRW22; SWN23; KZCE+24] in the span of a few years. Recent progress also address its accelerated (underdamped) version [CLRW24; FW23], which could also be of interest in our setting.

**Multiple kernel learning and bilevel training of NNs.** The lifting reductions we consider are inspired by the unbalanced optimal transport literature [LMS18], while the bilevel reduction comes from the Multiple Kernel Learning (MKL) literature [CVBM02; LCBGJ04; RBCG08] (see [Bac19] for an account). While the latter is usually studied with a discrete domain $\mathcal{W}$ (see also [PP21; PP23] for recent computational considerations), it was suggested for the training of large width 2NN in [Bac21] and used in conjonction with MFLD in [TS24] (more details below). Relatedly, a recent line of work studies the (noiseless) training of 2NN in a two-timescale regime, where the outer layer is trained at a much faster rate than the inner layer [BMZ23; MB23; BBP23]. This implicitly corresponds to optimizing the bilevel objective and leads to improved convergence guarantees.

The work that is closest to ours is [TS24], which considers the MFLD on a 2NN with weight decay where the outer layer is optimized at each step. They interpret the resulting dynamics as a kernel learning dynamics and study properties of the learnt kernel and its associated RKHS. While they do not formulate explicitly the problem Eq. (1.1), it can be shown that our approaches are equivalent when considering $\mathcal{W} = \mathbb{R}^{d+1}$ in Eq. (1.2) (and adding an extra regularization). The details are given in Sec. A.2. Key advantages of our formulation with $\mathcal{W} = \mathbb{S}^d$ are that we cover the case of unbounded homogeneous activation functions (such as ReLU), and can obtain improved LSI.

## 2 Background on guarantees for mean-field Langevin dynamics

The MFLD is defined as the Wasserstein gradient flow $(\mu_t)_{t \in \mathbb{R}_+}$ in $\mathcal{P}(\Omega)$ of an objective of the form Eq. (1.3). It is characterized as the solution to the partial differential equation (PDE)

$$\partial_t \mu_t = \operatorname{div}(\mu_t \nabla F'[\mu_t]) + \beta^{-1} \Delta \mu_t, \qquad \mu_0 \in \mathcal{P}(\Omega). \tag{2.1}$$

where $F'[\mu] : \Omega \to \mathbb{R}$ is the *first variation* of $F$ at $\mu$ [San15, Sec. 7.2], defined by $\lim_{\epsilon \downarrow 0} \frac{1}{\epsilon}(F(\mu + \epsilon(\mu' - \mu)) - F(\mu)) = \int F'[\mu] \mathrm{d}(\mu' - \mu)$ for any $\mu' \in \mathcal{P}(\Omega)$. This PDE corresponds to the mean-field limit ($N \to \infty$) of the noisy particle gradient flow $\omega_t \in \Omega^N$:

$$\forall i \leq N, \ \mathrm{d}\omega_t^i = -N \nabla_{\omega_t^i} F^{(N)} \left( \omega_t^1, ..., \omega_t^N \right) \mathrm{d}t + \sqrt{2\beta^{-1}} \mathrm{d}B_t^i, \qquad \omega_0^i \overset{\text{i.i.d.}}{\sim} \mu_0$$

where $F^{(N)} \left( \omega^1, ..., \omega^N \right) = F \left( \frac{1}{N} \sum_{i=1}^N \delta_{\omega^i} \right)$ and the $B_t^i$ are $N$ independent Brownian motions on $\Omega$. The convergence guarantees for MFLD rely on three key properties:

(P0) **(Convexity)** $F$ is convex and is such that $F_\beta$ admits a minimizer $\mu_\beta^*$.

(P1) **(Displacement smoothness)** $F$ is $L$-displacement smooth, in the sense that[3]

$$\forall \mu \in \mathcal{P}_2(\Omega), \ \forall \omega \in \Omega, \ \max_{\substack{s \in T_\omega \Omega \\ \|s\|_\omega \leq 1}} \left| \nabla^2 F'[\mu](s, s) \right| \leq L,$$

$$\text{and} \quad \forall \mu, \mu' \in \mathcal{P}_2(\Omega), \ \forall \omega \in \Omega, \ \|\nabla F'[\mu] - \nabla F'[\mu']\|_\omega \leq L \, W_2(\mu, \mu'),$$

where $\nabla^2$ denotes the Riemannian Hessian.

(P2) **(Uniform LSI)** There exists $\alpha > 0$ such that $\forall t \geq 0$, $F_\beta$ satisfies local $\alpha$-LSI at $\mu_t$, as in Def. 2.1.

---

[3] Strictly speaking, (P1) is only a sufficient condition for displacement smoothness (see details in App. B). We refer to (P1) as displacement smoothness in this paper for conciseness only.

**Definition 2.1** (Local LSI)**.** We say that a functional $F_\beta = F + \beta^{-1}H$ satisfies local $\alpha$-LSI at $\mu \in \mathcal{P}(\Omega)$ if $Z := \int_\Omega \exp\left(-\beta F'[\mu]\right) \mathrm{d}\omega < \infty$ and the *proximal Gibbs measure* $\hat{\mu} := Z^{-1} \exp(-\beta F'[\mu]) \in \mathcal{P}(\Omega)$ satisfies $\alpha$-LSI, that is

$$\forall \mu' \in \mathcal{P}(\Omega),\ H\left(\mu'|\hat{\mu}\right) \leq \frac{1}{2\alpha} I(\mu'|\hat{\mu}),$$

where the relative entropy and relative Fisher Information are respectively defined as

$$H\left(\mu'|\hat{\mu}\right) := \int_\Omega \log\left(\frac{\mathrm{d}\mu'}{\mathrm{d}\hat{\mu}}\right)\mathrm{d}\mu', \qquad I(\mu'|\hat{\mu}) := \int_\Omega \left\|\nabla \log \frac{\mathrm{d}\mu'}{\mathrm{d}\hat{\mu}}(\omega)\right\|_\omega^2 \mathrm{d}\mu'(\omega),$$

and $\|\cdot\|_\omega$ denotes the Riemannian metric.

We review some useful criteria for LSI in App. B. In particular, the uniform LSI property (P2) holds for example when training two-layer neural networks with a frozen second layer, under some technical assumptions such as bounded activation function. In fact in that case, the proximal Gibbs measures $\hat{\mu}$ even satisfy LSI uniformly for *all* $\mu \in \mathcal{P}(\Omega)$ [Chi22b; NWS22].

Note that the Riemannian gradient $\nabla$ and the Laplace-Beltrami operator $\Delta$ appearing in (2.1), as well as the definition of Brownian motion, depend on the Riemannian metric of $\Omega$. This dependency is reflected in (P1) and (P2).

The global convergence of MFLD is guaranteed by the following theorem, with a rate.

**Theorem 2.1** ([Chi22b, Thm. 3.2][NWS22, Thm. 1])**.** *Consider $F : \mathcal{P}(\Omega) \to \mathbb{R}$ and $(\mu_t)$ as in (2.1). If (P0), (P1) and (P2) are satisfied then for $t \geq 0$ it holds*

$$\beta^{-1} H(\mu_t|\mu_\beta^*) \leq F_\beta(\mu_t) - F_\beta(\mu_\beta^*) \leq \exp(-2\beta^{-1}\alpha\, t)\Big(F_\beta(\mu_0) - F_\beta(\mu_\beta^*)\Big).$$

Note that although the $L$-smoothness constant does not appear in Thm. 2.1, it does appear in the discrete-time guarantees of [SWN23], and is thus an important quantity in practice. In this paper, we limit our analysis to the mean-field dynamics (2.1) because its time-discretization has not yet been studied on Riemannian manifolds. In continuous time, the proof of Thm. 2.1 translates directly to Riemannian manifolds thanks to our definition of (P1), see App. B.

## 3 Reductions from signed measures to probability measures

In order to apply the MFLD framework to solve our initial problem over signed measures (1.1), we must first recast it as an optimization problem over probability measures. In this section we build two such reductions, and discuss the properties (P0, P1 and P2) of the resulting problems.

### 3.1 Reduction by lifting

Reductions by lifting consist in representing signed measures as projections of probability measures in the higher dimensional space $\Omega = \mathbb{R} \times \mathcal{W}$. This construction involves the 1-homogeneous projection operator[4] $\boldsymbol{h} : \mathcal{P}_1(\Omega) \to \mathcal{M}(\mathcal{W})$ characterized by

$$\forall \varphi \in \mathcal{C}(\mathcal{W}, \mathbb{R}),\ \int_\mathcal{W} \varphi(w)(\boldsymbol{h}\mu)(\mathrm{d}w) = \int_\Omega r\varphi(w)\mu(\mathrm{d}r, \mathrm{d}w),$$

where $\mathcal{P}_p(\Omega)$ is the subset of $\mathcal{P}(\Omega)$ for which $\int |r|^p \mathrm{d}\mu(\mathrm{d}r, \mathrm{d}w) < +\infty$. For instance, it acts on discrete measures as $\boldsymbol{h}\left(\frac{1}{m}\sum_{j=1}^m \delta_{(r_j,w_j)}\right) = \frac{1}{m}\sum_{j=1}^m r_j \delta_{w_j}$. We also define, for $b \in [1,2]$ and $\mu \in \mathcal{P}_b(\Omega)$, $\Psi_b(\mu) := \left(\int_\Omega |r|^b \mathrm{d}\mu(r,w)\right)^{2/b}$. The objective functional of the lifted problem is then defined, for $\mu \in \mathcal{P}_b(\Omega)$, as

$$F_{\lambda,b}(\mu) := G(\boldsymbol{h}\mu) + \frac{\lambda}{2}\Psi_b(\mu). \tag{3.1}$$

It is equivalent to minimize $G_\lambda$ or $F_{\lambda,b}$, as shown in the following statement.

---

[4]We could consider more general $p$-homogeneous projections as in [LMS18], but we show in Sec. C.2 that we can always bring ourselves back to the case $p = 1$ up to a change of metric.

**Proposition 3.1.** *Let $\nu \in \mathcal{M}(\mathcal{W})$. For any $\mu \in \mathcal{P}_b(\mathcal{W})$ such that $\boldsymbol{h}\mu = \nu$, it holds $F_{\lambda,b}(\mu) \geq G_\lambda(\nu)$, and equality holds for $\mu(\mathrm{d}r, \mathrm{d}w) = \delta_{f(w)}(\mathrm{d}r)\frac{|\nu|(\mathrm{d}w)}{\|\nu\|_{TV}}$ where $f(w) = \|\nu\|_{TV}\frac{\mathrm{d}\nu}{\mathrm{d}|\nu|}(w)$ (and only for this $\mu$ when $b > 1$). In particular, if $G_\lambda$ admits a minimizer then $F_{\lambda,b}$ does too, and it holds*

$$\min_{\mu \in \mathcal{P}_b(\Omega)} F_{\lambda,b}(\mu) = \min_{\nu \in \mathcal{M}(\mathcal{W})} G_\lambda(\nu).$$

It is not difficult to see that $F_{\lambda,b}$ satisfies (P0) as long as $G_\lambda$ admits a minimizer. In order to study (P1) and (P2), we need to define a Riemannian metric on $\Omega$. Following [Chi22c], we consider a general class of Riemannian metrics on $\Omega^* := \mathbb{R}^* \times \mathcal{W}$, parameterized by $q_r, q_w \in \mathbb{R}$ and $\Gamma > 0$, defined by

$$\left\langle \begin{pmatrix} \delta r_1 \\ \delta w_1 \end{pmatrix}, \begin{pmatrix} \delta r_2 \\ \delta w_2 \end{pmatrix} \right\rangle_{(r,w)} = \Gamma^{-1} |r|^{q_r} \frac{\delta r_1 \delta r_2}{r^2} + |r|^{q_w} \langle \delta w_1, \delta w_2 \rangle_w. \tag{3.2}$$

This indeed defines an inner product on $T_{(r,w)}\Omega^* := \mathbb{R} \times T_w \mathcal{W}$ that varies smoothly, and so equips $\Omega^*$ with a (disconnected) Riemannian manifold structure [Lee18]. Intuitively, the parameter $\Gamma$ will govern the relative speed of the weight or position variables along gradient flows; larger $\Gamma$ means faster weight updates.

Two particular cases of this construction appear (sometimes implicitly) in the literature on 2NN:

(i) when $q_r = 2$ and $q_w = 0$, the metric (3.2) extends to the product metric on $\Omega = \mathbb{R} \times \mathcal{W}$. With $\mathcal{W} = \mathbb{R}^{d+1}$, this corresponds to the usual parameterization of 2NNs and is the setting of most previous works applying MFLD to 2NN (with a weight decay regularization on the second layer for $b = 2$ and $\lambda > 0$).

(ii) when $q_r = q_w = 1$, $\Omega^*$ is isometric to the union of two copies of the (tipless) metric cone over $\mathcal{W}$ [BBI01] (via the mapping $(r, \omega) \mapsto (\mathrm{sign}(r), \sqrt{|r|}, \omega)$). This is the natural setting for optimization over signed measures; and with $\mathcal{W} = \mathbb{S}^d$, is equivalent to the parameterization of 2NNs with ReLU activation and balanced initialization [CB20, App. H].

**Issues caused by the disconnectedness of $\Omega^*$.** On the level of the equivalence of variational problems, one can check that the statement of Prop. 3.1 also holds if $\Omega = \mathbb{R} \times \mathcal{W}$ is replaced by $\Omega^* = \mathbb{R}^* \times \mathcal{W}$. However, when the manifold $\Omega^*$ is truly disconnected,[5] then $\mathcal{P}(\Omega)$ is not connected in the sense of absolutely continuous curves in Wasserstein space. More precisely, $\Omega^*$ is the disjoint union of $\Omega_+^* = \mathbb{R}_+^* \times \mathcal{W}$ and $\Omega_-^* = \mathbb{R}_-^* \times \mathcal{W}$, and one can show that (for certain choices of $q_r, q_w$), if $(\mu_t)_t$ is a Wasserstein gradient flow (or any other absolutely continuous curve), then $\mu_t(\Omega_+^*) = \mu_0(\Omega_+^*)$ for all $t$.

Moreover, supposing for simplicity that $G_\lambda$ has a unique minimizer $\nu$ and that $b > 1$, then $F_{\lambda,b}$ has a unique minimizer $\mu^*$, and $\mu^*(\Omega_+^*) = \nu_+(\mathcal{W})/\|\nu\|_{TV}$ where $\nu = \nu_+ - \nu_-$ is the Jordan decomposition of $\nu$. Therefore, Wasserstein gradient flow for $F_{\lambda,b}$ can only converge to $\mu^*$ if it was initialized such that $\mu_0(\Omega_+^*) = \mu^*(\Omega_+^*)$. In terms of particle methods, this means that the fraction of the particles $(r_i, w_i)$ initialized with $r_i > 0$ must be precisely $\mu^*(\Omega_+^*)$. A similar problem arises if we apply MFLD to $F_{\lambda,b}$, since it is nothing else than Wasserstein gradient flow for $F_{\lambda,b} + \beta^{-1}H$; but it is more tedious to discuss formally, as $F_{\lambda,b} + \beta^{-1}H$ does not have a minimizer in general.

In order to bypass this limitation, one may focus on settings where the ratio $\nu_+(\mathcal{W})/\|\nu\|_{TV}$ for the optimal $\nu$ is known in advance, e.g., the problem (1.1) constrained to non-negative measures, or on choices of $q_r, q_w$ for which $\Omega^*$ can be extended into a connected manifold, such as the product metric $q_r = 2, q_w = 0$. However, even in those cases, MFLD on $F_{\lambda,b}$ presents other limitations.

**Incompatibility with MFLD.** We now show that, in spite of the degrees of freedom given by the parameters $q_r, q_w$ and $b$, satisfying both (P1) and (P2) requires restrictive assumptions. This suggests that the lifting approach is fundamentally incompatible with MFLD.

---

[5]This issue also occurs in the case $q_r = q_w = 1$, even though $\Omega^*$ can be completed into a topologically connected set by adding an element 0 "bridging" the two cones $\Omega_+^*$ and $\Omega_-^*$. Indeed, any particle reaching 0 remains at 0 for all subsequent times. Besides, this completion is not itself a manifold, as 0 is a singularity.

**Proposition 3.2.** *Consider $F_{\lambda,b}$ from Eq. (3.1) and $\Omega^*$ equipped with the metric (3.2). Suppose $G'[\nu]$ is continuous for all $\nu$ and that there exists $\nu$ such that $\nabla^2 G'[\nu]$ is not constant equal to $0$. Then*

- *If $q_r \neq 1$ or $q_w \neq 1$ or $b \neq 1$, then (P1) does not hold.*

- *If $q_r = q_w = b = 1$, then for any $\mu \in \mathcal{P}_1(\Omega)$, there exists $\lambda_0 > 0$ such that $F_{\lambda,b} + \beta^{-1}H$ does not satisfy local LSI at $\mu$ for any $\lambda < \lambda_0$ (in particular (P2) does not hold unless $\lambda$ is large enough).*

When $q_r = q_w = b = 1$ and $\lambda$ is large enough, then it can indeed be shown that Thm. 2.1 applies under natural conditions, see for instance [Chi22b, Sec. 5.1].

*Remark* 3.1. For functionals of the form $G_{\lambda,s} = G(\nu) + \frac{\lambda}{s}\|\nu\|_{TV}^s$, instead of (1.1) which corresponds to $s = 2$, one can formulate a similar reduction by posing $\Psi_{b,s}(\mu) = (\int_\Omega |r|^b \, \mathrm{d}\mu(r,w))^{s/b}$ and $F_{\lambda,b,s}(\mu) = G(\boldsymbol{h}\mu) + \frac{\lambda}{s}\Psi_{b,s}(\mu)$. The statements of Prop. 3.1 and Prop. 3.2 hold true with $G_\lambda$ replaced by $G_{\lambda,s}$, and $F_{\lambda,b}$ by $F_{\lambda,b,s}$, for any $1 \leq b \leq s$, as can be shown by very simple adaptations of the proofs (only the second inequality in the proof of Lem. C.1, and the definition of $\lambda'$ in (C.2), need to be adapted). Note that the problem considered in [Chi22c] is of the form $G(\nu) + \lambda\|\nu\|_{TV}$, and they analyzed Wasserstein gradient flow on $F_{\lambda,1,1}$ with $q_r = q_w = 1$ (in particular the issues caused by the disconnectedness of $\Omega^*$ are bypassed thanks to the choice $b = 1$). The above discussion shows that applying MFLD to that problem would only yield convergence guarantees for $\lambda$ large enough.

## 3.2 Reduction by bilevel optimization

We define the bilevel objective functional $J_\lambda$ for $\eta \in \mathcal{P}(\mathcal{W})$ as[6]

$$J_\lambda(\eta) \coloneqq \inf_{\nu \in \mathcal{M}(\mathcal{W})} G(\nu) + \frac{\lambda}{2}\int_{\mathcal{W}} \frac{|\nu|^2}{\eta}. \tag{3.3}$$

It can be derived using the variational representation of the squared TV-norm [LCBGJ04; Bac19]: for any $\nu \in \mathcal{M}(\Omega)$, one has $\|\nu\|_{TV}^2 = \min_{\eta \in \mathcal{P}(\mathcal{W})}\int_{\mathcal{W}} \frac{|\nu|^2}{\eta}$. By exchanging infima, it thus holds $\inf_{\nu \in \mathcal{M}(\mathcal{W})} G_\lambda(\nu) = \inf_{\eta \in \mathcal{P}(\mathcal{W}), \nu \in \mathcal{M}(\mathcal{W})} G(\nu) + \frac{\lambda}{2}\int \frac{|\nu|^2}{\eta} = \inf_{\eta \in \mathcal{P}(\mathcal{W})} J_\lambda(\eta)$. Moreover, the objective minimized in (3.3) is jointly convex in $(\eta, \nu)$ and partial minimization preserves convexity, so $J_\lambda$ is convex. Let us gather these crucial remarks in a formal statement.

**Proposition 3.3.** *The bilevel objective $J_\lambda$ is convex and $\inf_{\mathcal{P}(\mathcal{W})} J_\lambda = \inf_{\mathcal{M}(\mathcal{W})} G_\lambda$. Moreover, if $G_\lambda$ admits a minimizer $\nu \in \mathcal{M}(\mathcal{W})$, then $\arg\min J_\lambda = \left\{\frac{|\nu|}{\|\nu\|_{TV}}, \nu \in \arg\min G_\lambda\right\}$.*

**Link between the lifted and bilevel reductions.** The equality case in the statement of Prop. 3.1 shows that we can restrict the lifted reduction to measures $\mu \in \mathcal{P}_b(\Omega)$ of the form $\mu(\mathrm{d}r, \mathrm{d}w) = \delta_{f(w)}(\mathrm{d}r)\eta(\mathrm{d}w)$ for some $f : \mathcal{W} \to \mathbb{R}$ and $\eta \in \mathcal{P}(\mathcal{W})$. Since they satisfy $\boldsymbol{h}\mu(\mathrm{d}w) = f(w)\eta(\mathrm{d}w)$, the lifted reduction with $b = 2$ thus rewrites

$$\min_{\eta \in \mathcal{P}(\mathcal{W})} \min_{f \in L^2(\eta)} G(f\eta) + \frac{\lambda}{2}\int_{\mathcal{W}} f(w)^2 \mathrm{d}\eta(w).$$

After the change of variable $(\nu, \eta) = (f\eta, \eta)$, the outer objective is precisely $J_\lambda(\eta)$. Thus, Wasserstein gradient flow on $J_\lambda$ can be seen as a two-timescale optimization dynamics: it is the Wasserstein gradient flow on $F_{\lambda,2}$ in the limit where $\Gamma \to \infty$. In the context of 2NN training with the parametrization (i), this amounts to training the output layer infinitely faster than the input layer, as done in [BMZ23; MB23; BBP23; TS24]. This remark allows to implement the bilevel MFLD numerically by discretizing in time the system of SDEs, for fixed large $N$ and $\Gamma$,

$$\forall i \leq N, \quad \mathrm{d}r_t^i = -\Gamma \, \nabla_{r^i} F_{\lambda,2}'[\mu_t](r_t^i, w_t^i)\mathrm{d}t \qquad = -\Gamma\left(G'[\nu_t](w_t^i) + \lambda r_t^i\right)\mathrm{d}t \tag{3.4}$$

$$\mathrm{d}w_t^i = -\nabla_{w^i} F_{\lambda,2}'[\mu_t](r_t^i, w_t^i)\mathrm{d}t + \sqrt{2\beta^{-1}}\mathrm{d}B_t^i = -r_t^i \nabla G'[\nu_t](w_t^i)\mathrm{d}t + \sqrt{2\beta^{-1}}\mathrm{d}B_t^i$$

---

[6]We use $\int_{\mathcal{W}} \frac{|\nu|^2}{\eta}$ as a shorthand for $\int_{\mathcal{W}}\left(\frac{\mathrm{d}\nu}{\mathrm{d}\eta}(w)\right)^2 \mathrm{d}\eta(w)$.

where $\mu_t = \frac{1}{N}\sum_{i=1}^N \delta_{(r_t^i, w_t^i)}$ and $\nu_t = \frac{1}{N}\sum_{i=1}^N r_t^i \delta_{w_t^i}$, and taking $\eta_t = \frac{1}{N}\sum_{i=1}^n \delta_{w_t^i}$. Notice the absence of noise term on the weight variables $r$; it reflects the fact that MFLD for the bilevel objective is *not* a limit case of MFLD for the lifted objective, as the noise would prevent to reach optimality in the inner problem.

**Compability with MFLD.** We now show that, in contrast to the lifting reduction, the bilevel reduction is amenable to MFLD. The main assumption on (1.1) is as follows.

**Assumption 1.** $G : \mathcal{M}(\mathcal{W}) \to \mathbb{R}$ is non-negative and admits second variations, and for each $i \in \{0, 1, 2\}$, there exist $L_i, B_i < \infty$ such that $\left\|\nabla^i G''[\nu](w, w')\right\|_w \leq L_i$ and $\left\|\nabla^i G'[\nu]\right\|_w \leq L_i \|\nu\|_{TV} + B_i$ for all $\nu \in \mathcal{M}(\mathcal{W})$ and $w, w' \in \mathcal{W}$. Moreover there exists $\widetilde{L}_2 < \infty$ such that $\|\nabla_w \nabla_{w'} G''[\nu](w, w')\| \leq \widetilde{L}_2$ for all $\nu, w, w'$. Furthermore, $\mathcal{W}$ is compact and the uniform probability measure $\tau$ on $\mathcal{W}$ satisfies LSI with constant $\alpha_\tau$.

Concrete settings that satisfy Assumption 1 are discussed in Sec. 5. The following proposition confirms the compatibility with MFLD and gives quantitative bounds on the LSI constant.

**Proposition 3.4.** *Under Assumption 1, $J_\lambda$ satisfies* (P0)*,* (P1) *and* (P2)*. More precisely, for any* $\eta \in \mathcal{P}(\mathcal{W})$*, $J_\lambda + \beta^{-1} H$ satisfies local LSI at $\eta$ with the constant $\alpha_{\hat{\eta}} = \alpha_\tau \exp\left(-\frac{1}{\lambda}L_0 \beta J_\lambda(\eta)\right)$. Further, $J_\lambda + \beta^{-1} H$ satisfies $\alpha$-LSI uniformly along the MFLD trajectory $(\eta_t)_t$ with the constant* $\alpha = \alpha_\tau \exp\left(-\frac{1}{\lambda}L_0\beta \min\left\{G(0), J_\lambda(\eta_0) + \beta^{-1}H(\eta_0|\tau)\right\}\right)$.

In view of the negative result of Prop. 3.2 for the lifting reduction, and the positive result of Prop. 3.4 for the bilevel reduction, in the sequel we focus on MFLD applied on $J_\lambda$, which we will refer to as MFLD-Bilevel.

# 4 Global convergence and annealing for MFLD-Bilevel

While the bounds from Prop. 3.4 along with Thm. 2.1 allow to establish global convergence to minimizers of $J_\lambda + \beta^{-1}H$, our aim is to minimize the unregularized bilevel objective $J_\lambda$. This can be achieved by annealing the temperature parameter $\beta^{-1}$ along the dynamics. Namely, Theorem 4.1 of [Chi22b] guarantees that by choosing $\beta_t = c\log(t)$ for an appropriate constant $c$, the annealed MFLD trajectory

$$\partial_t \eta_t = \mathrm{div}(\eta_t \nabla J'_\lambda[\eta_t]) + \beta_t^{-1}\Delta \eta_t$$

satisfies $J_\lambda(\eta_t) - \inf J_\lambda = O\left(\frac{\log\log t}{\log t}\right)$. This is a very slow rate however.

In this section, we show that the structure of $J_\lambda$ originating from the bilevel reduction can be exploited to go beyond the generic guarantees from [Chi22b, Thm. 4.1]. Namely, we study in detail an alternative temperature annealing strategy, and we show that it improves upon the classical one $\beta_t \sim \log(t)$ in terms of convergence to a fixed multiplicative accuracy.

## 4.1 Faster convergence to a fixed multiplicative accuracy

**Definition 4.1.** Suppose $0 \notin \arg\min G$, so that $J_\lambda^* := \inf J_\lambda > 0$. We will say that MFLD-Bilevel with a given temperature annealing schedule $(\beta_t)_{\geq 0}$ *converges to* $(1 + \Delta)$*-multiplicative accuracy in time-complexity* $T_\Delta$, for a fixed positive constant $\Delta$ (say $\Delta = 0.01$), if $J_\lambda(\eta_{T_\Delta}) \leq (1 + \Delta)J_\lambda^*$.

Note that in machine learning settings where the problem (1.1) corresponds to learning with overparameterized models, it is realistic to assume $J_\lambda^*$ to be small (as long as the regularization $\lambda$ is small), and $T_\Delta$ is the time it takes for the annealed MFLD to achieve a suboptimality of at most $\Delta J_\lambda^*$.

For ease of comparison, let us report the time-complexity $T_\Delta$ that can be achieved by simply running MFLD-Bilevel with a constant but well-chosen $\beta$, based on the bounds from Prop. 3.4 and Thm. 2.1.

**Proposition 4.1** (Baseline "annealing" schedule: constant $\beta_t$)**.** *Under Assumption 1, let $\Delta > 0$ and assume that $\Delta \leq \frac{L_0 L_1 G(0)}{\lambda^2 J_\lambda^*}$. Then, MFLD-Bilevel with the temperature schedule $\forall t, \beta_t = \frac{4d}{\Delta J_\lambda^*}\log\left(\frac{CB}{\Delta J_\lambda^*}\right)$ converges to $(1 + \Delta)$-multiplicative accuracy in time*

$$T_\Delta \leq \frac{C'}{\Delta J_\lambda^*}\log\left(\frac{CB}{\Delta J_\lambda^*}\right) \cdot \exp\left(\frac{C' L_0 G(0)}{\lambda \, \Delta J_\lambda^*}\log\left(\frac{CB}{\Delta J_\lambda^*}\right)\right) \cdot \log\left(\frac{2G(0)}{\Delta J_\lambda^*} + C' H(\eta_0|\tau)\right)$$

*where $B = \mathrm{poly}(L_0, L_1, B_1, G(0), \lambda^{-1})$ and $C, C'$ are constants dependent on $\mathcal{W}$ (and $d$ and $\alpha_\tau$).*

For the annealing schedule $\beta_t \sim \log(t)$, the time-complexity $T_\Delta$ that can be guaranteed from inspecting the proof of [Chi22b, Thm. 4.1] has the same dependency on $d, \lambda$ and $J_\lambda^*$ as for the baseline $\beta_t = \mathrm{cst}$.

**Improved annealing schedule.** Recall the result of Prop. 3.4: for any $\beta > 0$, $J_\lambda + \beta^{-1}H$ satisfies local $\alpha_{\hat{\eta}}$-LSI at $\eta$ with $\alpha_{\hat{\eta}} = \alpha_\tau \exp(-\frac{L_0}{\lambda}\beta J_\lambda(\eta))$. Informally, if we manage to control $J_\lambda(\eta_t)$ along the annealed MFLD trajectory and show that it decreases, then we can increase $\beta_t$ at the same rate, while retaining the same local LSI constant. This observation and the resulting annealing procedure were introduced in [SWON23], in a 2NN classification setting with the logistic loss. There the optimal value of the loss functional, corresponding to our $J_\lambda^*$, is 0, and the annealing procedure yields favorable rates for global convergence. Here we show that this procedure is also applicable for MFLD-Bilevel, as soon as $G$ satisfies the mild Assumption 1, yielding favorable rates for convergence to a fixed multiplicative accuracy.[7]

**Theorem 4.2.** *Under Assumption 1, there exist constants $B = \mathrm{poly}(L_i, B_i, G(0), \lambda^{-1})$ and $C_i$ dependent only on $G(0)$, $H(\eta_0)$, $\mathcal{W}$ (and $d$ and $\alpha_\tau$) such that the following holds. For any $\Delta \leq \frac{B}{J_\lambda^*}$, MFLD-Bilevel with the temperature schedule $(\beta_t)_{t \geq 0}$ defined by $\forall k \leq K, \forall t \in [t_k, t_{k+1}], \beta_t = 2^k d$ where $t_0 = 0$ and $K = \lceil 2 \log_2(B/(\Delta J_\lambda^*)) \rceil$ and*

$$t_{k+1} - t_k = C_1 2^k \, k \cdot \exp\left( \frac{L_0 d}{\lambda} \left( \frac{C_3}{\Delta} \log\left( \frac{B}{\Delta J_\lambda^*} \right) + C_2 \right) \right),$$

*achieves $(1 + \Delta)$-multiplicative accuracy, with time-complexity*

$$T_\Delta \leq t_{K+1} \leq \frac{C_4}{\Delta J_\lambda^*} \log\left( \frac{B}{\Delta J_\lambda^*} \right)^2 \cdot \exp\left( \frac{L_0 d}{\lambda} \left( \frac{C_3}{\Delta} \log\left( \frac{B}{\Delta J_\lambda^*} \right) + C_2 \right) \right).$$

Note that assuming that $G$ admits a minimizer $\nu_0$ and that $\min G = 0$, as is typically the case in over-parametrized machine learning settings, then by the envelope theorem $J_\lambda^* = \inf\left( G + \frac{\lambda}{2} \|\cdot\|_{TV}^2 \right) = \frac{\|\nu_0\|_{TV}^2}{2}\lambda + o(\lambda)$. So in the regime of small $\lambda$, ignoring the subexponential factors, the time complexity bound achieved by the annealing schedule of Thm. 4.2 scales as $\exp\left( c\lambda^{-1} \log \lambda^{-1} \right)$ for a constant $c$. This improves upon the time complexity bound of the classical annealing procedure $\beta_t \sim \log(t)$ (the same as in Prop. 4.1), which scales as $\exp(c'\lambda^{-2})$.

## 5 Local LSI constant at optimality for learning a single neuron

Devising temperature annealing schemes for global convergence, as illustrated in the previous section, relies on bounds on the local LSI constant at every iterate $\eta_t$ of the (annealed) MFLD. Such bounds are readily provided by the widely applicable Holley-Stroock perturbation argument, on which for example our Prop. 3.4 is based, but may be overly pessimistic. Indeed in this section, we demonstrate that for MFLD-Bilevel, *the LSI constant at convergence can be independent of $\beta$, $\lambda$ and $d$, instead of exponential in $\beta$* as a global analysis would suggest.

More precisely, we are interested in $\alpha^*$, the best local LSI constant of $J_{\lambda,\beta} := J_\lambda + \beta^{-1}H(\cdot|\tau)$, at $\eta_{\lambda,\beta} := \arg\min J_{\lambda,\beta}$. In fact the proximal Gibbs measure of the optimum is the optimum itself: $\widehat{\eta_{\lambda,\beta}} = \eta_{\lambda,\beta}$, so $\alpha^*$ is precisely the LSI constant of $\eta_{\lambda,\beta}$. A bound on $\alpha^*$ is of interest, especially in the regime of large $\beta$ (low entropic regularization), for two reasons. Firstly, it directly implies a local convergence bound on MFLD-Bilevel, as shown in the proposition below. Secondly, characterizing the dependency of $\alpha^*$ on $\beta$ may open the way to more efficient temperature annealing strategies; but this is out of the scope of this paper.

**Proposition 5.1.** *Under Assumption 1, suppose $\eta_{\lambda,\beta}$ satisfies LSI with some constant $\alpha_\beta^*$. For any $\varepsilon > 0$, there exists a sublevel set of $J_{\lambda,\beta}$ such that, for any initialization $\eta_0$ in this sublevel set,*

$$J_{\lambda,\beta}(\eta_t) - \inf J_{\lambda,\beta} \leq (J_{\lambda,\beta}(\eta_0) - \inf J_{\lambda,\beta}) \, e^{-\left(\alpha_\beta^* \beta^{-1} - \varepsilon\right)t}.$$

---

[7]In fact, the annealing procedure of Thm. 4.2 would also yield a rate of convergence for any $\mathcal{J} : \mathcal{P}(\mathcal{W}) \to \mathbb{R}$ with $\mathcal{J}''[\eta](w, w')$ uniformly bounded and $\inf \mathcal{J} > 0$, instead of $J_\lambda$; but the resulting bound on $T_\Delta$ would have an additional factor of $(\inf \mathcal{J})^{1/2}$ inside the exponential. See Sec. E.2 for a detailed discussion.

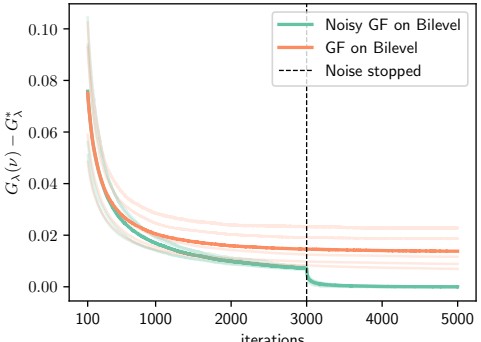 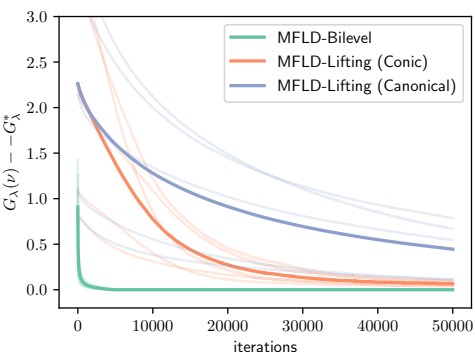

(a) A comparison of Wasserstein GF on the bilevel objective with (i.e. MFLD) and without noise.

(b) A comparison of MFLD applied to the Bilevel vs. Lifted formulations.

Figure 1: The regularized training loss $G_\lambda(\nu)$ (1.1) of a 2NN with the ReLU activation, learning a teacher 2NN with the 4th degree Hermite polynomial as its activation. In both plots, $d = 10$ and $\lambda = \beta^{-1} = 10^{-3}$. The implementation details are provided in Sec. F.4. Plots are averaged over 5 experiments. $G_\lambda^*$ is the best value achieved at each experiment. In Fig. (1b), "Conic" refers to using the metric (3.2) with $q_r = 1, q_w = 1$, while "Canonical" refers to the choice of $q_r = 2, q_w = 0$.

For the local LSI analysis, we focus on a specific setting of (1.1), namely, least-squares regression using a 2NN with a normalization constraint on the first-layer weights, and a single-neuron teacher network. See Fig. 1 for an illustrative numerical experiment. Note that Assumption 2, with additional bounded-moment assumptions on $\varphi$ and $\rho$, is a special case of Assumption 1, as shown in Prop. F.4.

**Assumption 2.** $\mathcal{W} = \mathbb{S}^d$ is the Euclidean sphere in $\mathbb{R}^{d+1}$ and there exist $\rho$ a covariate distribution over $\mathbb{R}^{d+1}$, $y \in L_\rho^2(\mathbb{R}^{d+1})$ a fixed target function, and $\varphi : \mathbb{R} \to \mathbb{R}$ a $\mathcal{C}^2$ activation function such that $G(\nu) = \frac{1}{2}\mathbb{E}_{x\sim\rho}|\hat{y}_\nu(x) - y(x)|^2$ where $\hat{y}_\nu(x) = \int_{\mathcal{W}} \varphi(\langle w, x\rangle)\mathrm{d}\nu(w)$.

Under the above assumption, we show in Prop. F.1 a simplified expression for the bilevel objective and its first variation,

$$J_\lambda(\eta) = \frac{\lambda}{2}\langle y, (K_\eta + \lambda\,\mathrm{id})^{-1}y\rangle_{L_\rho^2}, \qquad J_\lambda'[\eta](w) = -\frac{\lambda}{2}\langle\varphi(\langle w, \cdot\rangle), (K_\eta + \lambda\,\mathrm{id})^{-1}y\rangle_{L_\rho^2}^2,$$

where $K_\eta$ is the integral operator in $L_\rho^2$ of the kernel $k_\eta(x, x') = \int \varphi(\langle w, x\rangle)\varphi(\langle w, x'\rangle)\mathrm{d}\eta(w)$ and id is the identity operator on $L_\rho^2$. Additionally, we make the following assumption on the data distribution $\rho$ and on the response $y$.

**Assumption 3.** $\rho$ is rotationally invariant and the labels come from a single-index model: $y = \varphi(\langle v, x\rangle)$ for some fixed $v \in \mathcal{W}$.

With the above assumptions, we can state the main theorem of this section.

**Theorem 5.2.** *Under Assumptions 2 and 3, there exists a function $g : [-1, +1] \to \mathbb{R}_+$ such that $J_\lambda'[\delta_v](w) = -\lambda g(\langle w, v\rangle)$ for any $w \in \mathbb{S}^d$. Suppose that $\lambda \leq 1$ and that there exist constants $c_i, C_i > 0$ such that for all $r \in [-1, +1]$,*

$$c_1 \leq g'(r) \leq C_1, \quad g''(r) \geq -C_2, \quad \left|g''(r)(1-r^2)^{1/2}\right| \leq C_3, \quad \left|g'''(r)(1-r^2)^{3/2}\right| \leq C_4.$$

*Then there exist constants $\alpha_v$, $D_0$ (dependent only on the $c_i, C_i$) such that for any $\beta \geq D_0 d\lambda^{-1}$, $\widehat{\delta_v} \propto e^{-\beta J_\lambda'[\delta_v]}\tau$ satisfies $\alpha_v$-LSI. Furthermore, if additionally $\frac{1}{d^2}\mathbb{E}_{x\sim\rho}\|x\|^4, \|\varphi^{(i)}\|_{L^4(\rho)} < \infty$ for $i \in \{0, 1, 2\}$ where $\|\varphi\|_{L^p(\rho)}^p := \int |\varphi(\langle w, x\rangle)|^p\mathrm{d}\rho(x)$ (independent of $w$ as $\rho$ is rotationally invariant), then there exists a constant $\alpha^*$ dependent only on those constants and on the $c_i, C_i$ such that, provided that $\beta \geq \mathrm{poly}(d, \lambda^{-1})$, $\eta_{\lambda,\beta}$ satisfies $\alpha^*$-LSI.*

The proof is based on the observation that $\eta_{\lambda,\beta} \approx \arg\min J_\lambda = \delta_v$ the Dirac measure at $v$, for certain regimes of $\beta$ and $\lambda$, in the Wasserstein metric. Thus we show that $J_\lambda'[\delta_v]$ is amenable to a Lyapunov type argument inspired from [MS14; LE23], and then transfer its properties to $J_\lambda'[\eta_{\lambda,\beta}]$.

We now verify the assumptions of Thm. 5.2 for a class of smooth, non-negative, and monotone activations which includes some popular practical choices such as the Softplus $\varphi(z) = \ln(1 + e^z)$ and sigmoid $\varphi(z) = 1/(1 + e^{-z})$. While we only consider smooth activations here for simplicity, certain non-smooth activations such as a leaky version of ReLU can also satisfy the conditions of Thm. 5.2.

**Proposition 5.3.** *Suppose Assumptions 2 and 3 hold, and $b_1(d+1) \leq \mathbb{E}[\|x\|^2] \leq \mathbb{E}[\|x\|^{12}]^{1/6} \leq b_2(d+1)$ for constants $b_1, b_2 > 0$. Let $m \coloneqq 2b_2^{3/2}/b_1$. Suppose $\varphi$ and $\varphi'$ are non-negative, $\inf_{|z| \leq m} \varphi(z) \wedge \varphi'(z) > 0$ and $\left\|\varphi^{(i)}\right\|_{L^4(\rho)} < \infty$ for $i \leq 3$. Then, $\varphi$ satisfies the assumptions of Thm. 5.2 with constants that only depend on $b_1$, $b_2$, and $\varphi$.*

## 6   Conclusion

In this paper, we investigated how mean-field Langevin dynamics (MFLD), an optimization dynamics over probability measures with global convergence guarantees, can be leveraged to solve convex optimization problems over signed measures of the form (1.1). For a large class of objectives $G$, we highlighted that MFLD with a lifting approach necessarily runs into some issues, whereas the bilevel approach always inherits the guarantees of MFLD, leading to convergence guarantees for $G_\lambda$ via annealing. Finally, turning to a 2-layer NN learning task which can be stated as an instance of (1.1), we showed that the local LSI constant of MFLD-Bilevel can scale much more favorably with $d$ and $\beta$ than a generic analysis would suggest.

Another approach to tackle (1.1) could be to build noisy particle dynamics directly in the space of signed measures, complementing the MFLD updates with, for instance, a birth-death process. A challenge then is to build such dynamics that can be efficiently discretized. It is also an interesting question for future works to find other settings to which MFLD can be extended, beyond signed measures.

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

# A   Details for Sec. 1 (introduction)

## A.1   Using $\|\cdot\|_{TV}^2$ vs. $\|\cdot\|_{TV}$ as the regularization term

The optimization problems we consider in this paper are of the form (1.1), that is, for ease of reference,

$$\min_{\nu \in \mathcal{M}(\mathcal{W})} G_\lambda(\nu), \qquad\qquad G_\lambda(\nu) := G(\nu) + \frac{\lambda}{2} \|\nu\|_{TV}^2 .$$

Note the regularization term $\frac{\lambda}{2} \|\nu\|_{TV}^2$. This is to be contrasted with the more usual form of optimization problems

$$\min_{\nu \in \mathcal{M}(\mathcal{W})} \widetilde{G}_{\tilde\lambda}(\nu), \qquad\qquad \widetilde{G}_{\tilde\lambda}(\nu) := G(\nu) + \tilde\lambda \|\nu\|_{TV} ,$$

which uses $\|\nu\|_{TV}$ as the regularization.

On the level of variational problems, these two classes of problems are equivalent, in the sense that

$$\{0\} \cup \bigcup_{\lambda \geq 0} \arg\min G_\lambda = \{0\} \cup \bigcup_{\tilde\lambda \geq 0} \arg\min \widetilde{G}_{\tilde\lambda}$$

where "0" refers to the zero measure on $\mathcal{W}$. Indeed, note that by convexity, the argmins are determined by the respective first-order optimality conditions, so that

$$\bigcup_{\lambda \geq 0} \arg\min G_\lambda = \left\{ \nu \in \mathcal{M}(\mathcal{W}); \ \forall w, G'[\nu](w) + \lambda \|\nu\|_{TV} \frac{\nu(\mathrm{d}w)}{|\nu(\mathrm{d}w)|} = 0, \ \lambda \in \mathbb{R}_+ \right\}$$

$$\bigcup_{\tilde\lambda \geq 0} \arg\min \widetilde{G}_{\tilde\lambda} = \left\{ \nu \in \mathcal{M}(\mathcal{W}); \ \forall w, G'[\nu](w) + \tilde\lambda \frac{\nu(\mathrm{d}w)}{|\nu(\mathrm{d}w)|} = 0, \ \tilde\lambda \in \mathbb{R}_+ \right\} .$$

To see that the set on the first line is contained in the second, let $\nu \in \arg\min G_\lambda$, then $\nu$ satisfies the first-order optimality condition for $\widetilde{G}_{\tilde\lambda}$ with $\tilde\lambda = \lambda \|\nu\|_{TV}$. Conversely, if $\nu \in \arg\min \widetilde{G}_{\tilde\lambda}$ then either $\nu = 0$ or $\nu \in \arg\min G_\lambda$ with $\lambda = \frac{\tilde\lambda}{\|\nu\|_{TV}}$.

In terms of optimization convergence guarantees, when using the reduction by lifting, the problems with $\|\cdot\|_{TV}$ vs. with $\|\cdot\|_{TV}^2$ regularization give rise to similar analyses, as discussed in Rem. 3.1. However when using the reduction by bilevel optimization, it seems that only the problem with $\|\cdot\|_{TV}^2$ regularization is amenable to a precise analysis. This is perhaps most apparent in our derivation of the simplified expression for the bilevel objective, Prop. D.2.

## A.2   Detailed comparison with Takakura and Suzuki [TS24]

In this subsection, we show that the learning dynamics considered by [TS24, Sec. 2, 3] is an instance of a variant of MFLD applied to the bilevel reduction of (1.1). We do this by recalling their setting (in the case of single-task learning for simplicity) in notations that are compatible with ours.

- For a set of first-layer weights $w_i \in \mathcal{W} := \mathbb{R}^d$ and second-layer weights $a_i \in \mathbb{R}$ (for $1 \leq i \leq N$), and an activation function $\varphi : \mathbb{R} \to \mathbb{R}$, the associated 2NN is defined as $x \mapsto \frac{1}{N} \sum_{i=1}^N a_i \varphi(w_i^\top x)$.

- For $\mu \in \mathcal{P}(\mathbb{R} \times \mathcal{W})$, the associated infinite-width 2NN is $x \mapsto \int_{\mathbb{R} \times \mathcal{W}} a\varphi(w^\top x)\mathrm{d}\mu(a, w)$. Note that in our notation of Sec. 3.1, this also writes $x \mapsto \int_{\mathcal{W}} \varphi(w^\top x)\mathrm{d}[\boldsymbol{h}\mu](w)$.

- Consider a data distribution $\rho(\mathrm{d}x, \mathrm{d}y) \in \mathcal{P}(\mathbb{R}_x^d \times \mathbb{R}_y)$. We may define the Hilbert space of predictors $\mathcal{H} = L_{\rho^x}^2(\mathbb{R}_x^d)$, and the "single first-layer neuron predictor" mapping $\phi : \mathcal{W} \to \mathcal{H}$ by $\phi(w)(x) = \varphi(w^\top x)$. The predictor associated to an infinite-width 2NN parametrized by $\mu$ is then $\int_{\mathbb{R} \times \mathcal{W}} a\phi(w)\mathrm{d}\mu(a, w)$.

- Consider a loss function $\ell(\hat{y}, y) : \mathbb{R}_y \times \mathbb{R}_y \to \mathbb{R}$, inducing a risk functional over predictors given by $R(h) = \mathbb{E}_{(x,y) \sim \rho}[\ell(h(x), y)]$. We may define the unregularized risk functional over (infinite-width) 2NN weights by

$$\mathcal{L}(\mu) = R\left(\int_{\mathbb{R} \times \mathcal{W}} a\phi(w)\mathrm{d}\mu(a, w)\right) = R\left(\int_{\mathcal{W}} \phi(w)\mathrm{d}[\boldsymbol{h}\mu](w)\right).$$

Accordingly, let the operator $\Phi : \mathcal{M}(\mathcal{W}) \to \mathcal{H}$ such that $\Phi\nu = \int_{\mathcal{W}} \phi(w)\mathrm{d}\nu(w)$, and

$$G(\nu) = R(\Phi\nu) = R\left(\int_{\mathcal{W}} \phi(w)\mathrm{d}\nu(w)\right).$$

Then the unregularized risk is $\mathcal{L}(\mu) = G(\boldsymbol{h}\mu)$.

- The regularized risk functional considered in [TS24, Sec. 2.1] is

$$\mathcal{F}(\mu) = R\left(\int_{\mathbb{R} \times \mathcal{W}} a\phi(w)\mathrm{d}\mu(a, w)\right) + \frac{\lambda}{2}\int_{\mathbb{R} \times \mathcal{W}} a^2\mathrm{d}\mu(a, w) + \frac{1}{2\sigma^2}\int_{\mathbb{R} \times \mathcal{W}} \|w\|^2\,\mathrm{d}\mu(a, w) \tag{A.1}$$

$$= G(\boldsymbol{h}\mu) + \frac{\lambda}{2}\int_{\mathbb{R} \times \mathcal{W}} a^2\mathrm{d}\mu(a, w) + \frac{1}{2\sigma^2}\int_{\mathbb{R} \times \mathcal{W}} \|w\|^2\,\mathrm{d}\mu(a, w).$$

(More precisely, "$F(f, \eta)$" in their notation corresponds to our $\mathcal{F}\left(\delta_{f(w)}(\mathrm{d}a)\eta(\mathrm{d}w)\right)$, their "$\overline{\lambda}_a$" corresponds to our $\lambda$, and their "$\overline{\lambda}_w$" corresponds to our $1/\sigma^2$.) Note that, in our notation of Sec. 3.1,

$$\mathcal{F}(\mu) = F_{\lambda,2}(\mu) + \frac{1}{2\sigma^2}\int_{\mathbb{R} \times \mathcal{W}} \|w\|^2\,\mathrm{d}\mu(a, w).$$

- The bilevel limiting functional, which is the main object of study of [TS24, Sec. 2.1], is then defined as the mapping $\mathcal{G} : \mathcal{P}(\mathcal{W}) \to \mathbb{R}$ such that

$$\mathcal{G}(\eta) = \inf_{f:\mathcal{W} \to \mathbb{R}} \mathcal{F}\left(\delta_{f(w)}(\mathrm{d}r)\eta(\mathrm{d}w)\right), \quad \text{corresponding precisely to}$$

$$\mathcal{G}(\eta) = J_\lambda(\eta) + \frac{1}{2\sigma^2}\int_{\mathcal{W}} \|w\|^2\,\mathrm{d}\eta(w)$$

in our notation of Sec. 3.2 (see the paragraph "Link between the lifted and bilevel reductions"). Interestingly, the convexity of $\mathcal{G}$ is almost immediate with our presentation, as it is expressed as a partial minimization of a convex function, whereas the proof of the convexity of $\mathcal{G}$ in [TS24] is quite involved.

They also introduce a functional "$U$" which corresponds precisely to our $J_\lambda(\eta)$, and which is an important auxiliary object in their analysis.

- The learning dynamics studied from Section 2.3 onwards in [TS24] (except for the label noise procedure in Section 5), is precisely MFLD for $\mathcal{G}(\eta)$:

$$\partial_t \eta_t = \beta^{-1}\Delta\eta_t + \mathrm{div}\left(\eta_t \nabla\mathcal{G}'[\eta_t]\right)$$

$$= \beta^{-1}\Delta\eta_t + \mathrm{div}\left(\eta_t\left(\nabla J'_\lambda[\eta_t] + \frac{1}{\sigma^2}w\right)\right) \tag{A.2}$$

(and their constant "$\lambda$" corresponds to our $\beta^{-1}$).

**"MFL + confining" dynamics.** The PDE (A.2) can be interpreted as a variant of MFLD for $J_\lambda$ in two ultimately equivalent ways: one is as the MFLD PDE (2.1) with an added "confining" term $-\frac{1}{\sigma^2}w$, which intuitively encourages the noisy particles to remain close to the origin. Another is as Wasserstein gradient flow for the regularized functional

$$J_{\lambda,\beta,\sigma} = J_\lambda + \beta^{-1}H + \frac{1}{2\sigma^2}\int_{\mathcal{W}} \|w\|^2\,\mathrm{d}\eta(w)$$

$$= J_\lambda + \beta^{-1}H\left(\cdot\,\middle|\,\beta^{-1/2}\sigma\gamma\right) \quad \text{where} \quad \beta^{-1/2}\sigma\gamma := \mathcal{N}(0, \beta^{-1}\sigma^2 I_d),$$

whereas MFLD for $J_\lambda$ is the Wasserstein gradient flow for the functional regularized by entropy only, $J_{\lambda,\beta} = J_\lambda + \beta^{-1} H(\cdot|\tau) = J_\lambda + \beta^{-1} H + \text{cst}$. Unsurprisingly in view of this second interpretation, the distribution $\beta^{-1/2}\sigma\gamma$ plays a similar role in the analysis of convergence of (A.2) [TS24, Lemma 3.5], as played by the uniform measure $\tau$ in our paper: the local LSI property of $J_{\lambda,\beta,\sigma}$ (resp. $J_{\lambda,\beta}$) is obtained by applying the Holley-Stroock argument using $\beta^{-1/2}\sigma\gamma$ (resp. $\tau$) as a reference measure.

Note that the additional confining term $-\frac{1}{\sigma^2}w$ in (A.2) cannot be captured straightforwardly by any additional penalty term on the objective $G$ from (1.1). Indeed, informally, the three terms in (A.1) each have a different homogeneity in the variable $a$. Rather, the confining term in $\sigma$ should be viewed as corresponding to another regularization term added to (1.3), besides the entropy one in $\beta^{-1}$.

In short, while our work considers MFLD i.e. Wasserstein gradient flow for $F + \beta^{-1} H$ as the main "algorithmic primitive", the work of [TS24] considers a MFL+confining dynamics, i.e. Wasserstein gradient flow for $F + \beta^{-1} H\left(\cdot\middle|\beta\sigma^2\gamma\right)$.

**Summary of differences.** On a technical level, the learning dynamics considered by [TS24] corresponds to a special case of a variant of the MFLD-bilevel we consider from Sec. 3.2 onwards. Namely, they focus on instances of the problem (1.1) where $G$ has a particular form, corresponding to learning with 2NN; and they consider $\mathcal{W} = \mathbb{R}^d$ and use an additional confining term $-\frac{1}{\sigma^2}w$ in the MFLD dynamics, while we consider settings where $\mathcal{W}$ is a compact Riemannian manifold, and no additional confining term is needed.

We also emphasize that, while our work and that of [TS24] cover some similar settings, our focus is quite different. In that work, the key object of interest is the kernel that is learned by MFLD in a 2NN setting $((x, x') \mapsto \int \varphi(x^\top w)\varphi(x^\top w')\mathrm{d}\eta(w)$ in the notation of our second bullet point above). By contrast, our main motivation is a general optimization question: how to use MFLD as an algorithmic primitive for problems of the form (1.1). In particular we do not assume a particular form for $G$ except in Sec. 5, and we pay special attention to the bounds on the local LSI constants of $J_\lambda$ along the MFLD trajectory, instead of using the global uniform LSI bound (compare Prop. 3.4 and [TS24, Lemma 3.5]).

# B Details for Sec. 2 (background about MFLD)

## B.1 The displacement smoothness property

For MFLD (Eq. (2.1)) to be well-posed, we require that $F$ is $L$-smooth along Wasserstein geodesics for some $L < +\infty$. More precisely, for any constant-speed Wasserstein geodesic $(\mu_t)_{t\in[0,1]} \subset \mathcal{P}_2(\Omega)$ with $W_2(\mu_0, \mu_1) = 1$, $t \mapsto F(\mu_t)$ should be $L$-smooth in the usual sense of continuous optimization. This property ensures that the PDE defining MFLD has a unique solution [Chi22b, App. A], and is also helpful to ensure convergence of explicit time-discretization schemes [SWN23]. The following proposition gives a practical sufficient condition.

**Proposition B.1.** *Suppose* $F : \mathcal{P}_2((\Omega, g)) \to \mathbb{R}$ *is twice differentiable in the Wasserstein sense. Let* $0 \leq L < \infty$. *Suppose that* $F$ *satisfies* (P1), *i.e.,*

$$\forall\mu \in \mathcal{P}_2(\Omega),\ \forall\omega \in \Omega,\ \max_{\substack{s\in T_\omega\Omega \\ \|s\|_\omega \leq 1}} \left|\nabla^2 F'[\mu](s, s)\right| \leq L$$

$$\text{and}\quad \forall\mu, \mu' \in \mathcal{P}_2(\Omega),\ \forall\omega \in \Omega,\ \left\|\nabla F'[\mu] - \nabla F'[\mu']\right\|_\omega \leq L\, W_2(\mu, \mu')$$

*where* $\nabla^2$ *denotes the Riemannian Hessian. Then* $F$ *is* $2L$-*smooth along Wasserstein geodesics.*

The first condition can be stated as $F'[\mu] : \Omega \to \mathbb{R}$ having Lipschitz-continuous gradients in the Riemannian sense [Bou23, Coroll. 10.47], whereas the second condition can be interpreted as a displacement Lipschitz-continuity of $\mu \mapsto F'[\mu](\omega)$ for each $\omega$ uniformly.

*Proof.* Let a constant-speed Wasserstein geodesic $(\mu_t)_{t\in[0,1]} \subset \mathcal{P}_2(\Omega)$ with $W_2(\mu_0, \mu_1) = 1$, and pose $f(t) = F(\mu_t)$. We want to show that $f$ is $2L$-smooth in the usual sense of continuous optimization, for which it suffices to show that $\forall t,\ |f''(t)| \leq 2L$.

By [Vil09, Eq. (13.6)] there exist functions $\phi_t : \Omega \to \mathbb{R}$ such that $\begin{cases} \partial_t \mu_t = -\text{div}(\nabla \phi_t \mu_t) \\ \partial_t \phi_t = -\frac{1}{2} \|\nabla \phi_t\|^2 \end{cases}$ and $\int d\mu_t \|\nabla \phi_t\|^2 = W_2^2(\mu_0, \mu_1) = 1$ for all $t$. So we can compute explicitly:

$$f'(t) = \frac{d}{dt} F(\mu_t) = \int d\mu_t \ \langle \nabla F'[\mu_t], \nabla \phi_t \rangle$$

$$f''(t) = \int d(\partial_t \mu_t) \ \langle \nabla F'[\mu_t], \nabla \phi_t \rangle + \int d\mu_t \ \frac{d}{dt} \left\langle \nabla F'[\mu_t], \frac{d}{dt} \nabla \phi_t \right\rangle$$

$$= \int d\mu_t \ \left\langle \nabla \left[ \langle \nabla F'[\mu_t], \nabla \phi_t \rangle \right], \nabla \phi_t \right\rangle + \int d\mu_t \ \left( \left\langle \nabla F'[\mu_t], \frac{d}{dt} \nabla \phi_t \right\rangle + \left\langle \frac{d}{dt} \nabla F'[\mu_t], \nabla \phi_t \right\rangle \right)$$

$$= \int d\mu_t \ \nabla^2 F'[\mu_t](\nabla \phi_t, \nabla \phi_t)$$

$$+ \int d\mu_t \ \nabla^2 \phi_t \ (\nabla F'[\mu_t], \nabla \phi_t) + \int d\mu_t \ \langle \nabla F'[\mu_t], \nabla \partial_t \phi_t \rangle$$

$$+ \int d\mu_t \ \left\langle \frac{d}{dt} \nabla F'[\mu_t], \nabla \phi_t \right\rangle .$$

Now the first line can be bounded using the first condition of (P1): writing $s_t(\omega) = \frac{\nabla \phi_t(\omega)}{\|\nabla \phi_t(\omega)\|}$ for all $t$ and $\omega$,

$$\left| \int d\mu_t \ \nabla^2 F'[\mu_t](\nabla \phi_t, \nabla \phi_t) \right| = \left| \int d\mu_t \ \|\nabla \phi_t\|^2 \ \nabla^2 F'[\mu_t](s_t, s_t) \right| \le L \cdot \int d\mu_t \ \|\nabla \phi_t\|^2 = L.$$

Moreover, one can show by direct computation that the second line is zero, using that $\partial_t \phi_t = -\frac{1}{2} \|\nabla \phi_t\|^2$. For the third line, we have

$$\left| \int d\mu_t \ \left\langle \frac{d}{dt} \nabla F'[\mu_t], \nabla \phi_t \right\rangle \right| \le \int d\mu_t \ \|\nabla \phi_t\| \cdot \sup_{t \in [0,1]} \sup_{\omega \in \Omega} \left\| \frac{d}{dt} \nabla F'[\mu_t](\omega) \right\|$$

since $\left( \int d\mu_t \ \|\nabla \phi_t\| \right)^2 \le \int d\mu_t \ \|\nabla \phi_t\|^2 = 1$. Finally, let us show that the second condition of (P1) implies a bound on the last quantity: for all $\omega \in \Omega$, by applying the assumption to $\mu = \mu_t$ and $\mu' = \mu_s$,

$$\frac{\|\nabla F'[\mu_s](\omega) - \nabla F'[\mu_t](\omega)\|_\omega}{s - t} \le \frac{L \ W_2(\mu_s, \mu_t)}{s - t} = L$$

since $(\mu_t)_t$ is a constant-speed geodesic with $W_2(\mu_0, \mu_1) = 1$. So by letting $s \to t$ we obtain that $\left\| \frac{d}{dt} \nabla F'[\mu_t](\omega) \right\| \le L$ for all $t \in [0,1]$, $\omega \in \Omega$. Thus we have shown $|f''(t)| \le 2L$, and so $F$ is $2L$-smooth along Wasserstein geodesics. $\square$

## B.2 Classical sufficient conditions for LSI

For ease of reference we reproduce here a classical sufficient condition for a probability measure $\mu \in \mathcal{P}(\Omega)$ to satisfy LSI.

**Lemma B.2** (Holley-Stroock bounded perturbation argument [HS86])**.** *Let $\mu, \mu_0 \in \mathcal{P}(\Omega)$ such that $\mu$ is absolutely continuous w.r.t. $\mu_0$. Suppose that $\mu_0$ satisfies LSI with constant $\alpha$ and that $-M \le \log \frac{d\mu}{d\mu_0}(\omega) + c \le M$ for all $\omega \in \text{supp}(\mu_0)$, for some $c \in \mathbb{R}$ and $M \ge 0$. Then $\mu$ satisfies LSI with constant $\alpha e^{-M}$.*

## C Details for Sec. 3.1 (reduction by lifting)

### C.1 Proof of Prop. 3.1

Here we present a slightly stronger version of Prop. 3.1 that uses the $p$-homogeneous projection operator for arbitrary $p > 0$, in preparation for the next subsection, where we show that one can restrict attention to the case $p = 1$ as done in the main text.

Recall that we let $\Omega = \mathbb{R} \times \mathcal{W}$. For any $p > 0$, we denote by $\boldsymbol{h}^p : \mathcal{P}(\Omega) \to \mathcal{M}(\mathcal{W})$ the signed $p$-homogeneous projection operator [LMS18] defined by

$$\forall \varphi \in \mathcal{C}(\mathcal{W}, \mathbb{R}), \quad \int_{\mathcal{W}} \varphi(w)(\boldsymbol{h}^p \mu)(\mathrm{d}w) = \int_{\Omega} \operatorname{sign}(r) \, |r|^p \, \varphi(w) \mu(\mathrm{d}r, \mathrm{d}w).$$

More concretely, for atomic measures, $\boldsymbol{h}^p \left( \frac{1}{m} \sum_{j=1}^m \delta_{(r_j, w_j)} \right) = \frac{1}{m} \sum_{j=1}^m \operatorname{sign}(r_j) \, |r_j|^p \, \delta_{w_j}$.

**Lemma C.1.** *For $b \in [1, 2]$ and $p > 0$, let $\Psi_{b,p} : \mathcal{P}(\Omega) \to \mathbb{R} \cup \{+\infty\}$ defined by $\Psi_{b,p}(\mu) := \left( \int_{\Omega} |r|^{pb} \, \mathrm{d}\mu(r, w) \right)^{2/b}$ if $\mu \in \mathcal{P}_{pb}(\Omega)$, and $+\infty$ otherwise. Then*

$$\min_{\mu \ s.t. \ \boldsymbol{h}^p \mu = \nu} \Psi_{b,p}(\mu) = \|\nu\|_{TV}^2 .$$

*Moreover, if $b = 1$ then the set of minimizers is*

$$\{\mu \in \mathcal{P}(\mathcal{W}); \ \boldsymbol{h}^p \mu = \nu \ and \ \forall w, \operatorname{supp}(\mu(\cdot|w)) \subset \mathbb{R}_+ \ or \ \operatorname{supp}(\mu(\cdot|w)) \subset \mathbb{R}_- \},$$

*and if $b > 1$ there is a unique minimizer which is $\delta_{f(w)}(\mathrm{d}r) \frac{|\nu|(\mathrm{d}w)}{\|\nu\|_{TV}}$ where $f(w) = \|\nu\|_{TV}^{1/p} \frac{\mathrm{d}\nu}{\mathrm{d}|\nu|}(w)$.*

*Proof.* For any $\mu \in \mathcal{P}(\Omega)$ such that $\boldsymbol{h}^p = \nu$,

$$\|\boldsymbol{h}^p \mu\|_{TV} = \max_{\phi: \mathcal{W} \to [-1,1]} \int_{\Omega} \operatorname{sign}(r) \, |r|^p \, \phi(w) \mathrm{d}\mu(r, w) \le \int_{\Omega} |r|^p \, \mathrm{d}\mu(r, w)$$

$$\text{so } \|\nu\|_{TV}^2 = \|\boldsymbol{h}^p \mu\|_{TV}^2 \le \left( \left( \int_{\Omega} |r|^p \, \mathrm{d}\mu(r, w) \right)^b \right)^{2/b} \le \left( \int_{\Omega} |r|^{pb} \, \mathrm{d}\mu(r, w) \right)^{2/b} = \Psi_{b,p}(\mu),$$

where the first inequality follows from the triangle inequality, and the second inequality follows from Jensen's inequality since $t \mapsto t^b$ is convex on $\mathbb{R}_+$. Note that the first inequality above holds with equality if and only if there exists $\phi : \mathcal{W} \to [-1, 1]$ such that $\operatorname{sign}(r)\phi(w) \ge 0$ for all $(r, w) \in \operatorname{supp}(\mu)$, i.e., if the conditional distribution $\mu(\mathrm{d}r|w)$ is either supported on $\mathbb{R}_+$ or supported on $\mathbb{R}_-$ for each $w$. Conversely, the value $\|\nu\|_{TV}^2$ is attained by letting $\mu(\mathrm{d}r, \mathrm{d}w) = \delta_{f(w)}(\mathrm{d}r) \frac{|\nu|(\mathrm{d}w)}{\|\nu\|_{TV}}$ where $f(w) = \|\nu\|_{TV}^{1/p} \frac{\mathrm{d}\nu}{\mathrm{d}|\nu|}(w)$. This proves that $\min_{\mu: \boldsymbol{h}^p \mu = \nu} \Psi_{b,p}(\mu) = \|\nu\|_{TV}^2$.

For $b = 1$, $t \mapsto t^b = t$ is linear, so equality always holds in Jensen's inequality. So the set of minimizers is all of $\{\mu \in \mathcal{P}(\mathcal{W}); \ \boldsymbol{h}^p \mu = \nu \ \text{and} \ \forall w, \operatorname{supp}(\mu(\cdot|w)) \subset \mathbb{R}_+ \ \text{or} \ \operatorname{supp}(\mu(\cdot|w)) \subset \mathbb{R}_- \}$.

For $b > 1$, $t \mapsto t^b$ is strictly convex, the second inequality above holds with equality if and only if there exists a constant $c$ such that $|r|^p = c$ for all $(r, w) \in \operatorname{supp}(\mu)$. So for $\mu$ to be a minimizer, the conditional distribution $\mu(\mathrm{d}r|w)$ must be concentrated on $\{c^{1/p}, -c^{1/p}\}$ for each $w$. Moreover, for the first inequality above to hold, the conditional distribution at each $w$ must be either supported on $\mathbb{R}_+$ or suported on $\mathbb{R}_-$, so there exists a function $f : \mathcal{W} \to \{c^{1/p}, -c^{1/p}\}$ such that $\mu(\mathrm{d}r, \mathrm{d}w) = \delta_{f(w)}(\mathrm{d}r)\mu^w(\mathrm{d}w)$ where $\mu^w \in \mathcal{P}(\mathcal{W})$ denotes the marginal distribution. Since $\boldsymbol{h}^p \mu = \nu$, then for all fixed $w$, $\int_{\mathbb{R}_+} \operatorname{sign}(r) \, |r|^p \, \mu(\mathrm{d}r, \mathrm{d}w) = \operatorname{sign}(f(w))c\mu^w(\mathrm{d}w) = \nu(\mathrm{d}w)$. So $\operatorname{sign}(f(w)) = \operatorname{sign}(\frac{\mathrm{d}\nu}{\mathrm{d}\mu^w}(w)) = \frac{\mathrm{d}\nu}{\mathrm{d}|\nu|}(w)$ and $\mu^w(\mathrm{d}w) = \frac{1}{c} |\nu| \, (\mathrm{d}w)$ since $\mu^w$ is a probability measure so non-negative, and integrating on both sides over $\Omega$ shows that $c = \|\nu\|_{TV}$. Hence the only minimizer is $\mu(\mathrm{d}r, \mathrm{d}w) = \delta_{f(w)}(\mathrm{d}r) \frac{|\nu|(\mathrm{d}w)}{\|\nu\|_{TV}}$ where $f(w) = c^{1/p} \frac{\mathrm{d}\nu}{\mathrm{d}|\nu|}(w)$. $\qquad \square$

Prop. 3.1 follows directly as a special case of the following proposition with $p = 1$.

**Proposition C.2.** *Let any $p > 0$ and $b \in [1, 2]$ and let $\Psi_{b,p} : \mathcal{P}(\Omega) \to \mathbb{R} \cup \{+\infty\}$ as in the lemma above. Consider the optimization problem over probability measures, with $\lambda > 0$,*

$$\min_{\mu \in \mathcal{P}(\Omega)} F_{\lambda, b, p}(\mu) \quad where \quad F_{\lambda, b, p}(\mu) = G(\boldsymbol{h}^p \mu) + \frac{\lambda}{2} \Psi_{b,p}(\mu). \tag{C.1}$$

*Then $\min_{\mathcal{P}(\Omega)} F_{\lambda, b, p} = \min_{\mathcal{M}(\mathcal{W})} G_\lambda$.*

*Moreover, if $b > 1$ then $\arg \min F = \left\{ \delta_{\|\nu\|_{TV}^{1/p} \frac{\mathrm{d}\nu}{\mathrm{d}|\nu|}(w)}(\mathrm{d}r) \frac{\nu(\mathrm{d}w)}{\|\nu\|_{TV}}; \nu \in \arg \min G \right\}$, and otherwise $\arg \min F = \{\mu; \ \boldsymbol{h}^p \mu \in \arg \min G \ and \ \forall w, \operatorname{supp}(\mu) \subset \mathbb{R}_+ \ or \ \operatorname{supp}(\mu) \subset \mathbb{R}_+\}$. Furthermore, $F$ is convex.*

*Proof.* The fact that $\min_{\mathcal{P}(\Omega)} F_{\lambda,b,p} = \min_{\mathcal{M}(\mathcal{W})} G_\lambda$ can be seen directly as follows:

$$\min_{\mu \in \mathcal{P}(\Omega)} F(\mu) = \min_{\mu \in \mathcal{P}(\Omega)} G(\boldsymbol{h}^p \mu) + \frac{\lambda}{2} \Psi_{b,p}(\mu)$$

$$= \min_{\nu \in \mathcal{M}(\Omega)} \left[ \min_{\mu \in \mathcal{P}(\Omega):\boldsymbol{h}^p=\nu} G(\boldsymbol{h}^p \mu) + \frac{\lambda}{2} \Psi_{b,p}(\mu) \right]$$

$$= \min_{\nu \in \mathcal{M}(\Omega)} G(\nu) + \frac{\lambda}{2} \left[ \min_{\mu \mathcal{P}(\Omega):\boldsymbol{h}^p=\nu} \Psi_{b,p}(\mu) \right]$$

$$= \min_{\nu \in \mathcal{M}(\Omega)} G(\nu) + \frac{\lambda}{2} \|\nu\|_{TV}^2 \;=\; \min_{\nu \in \mathcal{M}(\Omega)} G_\lambda(\nu)$$

where we used the lemma above at the fourth equality. The characterization of $\arg\min F$ in terms of $\arg\min G$ follows from the characterization of the minimizers of the inner minimization $\left[ \min_{\mu \in \mathcal{P}(\Omega):\boldsymbol{h}^p=\nu} \Psi_b(\mu) \right]$ in the third line, which is given by the lemma above.

Furthermore, $F_{\lambda,b,p}$ is convex since $G$ and $\Psi_{b,p}$ are. $\qquad\square$

## C.2 Equivalence of using $(cp, cq_r, cq_w, \Gamma/c^2)$ for any $c > 0$ by reparametrizing

**Equivalence of Riemannian structures on $\Omega^*$ for $(cq_r, cq_w, \Gamma/c^2)$ for $c > 0$.** Recall that we consider equipping $\Omega^* = \mathbb{R}^* \times \mathcal{W}$ with a Riemannian metric of the form (3.2), reproduced here for ease of reference:

$$\left\langle \begin{pmatrix} \delta r_1 \\ \delta w_1 \end{pmatrix}, \begin{pmatrix} \delta r_2 \\ \delta w_2 \end{pmatrix} \right\rangle_{(r,w)} = \Gamma^{-1} |r|^{q_r} \frac{\delta r_1 \delta r_2}{r^2} + |r|^{q_w} \langle \delta w_1, \delta w_2 \rangle_w, \text{ i.e., } g_{(r,w)} = \begin{bmatrix} \Gamma^{-1} |r|^{q_r-2} & 0 \\ 0 & |r|^{q_w} g_w \end{bmatrix}.$$

The following proposition shows that, in fact, different choices of $q_r, q_w$ and $\Gamma$ lead to the same geometry, up to a reparametrization of the form $(a, w) = (r^\alpha, w)$ (for $r > 0$). Namely it is equivalent to use the metric with exponents $(q_r, q_w)$ or with $\left(\frac{q_r}{\alpha}, \frac{q_w}{\alpha}\right)$, up to adjusting $\Gamma$.

**Proposition C.3.** *For any $q_r, q_w$, denote by $g_{[q_r,q_w,\Gamma]}$ the metric $g_{(r,w)} = \begin{bmatrix} \Gamma^{-1} |r|^{q_r-2} & 0 \\ 0 & |r|^{q_w} g_w \end{bmatrix}$ on $\Omega^* = \mathbb{R}^* \times \mathcal{W}$. Then for any $q_r, q_w \in \mathbb{R}$ and $\Gamma, \alpha > 0$, the map $T_\alpha : \left(\Omega^*, g_{[q_r,q_w,\Gamma]}\right) \to \left(\Omega^*, g_{\left[\frac{q_r}{\alpha}, \frac{q_w}{\alpha}, \alpha^2\Gamma\right]}\right)$ defined by $T_\alpha(r, w) = (\mathrm{sign}(r) |r|^\alpha, w)$ is an isometry.*

*Proof.* Since $\Omega^*$ is a disjoint manifold: $\Omega^* = \mathbb{R}_+^* \times \mathcal{W} \cup \mathbb{R}_-^* \times \mathcal{W}$, and since $T_\alpha(\mathbb{R}_+^* \times \mathcal{W}) = \mathbb{R}_+^* \times \mathcal{W}$, it suffices to check that the restricted map $T_\alpha^+ : \left(\mathbb{R}_+^* \times \mathcal{W}, g_{[q_r,q_w,\Gamma]}\right) \to \left(\mathbb{R}_+^* \times \mathcal{W}, g_{\left[\frac{q_r}{\alpha}, \frac{q_w}{\alpha}, \alpha^2\Gamma\right]}\right)$ is an isometry (as well as the analogous statement for the restricted map $T_\alpha^-$, but it will follow analogously).

Indeed, denote by $\tilde{g}$ the metric on $\mathbb{R}_+^* \times \mathcal{W}$ induced by $T_\alpha^+$. It is given by, for $(a, w) = T_\alpha^+(r, w) = (r^\alpha, w)$, so $\frac{da}{a} = \alpha \frac{dr}{r}$,

$$\begin{pmatrix} \delta r_1 \\ \delta w_1 \end{pmatrix} \cdot g_{(r,w)} \begin{pmatrix} \delta r_2 \\ \delta w_2 \end{pmatrix} = \begin{pmatrix} \delta a_1 \\ \delta w_1 \end{pmatrix} \cdot \tilde{g}_{(a,w)} \begin{pmatrix} \delta a_2 \\ \delta w_2 \end{pmatrix} = \begin{pmatrix} \alpha a \frac{1}{r} \delta r_1 \\ \delta w_1 \end{pmatrix} \cdot \tilde{g}_{(a,w)} \begin{pmatrix} \alpha a \frac{1}{r} \delta r_2 \\ \delta w_2 \end{pmatrix}$$

so $\tilde{g}_{(a,w)} = \begin{bmatrix} \frac{r}{\alpha a} & 0 \\ 0 & 1 \end{bmatrix} g_{(r,w)} \begin{bmatrix} \frac{r}{\alpha a} & 0 \\ 0 & 1 \end{bmatrix}$

$$= \begin{bmatrix} \frac{r^2}{\alpha^2 a^2} \Gamma^{-1} r^{q_r-2} & 0 \\ 0 & r^{q_w} g_w \end{bmatrix} = \begin{bmatrix} \Gamma^{-1} \alpha^{-2} a^{q_r/\alpha-2} & 0 \\ 0 & a^{q_w/\alpha} g_w \end{bmatrix}.$$

So $\tilde{g}$ is precisely $g_{\left[\frac{q_r}{\alpha}, \frac{q_w}{\alpha}, \alpha^2\Gamma\right]}$ on $\mathbb{R}_+^* \times \mathcal{W}$, which proves the claim. $\qquad\square$

**Equivalence of the Wasserstein gradient flow of $F_{\lambda,b,p}$ for $(cp, cq_r, cq_w, \Gamma/c^2)$ for any $c > 0$.**

**Proposition C.4.** *Let $T : (\Omega_1, g_{[1]}) \to (\Omega_2, g_{[2]})$ an isometry between Riemannian manifolds. Let $F : \mathcal{P}(\Omega_1) \to \mathbb{R}$ (sufficiently regular) and $(\mu_t)_t$ a Wasserstein gradient flow for $F$, i.e., $\partial_t \mu_t = -\mathrm{div}(\mu_t \nabla F'[\mu_t])$ (where $\nabla$ denotes Riemannian gradient in $(\Omega_1, g_{[1]})$). Then, $(\tilde{\mu})_t := (T_\sharp \mu_t)_t$ is a Wasserstein gradient flow for $\widetilde{F} : \mathcal{P}(\Omega_2) \to \mathbb{R}$ defined by $\widetilde{F}(\tilde{\mu}) = F(T_\sharp^{-1} \tilde{\mu})$.*

*Proof.* First note that $g_{[2]}$ is given by, for all $y = T(x) \in \Omega_2$, so $dy = DT(x)dx$ where $D$ denotes the differential,

$$\delta y^\top \, g_{[2]y} \, \delta y' = \delta x^\top \, g_{[1]x} \, \delta x' = \delta y^\top \left((DT(x))^{-1}\right)^\top g_{[1]x} \, (DT(x))^{-1} \delta y'$$
$$\text{so} \quad g_{[1]x}^{-1} = (DT(x))^{-1}) \, g_{[2]T(x)}^{-1} \left((DT(x))^{-1}\right)^\top.$$

Also note that $\widetilde{F}'[\tilde{\mu}](y) = F'[T_\sharp^{-1}\tilde{\mu}](T^{-1}(y))$, as one can check directly by computing $\lim_{\varepsilon \to 0} \frac{1}{\varepsilon}\left[\widetilde{F}(\tilde{\mu} + \varepsilon\tilde{\nu}) - \widetilde{F}(\tilde{\mu})\right] = \lim_{\varepsilon \to 0} \frac{1}{\varepsilon}\left[F(T_\sharp^{-1}\tilde{\mu} + \varepsilon T_\sharp^{-1}\tilde{\nu}) - F(T_\sharp^{-1}\tilde{\mu})\right]$. In particular $D\widetilde{F}'[\tilde{\mu}](y) = DF'[T_\sharp^{-1}\tilde{\mu}](T^{-1}(y))(DT(T^{-1}(y)))^{-1}$. Then for any $\varphi : \Omega_2 \to \mathbb{R}$,

$$\frac{d}{dt}\int_{\Omega_2}\varphi \mathrm{d}\tilde{\mu}_t = \frac{d}{dt}\int_{\Omega_1}\varphi(T(x))\mathrm{d}\mu_t(x)$$
$$= \int_{\Omega_1} D\varphi(T(x))DT(x)\, g_{[1]}^{-1}\, DF'[\mu_t](x)\mathrm{d}\mu_t(x)$$
$$= \int_{\Omega_1} D\varphi(y)\, g_{[2]}^{-1}\, D\widetilde{F}'[\tilde{\mu}_t](y)\mathrm{d}\tilde{\mu}_t(y).$$

That is, $\partial_t\tilde{\mu}_t = -\mathrm{div}(\tilde{\mu}_t g_{[2]}^{-1} D\widetilde{F}'[\tilde{\mu}_t])$, i.e., $(\tilde{\mu}_t)_t$ is a Wasserstein gradient flow for $\widetilde{F}$. $\qquad\square$

**Proposition C.5.** *Consider the functionals $F_{\lambda,b,p}$ over $\mathcal{P}(\Omega)$ from [Prop. C.2]{.underline} and the Riemannian metrics $g_{[q_r,q_w,\Gamma]}$ over $\Omega^*$ from [Prop. C.3]{.underline}, where $\Omega = \mathbb{R} \times \mathcal{W}$ and $\Omega^* = \mathbb{R}^* \times \mathcal{W}$.*

*Fix $q_r, q_w \in \mathbb{R}$, $\Gamma, p, \lambda > 0$ and $b \in [1, 2]$. Let $(\mu_t)_t$ the Wasserstein gradient flow for $F_{\lambda,b,p}$ over $(\Omega^*, g_{[q_r,q_w,\Gamma]})$, starting from some $\mu_0 \in \mathcal{P}(\Omega^*)$.*

*Let $\alpha > 0$ and $T_\alpha : \Omega^* \to \Omega^*$ defined by $T_\alpha(r, w) = (\mathrm{sign}(r)\, |r|^\alpha, w)$. Then $(\tilde{\mu}_t)_t := ((T_\alpha)_\sharp\mu_t)_t$ coincides with the Wasserstein gradient flow for $F_{\tilde{\lambda},\tilde{b},\tilde{p}}$ over $(\Omega^*, g_{[\tilde{q}_r,\tilde{q}_w,\widetilde{\Gamma}]})$ starting from $\tilde{\mu}_0 = (T_\alpha)_\sharp\mu_0$, where*

$$\tilde{p} = \frac{p}{\alpha}, \qquad \tilde{q}_r = \frac{q_r}{\alpha}, \qquad \tilde{q}_w = \frac{q_w}{\alpha}, \qquad \widetilde{\Gamma} = \alpha^2\Gamma, \qquad \tilde{\lambda} = \lambda, \qquad \tilde{b} = b.$$

*Proof.* The proposition follows from an application of [Prop. C.4]{.underline} with $T = T_\alpha$, $\Omega_1 = (\Omega^*, g_{[q_r,q_w,\Gamma]})$, $\Omega_2 = (\Omega^*, g_{[q'_r,q'_w,\Gamma']})$ and $F = F_{\lambda,b,p}$. Indeed the fact that $T_\alpha$ is an isometry from $\Omega_1$ to $\Omega_2$ was shown in [Prop. C.3]{.underline}. It only remains to show that $F \circ T_\sharp^{-1} = F_{\tilde{\lambda},\tilde{b},\tilde{p}}$. And indeed for any $\tilde{\mu} \in \mathcal{P}(\Omega^*)$,

$$F_{\lambda,b,p}((T_\alpha)_\sharp^{-1}\tilde{\mu}) = F_{\lambda,b,p}((T_{\alpha^{-1}})_\sharp\tilde{\mu}) = G\left(\boldsymbol{h}^p(T_{\alpha^{-1}})_\sharp\tilde{\mu}\right) + \frac{\lambda}{2}\Psi_{b,p}\left((T_{\alpha^{-1}})_\sharp\tilde{\mu}\right),$$

and $\boldsymbol{h}^p(T_{\alpha^{-1}})_\sharp\tilde{\mu} = \boldsymbol{h}^{p/\alpha}\tilde{\mu}$, since for any $\varphi : \mathcal{W} \to \mathbb{R}$,

$$\int_{\mathcal{W}}\varphi \mathrm{d}\left[\boldsymbol{h}^p(T_{\alpha^{-1}})_\sharp\tilde{\mu}\right] = \int_{\mathbb{R}}\int_{\mathcal{W}}\varphi(w)\,\mathrm{sign}(r)\,|r|^p\,[(T_{\alpha^{-1}})_\sharp\tilde{\mu}]\,(\mathrm{d}r, \mathrm{d}w)$$
$$= \int_{\mathbb{R}}\int_{\mathcal{W}}\varphi(w)\,\mathrm{sign}(\tilde{r})\,|\tilde{r}|^{p/\alpha}\,\tilde{\mu}(\mathrm{d}\tilde{r}, \mathrm{d}w) = \int_{\mathcal{W}}\varphi \mathrm{d}\left[\boldsymbol{h}^{p/\alpha}\tilde{\mu}\right],$$

and

$$\Psi_{b,p}((T_{\alpha^{-1}})_\sharp\tilde{\mu}) = \left(|r|^{pb}\,\mathrm{d}\,[(T_{\alpha^{-1}})_\sharp\tilde{\mu}]\right)^{2/b} = \left(|\tilde{r}|^{pb/\alpha}\,\mathrm{d}\tilde{\mu}(r, w)\right)^{2/b}.$$

This confirms that $F \circ T_\sharp^{-1} = F_{\tilde{\lambda},\tilde{b},\tilde{p}}$ and concludes the proof. $\qquad\square$

Thus, it is equivalent to consider the lifting reduction with the hyperparameters $(p, q_r, q_w, \Gamma)$ or with $\left(cp, cq_r, cq_w, \Gamma/c^2\right)$ for any $c > 0$.

*Remark* C.1. The choice $p = q_r = q_w$ plays a special role, as Wasserstein gradient flows $(\mu_t)_t$ on $\mathcal{P}(\mathbb{R}_+^* \times \mathcal{W})$ for functionals of the form $\mu \mapsto G(\boldsymbol{h}^p\mu)$ then correspond to gradient flows $(\nu_t)_t$ on $\mathcal{M}_+(\mathcal{W})$ for $G$ in the Wasserstein-Fisher-Rao geometry [Chi22c, Prop. 2.1], via $\nu_t = \boldsymbol{h}^p\mu_t$. This correspondence is lost however for functionals of the form of $F_{\lambda,b,p}$ as in [Prop. C.2]{.underline} with $\lambda \neq 0$.

**Equivalence of MFLD of $F_{\lambda,b,p}$ for $(cp, cq_r, cq_w, \Gamma/c^2)$ for any $c > 0$.** Since MFLD for $F_{\lambda,b,p}$ is the Wasserstein gradient flow of $F_{\lambda,b,p} + \beta^{-1}H$, then by Prop. C.4, by proceeding similarly as in the proof of Prop. C.5, it suffices to check that $\tilde{\mu} \mapsto H((T_{\alpha^{-1}})_\sharp \tilde{\mu})$ is equal to $H$ itself, up to an additive constant. And indeed, since $T_{\alpha^{-1}}$ is invertible, by data processing inequality for differential entropy, we have $H((T_{\alpha^{-1}})_\sharp \tilde{\mu}) = H(\tilde{\mu})$ for all $\tilde{\mu} \in \mathcal{P}(\Omega^*)$.

## C.3  Proof of Prop. 3.2

**Lemma C.6.** *Let $F_{\lambda,b,p}$ defined in* (C.1) *and $\Omega = \mathcal{W} \times \mathbb{R}$. For any $\mu \in \mathcal{P}(\Omega)$,*

$$F'_{\lambda,b,p}[\mu](r, w) = \mathrm{sign}(r)\, |r|^p\, G'[\boldsymbol{h}^p \mu](w) + \lambda' |r|^{pb} \tag{C.2}$$

*where $\lambda' = \lambda \frac{1}{b} \Psi_{b,p}(\mu)^{1-\frac{b}{2}}$.*

*Proof.* For any $\mu' \in \mathcal{P}(\Omega)$,

$$\lim_{\varepsilon \to 0} \frac{1}{\varepsilon} \left[ (G \circ \boldsymbol{h}^p)(\mu + \varepsilon\mu') - (G \circ \boldsymbol{h}^p)(\mu) \right] = \lim_{\varepsilon \to 0} \frac{1}{\varepsilon} \left[ G(\boldsymbol{h}^p \mu + \varepsilon \boldsymbol{h}^p \mu') - G(\boldsymbol{h}^p \mu) \right]$$

$$= \int_{\mathcal{W}} G'[\boldsymbol{h}^p \mu](w) \mathrm{d}\left[\boldsymbol{h}^p \mu'\right](w) = \int_{\mathbb{R} \times \mathcal{W}} \mathrm{sign}(r)\, |r|^p\, G'[\boldsymbol{h}^p \mu](w) \mathrm{d}\mu'(r, w)$$

and so $(G \circ \boldsymbol{h}^p)' [\mu](r, w) = \mathrm{sign}(r)\, |r|^p\, G'[\boldsymbol{h}^p \mu](w)$. Moreover

$$\Psi_{b,p}(\mu) = \left( \int_\Omega |r|^{pb}\, \mathrm{d}\mu(r, w) \right)^{\frac{2}{b}}$$

$$\Psi'_{b,p}[\mu](r, w) = \frac{2}{b} \left( \int_\Omega |r'|^{pb}\, \mathrm{d}\mu(r', w') \right)^{\frac{2}{b}-1} |r|^{pb} = \frac{2}{b} \Psi_{b,p}(\mu)^{1-\frac{b}{2}} |r|^{pb}.$$

Summing the results of these two calculations gives the first variation of $F_{\lambda,b,p} = G \circ \boldsymbol{h}^p + \frac{\lambda}{2}\Psi_{b,p}$. $\square$

**Lemma C.7.** *Let $f : \mathbb{R}_+^* \times \mathcal{W} \to \mathbb{R}$ defined by $f(r, w) = r^p \tilde{\phi}(w) + \lambda' r^{pb}$, for some $p, \lambda' > 0$, $b \in [1, 2]$, and $\tilde{\phi} : \mathcal{W} \to \mathbb{R}$. Assume that $\nabla^2 \tilde{\phi}$ is not constant equal to 0.*

*Consider $\mathbb{R}_+^* \times \mathcal{W}$ equipped with the Riemannian metric* (3.2)*. If $f$ has Lipschitz-continuous Riemannian gradients, then necessarily $b = 1$ and $p = q_r = q_w$, or $b = 1$ and $p = q_r/2 = q_w/2$ and $\nabla^2 \tilde{\phi}(w) = \Gamma p^2 \left( \tilde{\phi}(w) + \lambda' \right) g_w$ for all $w$.*

The proof of Lem. C.7 is technical, so it is deferred to the next section.

*Proof of Prop. 3.2.* Let us prove the first item in the proposition. Suppose by contraposition that $F_{\lambda,b}$ does satisfy (P1). Let any $\nu \in \mathcal{M}(\mathcal{W})$ such that $\nabla^2 G'[\nu]$ is not constant equal to 0, and consider some $\mu \in \mathcal{P}(\Omega)$ to be chosen such that $\boldsymbol{h}\mu = \nu$. Then by the first condition of (P1), $f := F'_{\lambda,b}[\mu]\big|_{\mathbb{R}_+^* \times \mathcal{W}}$ the restriction of $F'_{\lambda,b}[\mu]$ to $\mathbb{R}_+^* \times \mathcal{W}$ must have Lipschitz-continuous Riemannian gradients. More explicitly, by (C.2), $f(r, w) = rG'[\nu](w) + \lambda'_\mu r^b$ where $\lambda'_\mu = \frac{\lambda}{b}\Psi_b(\mu)^{1-\frac{b}{2}}$. So by Lem. C.7, necessarily $b = 1$, and so $\lambda'_\mu = \lambda \Psi_1(\mu)^{1/2}$. If $\tilde{\phi} := G'[\nu]$ satisfies $\nabla^2 \tilde{\phi}(w) = \Gamma p^2 \left( \tilde{\phi}(w) + \lambda'_\mu \right) g_w$ for all $w$, pick any other $\mu'$ such that $\boldsymbol{h}\mu' = \nu$ and $\Psi_1(\mu') \neq \Psi_1(\mu)$ – the existence of such a $\mu'$ follows from the first step in the proof of Lem. C.1. Then by applying the above reasoning to $F'_{\lambda,b}[\mu']\big|_{\mathbb{R}_+^* \times \mathcal{W}}$ instead of $f$, since $\lambda'_{\mu'} \neq \lambda'_\mu$, we also have by Lem. C.7 that $p = q_r = q_w$. This shows that if $F_{\lambda,b}$ satisfies (P1) then $(q_r, q_w, b) = (1, 1, 1)$, which was the announced necessary condition.

We now turn to the second item of the proposition. Suppose that $q_r = q_w = b = 1$. For any $\mu \in \mathcal{P}_1(\Omega)$, denote

$$\lambda_{0\mu} = \sup_{w \in \mathcal{W}} \frac{|G'[\boldsymbol{h}\mu](w)|}{\Psi_1(\mu)^{1/2}}.$$

Let us show that if $\lambda < \lambda_{0\mu}$, then $F_{\lambda,1}$ does not satisfy local LSI at $\mu$. Suppose that $\lambda < \lambda_{0\mu}$, i.e., there exists $w_0 \in \mathcal{W}$ such that
$$\Psi_1(\mu)^{1/2}\lambda < |G'[\boldsymbol{h}\mu](w_0)|.$$
Let us distinguish cases between $G'[\boldsymbol{h}\mu](w_0) \geq 0$ or $G'[\boldsymbol{h}\mu](w_0) < 0$.

First suppose $G'[\boldsymbol{h}\mu](w_0) \geq 0$, so that $\Psi_1(\mu)^{1/2}\lambda < G'[\boldsymbol{h}\mu](w_0)$. By continuity of $G'[\boldsymbol{h}\mu]$, let $N \subset \mathcal{W}$ an open neighborhood of $w_0$ such that $\forall w \in N, \Psi_1(\mu)^{1/2}\lambda < G'[\boldsymbol{h}\mu](w)$. Then, since $F'_{\lambda,1}[\mu](r,w) = |r|\left(\text{sign}(r)G'[\boldsymbol{h}\mu](w) + \lambda\Psi_1(\mu)^{1/2}\right)$ by (C.2),

$$\forall r \in \mathbb{R}_-, \forall w \in N, \ F'_{\lambda,1}[\mu](r,w) = |r|\left(-G'[\boldsymbol{h}\mu](w) + \lambda\Psi_1(\mu)^{1/2}\right) \leq 0$$

$$\text{and so} \quad \int_{\mathbb{R}}\int_{\mathcal{W}} e^{-\beta F'_{\lambda,1}[\mu](r,w)}\mathrm{d}r\mathrm{d}w \geq \int_{\mathbb{R}_-}\int_{N} e^{-\beta F'_{\lambda,1}[\mu](r,w)}\mathrm{d}r\mathrm{d}w$$

$$\geq \int_{\mathbb{R}_-}\int_{N} 1\,\mathrm{d}r\mathrm{d}w \ = +\infty.$$

This contradicts the exponential integrability condition in the definition of local LSI, and so $F_{\lambda,1}$ does not satisfy local LSI at $\mu$.

Likewise, now suppose that $G'[\boldsymbol{h}\mu](w_0) < 0$, so that $\Psi_1(\mu)^{1/2}\lambda < -G'[\boldsymbol{h}\mu](w_0)$. By continuity of $G'[\boldsymbol{h}\mu]$, let $N \subset \mathcal{W}$ an open neighborhood of $w_0$ such that $\forall w \in N, \Psi_1(\mu)^{1/2}\lambda < -G'[\boldsymbol{h}\mu](w)$. Then

$$\forall r \in \mathbb{R}_+, \forall w \in N, \ F'_{\lambda,1}[\mu](r,w) = |r|\left(G'[\boldsymbol{h}\mu](w) + \lambda\Psi_1(\mu)^{1/2}\right) \leq 0$$

$$\text{and so} \quad \int_{\mathbb{R}}\int_{\mathcal{W}} e^{-\beta F'_{\lambda,1}[\mu](r,w)}\mathrm{d}r\mathrm{d}w \geq \int_{\mathbb{R}_+}\int_{N} e^{-\beta F'_{\lambda,1}[\mu](r,w)}\mathrm{d}r\mathrm{d}w$$

$$\geq \int_{\mathbb{R}_+}\int_{N} 1\,\mathrm{d}r\mathrm{d}w \ = +\infty.$$

As in the previous case, we conclude that $F_{\lambda,1}$ does not satisfy local LSI at $\mu$. $\qquad\square$

## C.4 Proof of Lem. C.7 via computing the Hessians under the lifted Riemannian geometry

We start by a general lemma. We use $D$ to denote differentials, and for a function $f : \mathbb{R}_+^* \times \mathcal{W} \to \mathbb{R}$, we will write $D_r f = \frac{\partial f(r,w)}{\partial r}$ and $D_w f = \frac{\partial f(r,w)}{\partial w}$.

**Lemma C.8.** *Let $(\mathcal{W}, g)$ a Riemannian manifold. Let $\Omega_+^* = \mathbb{R}_+^* \times \mathcal{W}$ and consider*

$$\overline{g}_{(r,w)} = \begin{bmatrix} \alpha(r)^{-1} & 0 \\ 0 & \beta(r)^{-1}g_w \end{bmatrix}$$

*for smooth positive functions $\alpha, \beta : \mathbb{R}_+^* \to \mathbb{R}_+^*$. This defines a smooth Riemannian metric $\overline{g}$ on $\Omega_+^*$.*

*Denote by $\overline{g}_{(r,w)}, \overline{\nabla}, \overline{\Gamma}, \overline{\nabla^2}$ the Riemannian metric, gradient, Christoffel symbols, resp. Hessian on $\Omega_+^*$, and by $g_w, \nabla, \Gamma, \nabla^2$ the corresponding objects on the original space $\mathcal{W}$.*

*Let $f : \Omega_+^* \to \mathbb{R}$ a smooth scalar field. Write for convenience $f_r(w) = f(r,w)$, so that for example $\nabla f_r(w) = g_w^{-1}D_w f(r,w)$, and note that $D_r\nabla f_r(w) = \nabla D_r f_r(w)$. Fix a local coordinate chart on $\mathcal{W}$. This induces a local coordinate chart on $\Omega_+^*$ by adding the index $0$ for the variable $r$. Then the Riemannian Hessian $f$ at $(r,w)$ is given in coordinates by*

$$\overline{\nabla^2} f^{00} = \alpha(r)^2 D_{rr}^2 f + \frac{1}{2}\alpha(r)\alpha'(r)D_r f$$

$$\overline{\nabla^2} f^{i0} = \overline{\nabla^2} f^{0i} = \alpha(r)\beta(r)\nabla D_r f_r(w)^i + \frac{1}{2}\alpha(r)\beta'(r)\nabla f_r(w)^i$$

$$\overline{\nabla^2} f^{ij} = \beta(r)^2\,\nabla^2 f_r(w)^{ij} - \frac{1}{2}\alpha(r)\beta'(r)\cdot D_r f\cdot(g_w^{-1})^{ij}.$$

*Proof.* We will use uppercase letters for indes ranging over $[0,d]$ and lowercase for $[1,d]$, with the index $0$ corresponding to the variable $r$; for example $\overline{\nabla} f(r,w)^0 = \alpha(r)D_r f(r,w)$. We will use

Einstein summation notation freely. With slight abuse of notation we denote $(g^{ij})_{ij} = g^{-1}$ for the inverse matrix of the metric $(g_{ij})_{ij} = g$, and likewise for $\overline{g}^{IJ}, \overline{g}_{IJ}$, so that for example $\overline{g}^{00} = \alpha(r)$.

We start by using that [Lee18, Example 4.22, Eq. (5.10)]

$$\overline{\nabla^2} f(r, w)^{IJ} = \overline{g}^{IK}\overline{g}^{JL}\left[\frac{\partial^2 f}{\partial\omega^K \partial\omega^L} - \overline{\Gamma}^M_{KL}\frac{\partial f}{\partial^M\omega}\right]$$

$$\text{and} \quad \overline{\Gamma}^M_{IJ} = \frac{1}{2}\overline{g}^{MK}\left[\frac{\partial\overline{g}_{KI}}{\partial\omega^J} + \frac{\partial\overline{g}_{KJ}}{\partial\omega^I} - \frac{\partial\overline{g}_{IJ}}{\partial\omega^K}\right]$$

where $\omega = (r, w)$, and that the analogous formulas hold for $f_r : \mathcal{W} \to \mathbb{R}$ for all $r$ and for $\Gamma^m_{ij}$ the Christoffel symbols of $\mathcal{W}$.

By direct computations, we find that for all $i, j, m \in [1, d]$,

$$\overline{\Gamma}^0_{00} = -\frac{1}{2}\frac{\alpha'(r)}{\alpha(r)} \qquad \overline{\Gamma}^0_{i0} = \overline{\Gamma}^0_{0i} = 0 \qquad \overline{\Gamma}^0_{ij} = \frac{1}{2}\alpha(r)\frac{\beta'(r)}{\beta(r)^2}g_{ij}$$

$$\overline{\Gamma}^m_{00} = 0 \qquad \overline{\Gamma}^m_{i0} = \overline{\Gamma}^m_{0i} = -\frac{1}{2}\frac{\beta'(r)}{\beta(r)}\delta^m_i \qquad \overline{\Gamma}^m_{ij} = \Gamma^m_{ij}.$$

So by direct computations, we find that

$$\overline{\nabla^2} f^{00} = \alpha(r)^2 D^2_{rr}f + \frac{1}{2}\alpha(r)\alpha'(r)D_r f$$

$$\overline{\nabla^2} f^{i0} = \overline{\nabla^2} f^{0i} = \alpha(r)\beta(r)\nabla D_r f_r(w)^i + \frac{1}{2}\alpha(r)\beta'(r)\nabla f_r(w)^i$$

$$\overline{\nabla^2} f^{ij} = \beta(r)^2 \nabla^2 f_r(w)^{ij} - \frac{1}{2}\alpha(r)\beta'(r) \cdot D_r f \cdot g^{ij},$$

as announced. $\qquad\square$

**Corollary C.9.** *Let $f : \Omega^*_+ = \mathbb{R}^*_+ \times \mathcal{W} \to \mathbb{R}$ defined by $f(r, w) = r^p\tilde{\phi}(w) + \lambda'r^{pb}$, for some $p > 0$, $b \in [1, 2]$, $\lambda' \geq 0$ and $\tilde{\phi} : \mathcal{W} \to \mathbb{R}$.*

*Consider $\Omega^*_+$ equipped with the Riemannian metric (3.2). Then the Riemannian Hessian of $f$ is given in coordinates by*

$$\overline{\nabla^2} f^{00} = \Gamma^2 p(p - q_r/2)r^{2-2q_r+p}\tilde{\phi}(w) + \Gamma^2 pb\lambda'(pb - q_r/2)r^{2-2q_r+pb}$$

$$\overline{\nabla^2} f^{i0} = \overline{\nabla^2} f^{0i} = \Gamma(p - q_w/2)r^{1-q_r-q_w+p}\nabla\tilde{\phi}(w)^i$$

$$\overline{\nabla^2} f^{ij} = r^{p-2q_w}\nabla^2\tilde{\phi}(w)^{ij} + \frac{1}{2}\Gamma q_w r^{-q_r-q_w} \cdot \left(pr^p\tilde{\phi}(w) + pb\lambda'r^{pb}\right)(g_w^{-1})^{ij}.$$

*Proof.* Continuing with the same notations as in the proof of the lemma above, we have

$$D_r f = pr^{p-1}\tilde{\phi}(w) + pb\lambda'r^{pb-1} \qquad D^2_{rr}f = p(p-1)r^{p-2}\tilde{\phi}(w) + pb(pb-1)\lambda'r^{pb-2}$$

$$\nabla f_r(w)^i = r^p\nabla\tilde{\phi}(w)^i \qquad \nabla^2 f_r(w)^{ij} = r^p \nabla^2 \tilde{\phi}(w)^{ij}$$

$$\nabla D_r f_r(w)^i = pr^{p-1}\nabla\tilde{\phi}(w)^i$$

and so

$$\overline{\nabla^2} f^{00} = \alpha(r)^2\left(p(p-1)r^{p-2}\tilde{\phi}(w) + pb(pb-1)\lambda'r^{pb-2}\right) + \frac{1}{2}\alpha(r)\alpha'(r)\left(pr^{p-1}\tilde{\phi}(w) + pb\lambda'r^{pb-1}\right)$$

$$= \alpha(r)p\left(\alpha(r)(p-1) + \frac{1}{2}r\alpha'(r)\right)r^{p-2}\tilde{\phi}(w) + \alpha(r)pb\lambda'\left(\alpha(r)(pb-1) + \frac{1}{2}r\alpha'(r)\right)r^{pb-2}$$

$$\overline{\nabla^2} f^{i0} = \overline{\nabla^2} f^{0i} = \alpha(r)\beta(r) \cdot pr^{p-1}\nabla\tilde{\phi}(w)^i + \frac{1}{2}\alpha(r)\beta'(r) \cdot r^p\nabla\tilde{\phi}(w)^i$$

$$= \alpha(r)\left(\beta(r)p + \frac{1}{2}r\beta'(r)\right)r^{p-1}\nabla\tilde{\phi}(w)^i$$

$$\overline{\nabla^2} f^{ij} = \beta(r)^2 \cdot r^p \nabla^2 \tilde{\phi}(w)^{ij} - \frac{1}{2}\alpha(r)\beta'(r) \cdot \left(pr^{p-1}\tilde{\phi}(w) + pb\lambda'r^{pb-1}\right) \cdot g^{ij}.$$

By substituting $\alpha(r)^{-1} = \Gamma^{-1}r^{q_r-2}$ and $\beta(r)^{-1} = r^{q_w}$, i.e. $\alpha(r) = \Gamma r^{2-q_r}$ and $\beta(r) = r^{-q_w}$, we obtain the announced formulas. $\qquad\square$

*Proof of Lem. C.7.* Continuing with the same notations as in the proofs of the lemma and of the corollary above, note that $f : \Omega_+^* = \mathbb{R}_+^* \times \mathcal{W} \to \mathbb{R}$ having Lipschitz-continuous gradients in the Riemannian sense is equivalent to [Bou23, Coroll. 10.47]

$$\sup_{\substack{\omega \in \Omega_+^* \\ }} \sup_{\substack{s \in T_\omega \Omega_+^* \\ \|s\|_\omega = 1}} \left\| \overline{\nabla^2} f(\omega)^{IJ} \overline{g}_{JK} s^K \right\|_\omega < \infty.$$

Rewriting everything in coordinates, this means that the matrix $\widetilde{H}(\omega) = \left( \sqrt{\overline{g}}_{IK} \, \overline{\nabla^2} f(\omega)^{IJ} \sqrt{\overline{g}}_{JL} \right)_{KL} \in \mathbb{R}^{(d+1) \times (d+1)}$ must be bounded, uniformly in $\omega \in \Omega_+^*$, where $(\sqrt{\overline{g}}_{IJ})_{IJ} = \sqrt{\overline{g}}$ denotes the square root of the positive-definite matrix $\overline{g}$ (pointwise for each $\omega$). Concretely, for all $i, j \in [1, d]$,

$$\sqrt{\overline{g}}_{00} = \alpha(r)^{-1/2} = \Gamma^{-1/2} r^{q_r/2 - 1}, \qquad \sqrt{\overline{g}}_{i0} = 0, \qquad \sqrt{\overline{g}}_{ij} = \beta(r)^{-1/2} \sqrt{g}_{ij} = r^{q_w/2} \sqrt{g}_{ij}$$

and

$$\begin{aligned}
\widetilde{H}(\omega)_{00} &= \overline{g}_{00} \overline{\nabla^2} f^{00} \\
&= \Gamma p (p - q_r/2) r^{-q_r + p} \tilde{\phi}(w) + \Gamma p b \lambda' (pb - q_r/2) r^{-q_r + pb} \\
\widetilde{H}(\omega)_{j0} &= \sqrt{\overline{g}}_{00} \sqrt{\overline{g}}_{ji} \overline{\nabla^2} f^{i0} \\
&= \Gamma^{1/2} (p - q_w/2) r^{-q_r/2 - q_w/2 + p} \cdot \sqrt{g}_{ji} \nabla \tilde{\phi}(w)^i \\
\widetilde{H}(\omega)_{kl} &= \sqrt{\overline{g}}_{ki} \sqrt{\overline{g}}_{lj} \overline{\nabla^2} f^{ij} \\
&= r^{p - q_w} \cdot \sqrt{g}_{ki} \sqrt{g}_{lj} \nabla^2 \tilde{\phi}(w)^{ij} + \Gamma \frac{1}{2} q_w r^{-q_r} \cdot \left( pr^p \tilde{\phi}(w) + pb\lambda' r^{pb} \right) \delta_{kl}.
\end{aligned}$$

(Note that here the indes do not respect the covariant/contravariant convention, i.e., "$\sqrt{\overline{g}}_{IK}$" and "$\widetilde{H}(\omega)_{KL}$" do not stand for covariant tensors: we really manipulate everything in coordinates explicitly.)

Now, note that the desired condition means that $\widetilde{H}(\omega)_{KL}$ should remain bounded both for $r \to +\infty$ and $r \to 0$. That is, the exponents of $r$ in the non-zero terms must all be 0. Thus, since we assume that $\lambda' \neq 0$, and that $\nabla^2 \tilde{\phi}$ is not constant equal to 0 and so in particular $\tilde{\phi}$ and $\nabla \tilde{\phi}$ are not constant,

- Uniform boundedness of the second term in $\widetilde{H}(\omega)_{kl}$ implies that $b = 1$. Indeed $\lambda' \neq 0$, and the first term (in $\nabla^2 \tilde{\phi}$) cannot cancel out both the term in $\tilde{\phi}(w) r^{p-q_r}$ and the term in $\lambda' r^{pb-q_r}$ if they scale differently with $r$. This also implies that either $p = q_w = q_r$ or that $q_w = q_r$ and $\nabla^2 \tilde{\phi}(w)^{ij} = \frac{1}{2} \Gamma q_w p \left( \tilde{\phi}(w) + \lambda' \right) g^{ij}$ for all $w$.

- Uniform boundedness of $\widetilde{H}(\omega)_{00}$ implies that $p = q_r$ or $p = q_r/2$.

- Uniform boundedness of $\widetilde{H}(\omega){j0}$ implies that $p = \frac{q_r + q_w}{2}$ or $p = q_w/2$. We saw in the first point that $q_r = q_w$, so equivalently $p = q_r = q_w$ or $p = q_r/2 = q_w/2$.

Thus we get that $f$ can have Lipschitz-continuous Riemannian gradients only if $b = 1$ and $p = q_r = q_w$, or if $b = 1$ and $p = q_r/2 = q_w/2$ and $\nabla^2 \tilde{\phi}(w) = \Gamma p^2 \left( \tilde{\phi}(w) + \lambda' \right) g_w$ for all $w$. $\qquad \square$

## D  Details for Sec. 3.2 (reduction by bilevel optimization)

### D.1  Proof of Prop. 3.3

In preparation for the proof of Prop. 3.3, let us first provide a formal proof of the variational representation of the squared-TV norm mentioned at the beginning of Sec. 3.2, with a characterization of the set of minimizers. See [Chi17, App. 1] for the rigorous justification of these arguments in the more general context of minimization of convex and positively 1-homogeneous integral functionals over the space of signed measures.

**Lemma D.1** ("$\eta$-trick" for the squared TV-norm). *We have*

$$\|\nu\|_{TV}^2 = \left(\int_{\mathcal{W}} |\nu(dw)|\right)^2 = \inf_{\eta \in \mathcal{P}(\mathcal{W})} \int_{\mathcal{W}} \frac{|\nu(dw)|^2}{\eta(dw)} = \inf_{\substack{\eta \in \mathcal{P}(\mathcal{W}),\, f:\mathcal{W}\to\mathbb{R} \\ \text{s.t. } f\eta=\nu}} \int_{\mathcal{W}} |f|^2 \, d\eta.$$

*Moreover the infimum in the third expression is attained at (and only at) $\eta(dw) = \frac{|\nu(dw)|}{\|\nu\|_{TV}}$, and the infimum in the fourth expression is attained at (and only at) the same $\eta$ and $f = \frac{\nu(dw)}{|\nu(dw)|}\|\nu\|_{TV}$.*

*Proof.* The infimum in the third expression is the value of a convex constrained minimization problem, whose Lagrangian is $\mathcal{L}(\eta;\lambda) = \int \frac{|\nu|^2}{\eta} + \lambda\left(\int d\eta - 1\right)$. The dual optimality condition implies $\forall w \in \mathrm{supp}(\eta), \lambda = \frac{d\nu}{d\eta}(w)^2$, so the infinimum is attained at $\eta(dw) = \frac{|\nu(dw)|}{\|\nu\|_{TV}}$, with optimal value $\|\nu\|_{TV}^2$.

The optimality condition for the infimum in the fourth expression follows directly from the one for the third expression and from the constraint $f\eta = \nu$. $\qquad\square$

*Proof of Prop. 3.3.* By the lemma above,

$$\inf_{\eta \in \mathcal{P}(\mathcal{W})} J_\lambda(\eta) = \inf_{\eta \in \mathcal{P}(\mathcal{W}),\, f:\mathcal{W}\to\mathbb{R}} G(f\eta) + \frac{\lambda}{2}\int_{\mathcal{W}} |f|^2 \, d\eta$$

$$= \inf_{\nu \in \mathcal{M}(\mathcal{W})} \inf_{\substack{\eta \in \mathcal{P}(\mathcal{W}),\, f:\mathcal{W}\to\mathbb{R} \\ \text{s.t. } f\eta=\nu}} G(f\eta) + \frac{\lambda}{2}\int_{\mathcal{W}} |f|^2 \, d\eta$$

$$= \inf_{\nu \in \mathcal{M}(\mathcal{W})} G(\nu) + \frac{\lambda}{2}\left[\inf_{\substack{\eta \in \mathcal{P}(\mathcal{W}),\, f:\mathcal{W}\to\mathbb{R} \\ \text{s.t. } f\eta=\nu}} \int_{\mathcal{W}} |f|^2 \, d\eta\right]$$

$$= \inf_{\nu \in \mathcal{M}(\mathcal{W})} G(\nu) + \frac{\lambda}{2}\|\nu\|_{TV}^2 = \inf_{\nu \in \mathcal{M}(\mathcal{W})} G_\lambda(\nu).$$

Hence the equality of the optimal values. The claimed characterization of $\arg\min J_\lambda$ in terms of $\arg\min G_\lambda$ follows from the characterization of the minimizers of the inner minimization $\left[\inf_{\substack{\eta \in \mathcal{P}(\mathcal{W}),\, f:\mathcal{W}\to\mathbb{R} \\ \text{s.t. } f\eta=\nu}} \frac{\lambda}{2}\int_{\mathcal{W}} |f|^2 \, d\eta\right]$ in the third line, which is given by the lemma above.

Furthermore, $J_\lambda$ is convex as the partial minimization of $(\eta, \nu) \mapsto G(\nu) + \frac{\lambda}{2}\int \frac{|\nu|^2}{\eta}$, which is jointly convex. $\qquad\square$

## D.2 Proof of the explicit form of the two-timescale SDE (3.4)

For ease of reference, we recall here the two-timescale SDE (3.4):

$$\forall i \le N, \begin{cases} dr_t^i = -\Gamma \, \nabla_{r^i} F_{\lambda,2}'\left[\frac{1}{N}\sum_{j=1}^N \delta_{(r_t^j, w_t^j)}\right](r_t^i, w_t^i)dt \\ dw_t^i = -\nabla_{w^i} F_{\lambda,2}'\left[\frac{1}{N}\sum_{j=1}^N \delta_{(r_t^j, w_t^j)}\right](r_t^i, w_t^i)dt + \sqrt{2\beta^{-1}}dB_t^i. \end{cases}$$

By (C.2) with $b = 2$ and $p = 1$,

$$F_{\lambda,2}'[\mu](r, w) = rG'[h\mu](w) + \frac{\lambda}{2}|r|^2$$

$$\text{so} \quad \nabla_r F_{\lambda,2}'[\mu](r, w) = G'[h\mu](w) + \lambda r$$

$$\text{and} \quad \nabla_w F_{\lambda,2}'[\mu](r, w) = r\nabla G'[h\mu](w).$$

Finally, by definition $h\left[\frac{1}{N}\sum_{j=1}^N \delta_{(r^j, w^j)}\right] = \frac{1}{N}\sum_{j=1}^N r^j \delta_{w^j}$. Hence the second part of (3.4).

### D.3 Proof of Prop. 3.4 ($J_\lambda$ satisfies P0, P1 and P2)

**Simplifying the expression of the bilevel objective.** The following expressions will be useful throughout our analyses of the bilevel problem (3.3).

**Proposition D.2.** *We have that $J_\lambda(\eta) = G(f_\eta \eta) + \frac{\lambda}{2} \int |f_\eta|^2 \, \mathrm{d}\eta$ where $f_\eta$ is the unique solution of the fixed-point equation*

$$\forall w \in \mathcal{W}, \ f_\eta(w) = -\frac{1}{\lambda} G'[f_\eta \eta](w). \tag{D.1}$$

*Furthermore,*

$$J_\lambda'[\eta](w) = -\frac{\lambda}{2} |f_\eta|^2 (w). \tag{D.2}$$

*Proof.* Consider the optimization problem defining $J_\lambda(\eta)$, for a fixed $\eta$,

$$\min_{f \in L_\eta^2(\mathcal{W})} G(f\eta) + \frac{\lambda}{2} \int_{\mathcal{W}} |f|^2 \, \mathrm{d}\eta.$$

This problem is convex since $G$ is, and strongly convex in $L_\eta^2(\mathcal{W})$ thanks to the term in $\lambda$. So there exists a unique solution which we denote by $\tilde{f}_\eta \in L_\eta^2(\mathcal{W})$, and it is characterized by the first-order optimality condition:

$$G'[\tilde{f}_\eta \, \eta] \, \eta + \lambda \tilde{f}_\eta \, \eta = 0 \ \text{ in } \mathcal{M}(\mathcal{W}).$$

Now let $f_\eta = -\frac{1}{\lambda} G'[\tilde{f}_\eta \eta]$, which is defined over all of $\mathcal{W}$. Then $f_\eta$ satisfies the fixed-point equation (D.1) by construction. Conversely, for any solution $g_\eta$ of (D.1), its restriction to $\mathrm{supp}(\eta)$ viewed as an element $\tilde{g}_\eta$ of $L_\eta^2(\mathcal{W})$ must in particular satisfy $G'[\tilde{g}_\eta \eta]\eta + \lambda \tilde{g}_\eta \eta = 0$ in $\mathcal{M}(\mathcal{W})$, and so $\tilde{g}_\eta = \tilde{f}_\eta$, and so $g_\eta = -\frac{1}{\lambda} G'[\tilde{g}_\eta \eta] = -\frac{1}{\lambda} G'[\tilde{f}_\eta \eta] = f_\eta$.

Furthermore, by differentiability of $G$ then $\eta \mapsto \tilde{f}_\eta$ is continuous (in the total variation sense). So in turn, $\eta \mapsto f_\eta(w)$ the unique solution of (D.1) is continuous for each $w$ (in the total variation sense). So by the envelope theorem, since for any fixed $f$ the first variation of $\eta \mapsto G(f\eta) + \frac{\lambda}{2} \int |f|^2 \, \mathrm{d}\eta$ is $w \mapsto f(w)G'[f\eta](w) + \frac{\lambda}{2} |f(w)|^2$,

$$J_\lambda'[\eta](w) = f_\eta(w)G'[f_\eta \eta](w) + \frac{\lambda}{2} |f_\eta(w)|^2$$

$$= -\frac{\lambda}{2} |f_\eta(w)|^2 = -\frac{1}{2\lambda} |G'[f_\eta \eta]|^2 (w),$$

which is precisely Eq. (D.2). $\qquad\square$

We remark that the above manipulations rely crucially on the fact that the optimization problem (1.1) is over signed measures and not just non-negative measures – as otherwise we would additionally need to constrain $f \geq 0$ –, and on the regularization term being $\|\nu\|_{TV}^2$ instead of $\|\nu\|_{TV}$.

**Preliminary estimates.**

**Lemma D.3.** *Under Assumption 1, for any $\nu \in \mathcal{M}(\mathcal{W})$, we have*

$$\sup_{w \in \mathcal{W}} |G'[\nu](w)|^2 \leq 2L_0 G(\nu).$$

*Proof.* We follow the proof technique of [GGGM21, Appendix D]. Let $w_0 \in \mathcal{W}$ and $\nu' = \nu - \frac{1}{L_0} G'[\nu](w_0)\delta_{w_0}$. By mean-value theorem there exists $\theta \in (0,1)$ such that $G(\nu') - G(\nu) = \int G'[\nu + \theta(\nu' - \nu)]\mathrm{d}(\nu' - \nu)$, and so

$$\inf G \leq G(\nu') \leq G(\nu) + \int G'[\nu]\mathrm{d}(\nu' - \nu) + \frac{L_0}{2} \|\nu' - \nu\|_{TV}^2$$

$$= G(\nu) - \frac{1}{L_0} G'[\nu](w_0)^2 + \frac{1}{2L_0} G'[\nu](w_0)^2 = G(\nu) - \frac{1}{2L_0} G'[\nu](w_0)^2.$$

Hence, since $G$ is non-negative by Assumption 1,

$$\forall w \in \mathcal{W}, \ \frac{1}{2L_0} G'[\nu](w)^2 \leq G(\nu) - \inf G \leq G(\nu) \qquad\square$$

**Lemma D.4.** *Under Assumption 1, let $\eta \in \mathcal{P}(\mathcal{W})$ and let $f_\eta$ as in (D.1). Then*

$$\sup_{\mathcal{W}} |f_\eta| \leq \frac{1}{\lambda} \sqrt{2L_0 J_\lambda(\eta)}$$

*and for each $i \in \{1, 2\}$,*

$$\sup_{w \in \mathcal{W}} \left\| \nabla^i f_\eta \right\|_w \leq \frac{L_i}{\lambda^2} \sqrt{2L_0 J_\lambda(\eta)} + \frac{B_i}{\lambda}.$$

*Moreover, $J_\lambda(\eta) \leq G(0)$ for all $\eta \in \mathcal{P}(\mathcal{W})$.*

*Proof.* For the first inequality, by definition $G'[f_\eta \eta] = -\lambda f_\eta$ for all $w \in \mathcal{W}$, so

$$\lambda^2 |f_\eta(w)|^2 = |G'[f_\eta \eta](w)|^2 \leq 2L_0 G(f_\eta \eta) \leq 2L_0 \left( G(f_\eta \eta) + \frac{\lambda}{2} \int |f_\eta|^2 \, \mathrm{d}\eta \right) = 2L_0 J_\lambda(\eta)$$

where the first inequality follows from Lem. D.3.

For the second part, by Assumption 1, $\forall \nu \in \mathcal{M}(\mathcal{W})$, $\sup_w \left\| \nabla^i G'[\nu] \right\|_w \leq L_i \|\nu\|_{TV} + B_i$, so

$$\lambda \left\| \nabla^i f_\eta \right\|_w = \left\| \nabla^i G'[f_\eta \eta] \right\|_w \leq B_i + L_i \|f_\eta \eta\|_{TV} = B_i + L_i \int |f_\eta| \, \mathrm{d}\eta$$

$$\leq B_i + L_i \sup_{\mathcal{W}} |f_\eta| \leq B_i + L_i \frac{1}{\lambda} \sqrt{2L_0 J_\lambda(\eta)}$$

by the first part of the lemma.

Finally, the uniform bound on $J_\lambda(\eta)$ follows by taking $f = 0$ in the infimum defining $J_\lambda$: $J_\lambda(\eta) = \inf_{f \in L^2_\eta} G(f\eta) + \frac{\lambda}{2} \int |f|^2 \, \mathrm{d}\eta \leq G(0)$. $\square$

**Lemma D.5.** *Under Assumption 1, $J_\lambda : \mathcal{P}(\mathcal{W}) \to \mathbb{R}$ is weakly continuous and*

$$\forall \eta, \eta' \in \mathcal{P}(\mathcal{W}), \ |J_\lambda(\eta) - J_\lambda(\eta')| \leq BW_2(\eta, \eta')$$

*where $B = \sqrt{2L_0 G(0)} \cdot \left( \frac{L_1}{\lambda^2} \sqrt{2L_0 G(0)} + \frac{B_1}{\lambda} \right)$.*

*Proof.* For any $\eta \in \mathcal{P}(\mathcal{W})$, letting $f_\eta$ as in (D.1), we have $J'_\lambda[\eta](w) = -\frac{\lambda}{2} |f_\eta|^2 (w)$ so

$$\nabla J'_\lambda[\eta](w) = -\lambda f_\eta(w) \nabla f_\eta(w)$$

$$\|\nabla J'_\lambda[\eta](w)\|_w \leq \lambda \sup_{\mathcal{W}} |f_\eta| \cdot \sup_{\mathcal{W}} \|\nabla f_\eta\|$$

$$\leq \lambda \cdot \frac{1}{\lambda} \sqrt{2L_0 G(0)} \cdot \left( \frac{L_1}{\lambda^2} \sqrt{2L_0 G(0)} + \frac{B_1}{\lambda} \right) =: B < \infty$$

by Lem. D.4, uniformly in $\eta \in \mathcal{P}(\mathcal{W})$ and $w \in \mathcal{W}$. So by Lem. D.8 below, we have $|J_\lambda(\eta) - J_\lambda(\eta')| \leq BW_2(\eta, \eta')$ for all $\eta, \eta' \in \mathcal{P}(\mathcal{W})$. Moreover $W_2$ metrizes weak convergence, so $J_\lambda$ is weakly continuous. $\square$

**Lemma D.6.** *Under Assumption 1, let $w' \in \mathcal{W}$ and $\eta \in \mathcal{P}(\mathcal{W})$. Let $h : \mathcal{W} \to \mathbb{R}$ and suppose that*

$$\forall w \in \mathcal{W}, \ \lambda h(w) + \int G''[f_\eta \eta](w, w'') d\eta(w'') h(w'') = -G''[f_\eta \eta](w, w') f_\eta(w').$$

*Then $\sup_{w \in \mathcal{W}} |h(w)| \leq \left( 1 + \frac{L_0}{\lambda} \right) \frac{L_0}{\lambda} \sqrt{2L_0 G(0)}$.*

*Alternatively, suppose that there exists $s \in T_{w'}\mathcal{W}$ with $\|s\|_{w'} = 1$ such that*

$$\forall w \in \mathcal{W}, \ \lambda h(w) + \int G''[f_\eta \eta](w, w'') d\eta(w'') h(w'') = -\langle s', \nabla_{w'} [G''[f_\eta \eta](w, w') f_\eta(w')] \rangle_{w'}.$$

*Then $\sup_{w \in \mathcal{W}} |h(w)| \leq \left( 1 + \frac{L_0}{\lambda} \right) \cdot \left( \left( 1 + \frac{L_0}{\lambda} \right) \frac{L_1}{\lambda} \sqrt{2L_0 G(0)} + \frac{L_0 B_1}{\lambda} \right)$.*

*Proof.* Let $\mathcal{G} : L^2_\eta(\mathcal{W}) \to L^2_\eta(\mathcal{W})$ the operator

$$(\mathcal{G}\tilde{h})(w) = \int G''[f_\eta \eta](w, w'') d\eta(w'') \tilde{h}(w'').$$

$\mathcal{G}$ is well-defined as a bounded operator, since Assumption 1 implies that $|G''[f_\eta \eta](w, w')| \leq L_0$. Note that $G''[f_\eta \eta](w, w'')$ is symmetric in $w$ and $w''$, and that by convexity of $G$, $G''[f_\eta \eta](w, w'') \geq 0$ for all $w, w''$. Consequently, $\mathcal{G}$ is a symmetric positive-semi-definite operator from $L^2_\eta(\mathcal{W})$ to itself.

On the other hand, let $V_1(\cdot) = -G''[f_\eta \eta](\cdot, w') f_\eta(w')$. By Lem. D.4 we have

$$\|V_1\|_{L^2_\eta} \leq \sup_{\mathcal{W}} |V_1| \leq \sup_{\mathcal{W} \times \mathcal{W}} |G''[f_\eta \eta]| \cdot \sup_{\mathcal{W}} |f_\eta| \leq L_0 \cdot \frac{1}{\lambda} \sqrt{2L_0 G(0)} =: \overline{V}_1.$$

Also let $V_2(\cdot) = -\langle s', \nabla_{w'} [G''[f_\eta \eta](\cdot, w') f_\eta(w')] \rangle_{w'}$. Then by Lem. D.4,

$$\|V_2\|_{L^2_\eta} \leq \sup_{\mathcal{W}} |V_2| \leq \sup_{w, w'} \|\nabla_{w'} G''[f_\eta \eta](w, w')\| \cdot \sup_{\mathcal{W}} |f_\eta| + \sup_{\mathcal{W} \times \mathcal{W}} |G''[f_\eta \eta]| \cdot \sup_{\mathcal{W}} \|\nabla f_\eta\|$$

$$\leq L_1 \cdot \frac{1}{\lambda} \sqrt{2L_0 G(0)} + L_0 \cdot \left( \frac{L_1}{\lambda^2} \sqrt{2L_0 G(0)} + \frac{B_1}{\lambda} \right)$$

$$= \left( 1 + \frac{L_0}{\lambda} \right) \frac{L_1}{\lambda} \sqrt{2L_0 G(0)} + \frac{L_0 B_1}{\lambda} =: \overline{V}_2.$$

Denote by $\tilde{h}$ the restriction of $h$ to $\mathrm{supp}(\eta)$ viewed as an element of $L^2_\eta(\mathcal{W})$. Then, denoting by $\mathrm{id}$ the identity operator on $L^2_\eta(\mathcal{W})$, we may rewrite the assumption as $(\lambda \, \mathrm{id} + \mathcal{G})\tilde{h} = V_j$ for $j = 1$ or $2$, and so

$$\sqrt{\int |h|^2 \, d\eta} = \left\| \tilde{h} \right\|_{L^2_\eta} = \left\| (\lambda \, \mathrm{id} + \mathcal{G})^{-1} V_j \right\|_{L^2_\eta} \leq \lambda^{-1} \|V_j\|_{L^2_\eta} \leq \lambda^{-1} \overline{V}_j$$

since $\mathcal{G}$ is positive-semi-definite and $\lambda > 0$. Thus for any $w \in \mathcal{W}$, we get the point-wise bound

$$\lambda h(w) = V_j(w) - \int d\eta(w'') G''[f_\eta \eta](w, w'') h(w'')$$

$$\lambda |h(w)| \leq |V_j(w)| + \int d\eta(w'') |G''[f_\eta \eta](w, w'')| |h(w'')|$$

$$\leq \overline{V}_j + \|G''[f_\eta \eta](w, \cdot)\|_{L^2_\eta} \|h\|_{L^2_\eta}$$

$$\leq \overline{V}_j + L_0 \cdot \lambda^{-1} \overline{V}_j. \qquad \square$$

**Lemma D.7.** *Under Assumption 1, let $\eta, \eta' \in \mathcal{P}(\mathcal{W})$ and let $f_\eta, f_{\eta'}$ as in (D.1). Then there exist constants $H, H'$ dependent only on $\lambda^{-1}, G(0)$ and $L_0, L_1, B_1, \tilde{L}_2$ such that*

$$\sup_{\mathcal{W}} |f_\eta - f_{\eta'}| \leq H W_2(\eta, \eta') \qquad and \qquad \sup_{w \in \mathcal{W}} \|\nabla f_\eta - \nabla f_{\eta'}\|_w \leq H' W_2(\eta, \eta').$$

*Proof.* For each $w \in \mathcal{W}$, we denote the first variation of $\eta \mapsto f_\eta(w)$ by $w' \mapsto \frac{\delta f_\eta(w)}{\delta\eta(dw')}$. Let us show that this quantity is uniformly bounded.[8] By definition, for any $w \in \mathcal{W}$ and $\eta \in \mathcal{P}(\mathcal{W})$ and $w' \in \mathcal{W}$,

$$\lambda f_\eta(w) + G'[f_\eta \eta](w) = 0$$

$$\text{so} \quad \lambda \frac{\delta f_\eta(w)}{\delta\eta(w')} + G''[f_\eta \eta](w, w') f_\eta(w') + \int (G''[f_\eta \eta](w, \cdot)) \, d\left( \eta \frac{\delta f_\eta(\cdot)}{\delta\eta(w')} \right) = 0$$

$$\lambda \frac{\delta f_\eta(w)}{\delta\eta(w')} + \int G''[f_\eta \eta](w, w'') \eta(dw'') \frac{\delta f_\eta(w'')}{\delta\eta(w')} = -G''[f_\eta \eta](w, w') f_\eta(w'). \quad \text{(D.3)}$$

---

[8]The rigorous proof that the first variation $\frac{\delta f_\eta(w)}{\delta\eta(dw')}$ is well-defined for all $w, w' \in \mathcal{W}$ and $\eta \in \mathcal{P}(\mathcal{W})$ would follow from the same derivations as for the uniform bound, so we omit it here.

So by Lem. D.6 applied to $h = \frac{\delta f_\eta(\cdot)}{\delta\eta(w')}$, we indeed have that $\frac{\delta f_\eta(w)}{\delta\eta(w')}$ is bounded by a constant uniformly in $w, w'$ and $\eta$.

Let us now show that

$$
\sup_{w\in\mathcal{W}} \sup_{\eta\in\mathcal{P}(\mathcal{W})} \sup_{w'\in\mathcal{W}} \left\| \nabla_{w'} \frac{\delta f_\eta(w)}{\delta\eta(\mathrm{d}w')} \right\|_{w'} \leq H
$$

for a constant $H$ depending only on $\lambda^{-1}, L_0, L_1, B_1, G(0)$. Indeed, it suffices to show that for any $s' \in T_{w'}\mathcal{W}$ such that $\|s'\|_{w'} = 1$, $\left| \left\langle s', \nabla_{w'} \frac{\delta f_\eta(w)}{\delta\eta(\mathrm{d}w')} \right\rangle_{w'} \right| \leq H$. Now, starting from (D.3) – which holds for all $w, w', \eta$ – and differentiating with respect to $w'$ in the direction $s'$, we get that

$$
\lambda \left\langle s', \nabla_{w'} \frac{\delta f_\eta(w)}{\delta\eta(w')} \right\rangle_{w'} + \int G''[f_\eta\eta](w, w'')\eta(\mathrm{d}w'') \left\langle s', \nabla_{w'} \frac{\delta f_\eta(w'')}{\delta\eta(w')} \right\rangle_{w'}
$$
$$
= -\left\langle s', \nabla_{w'} \left[ G''[f_\eta\eta](w, w') f_\eta(w') \right] \right\rangle_{w'}
$$

and so $h(w) = \left\langle s', \nabla_{w'} \frac{\delta f_\eta(w)}{\delta\eta(\mathrm{d}w')} \right\rangle_{w'}$ satisfies the conditions of Lem. D.6, which proves the claim.

Next let us show that

$$
\sup_{\substack{w\in\mathcal{W} \\ \|s\|_w=1}} \sup_{s\in T_w\mathcal{W}} \sup_{\eta\in\mathcal{P}(\mathcal{W})} \sup_{w'\in\mathcal{W}} \left\| \nabla_{w'} \frac{\delta\langle s, \nabla f_\eta(w)\rangle_w}{\delta\eta(\mathrm{d}w')} \right\|_{w'} \leq H'
$$

for a constant $H'$ depending only on $\lambda^{-1}, L_0, L_1, B_1, G(0)$ and $\widetilde{L}_2$. Indeed, starting from (D.3) and differentiating with respect to $w'$ in the direction $s'$, and differentiating with respect to $w$ in the direction $s$, we get

$$
\lambda \left\langle s', \nabla_{w'} \frac{\delta\langle s, \nabla f_\eta(w)\rangle_w}{\delta\eta(w')} \right\rangle_{w'} + \int \nabla_w G''[f_\eta\eta](w, w'')\eta(\mathrm{d}w'') \left\langle s', \nabla_{w'} \frac{\delta f_\eta(w'')}{\delta\eta(w')} \right\rangle_{w'}
$$
$$
= -\left\langle s, \nabla_w \left\{ \left\langle s', \nabla_{w'} \left[ G''[f_\eta\eta](w, w') f_\eta(w') \right] \right\rangle_{w'} \right\} \right\rangle_w
$$

and so

$$
\lambda \left\| \nabla_{w'} \frac{\delta\langle s, \nabla f_\eta(w)\rangle_w}{\delta\eta(\mathrm{d}w')} \right\|_{w'} \leq \|\nabla_w \nabla_{w'} G''[f_\eta\eta]\| \cdot |f_\eta(w')| + \|\nabla_w G''[f_\eta\eta]\|_w \cdot \|\nabla f_\eta(w')\|_{w'}
$$
$$
+ \sup_{w''\in\mathcal{W}} \|\nabla_w G''[f_\eta\eta](w, w'')\|_w \cdot \sup_{w''\in\mathcal{W}} \left\| \nabla_{w'} \frac{\delta f_\eta(w'')}{\delta\eta(\mathrm{d}w')} \right\|_{w'}
$$
$$
\leq \widetilde{L}_2 \cdot \frac{1}{\lambda}\sqrt{2L_0 G(0)} + L_1 \cdot \left( \frac{L_1}{\lambda^2}\sqrt{2L_0 G(0)} + \frac{B_1}{\lambda} \right) + L_1 \cdot H =: H'
$$

by Assumption 1.

Now fix $w \in \mathcal{W}$. By Lem. D.8 below applied to $F(\eta) = f_\eta(w)$, we have that

$$
|f_\eta(w) - f_{\eta'}(w)| \leq \sup_{\eta''\in\mathcal{P}(\mathcal{W})} \sup_{w'\in\mathcal{W}} \left\| \nabla_{w'} \frac{\delta f_{\eta''}(w)}{\delta\eta''(\mathrm{d}w')} \right\|_{w'} W_2(\eta, \eta') \leq H W_2(\eta, \eta').
$$

Likewise, fix any $w \in \mathcal{W}$ and let $s = \frac{\nabla f_{\eta'}(w) - \nabla f_\eta(w)}{\|\nabla f_{\eta'}(w) - \nabla f_\eta(w)\|_w} \in T_w\mathcal{W}$. Then by Lem. D.8 below applied to $F(\eta) = \langle s, \nabla f_\eta(w)\rangle_w$,

$$
\|\nabla f_{\eta'}(w) - \nabla f_\eta(w)\| = \langle s, \nabla f_{\eta'}(w)\rangle_w - \langle s, \nabla f_\eta(w)\rangle_w \leq H' W_2(\eta, \eta'). \qquad \square
$$

**Lemma D.8.** *Let $\mathcal{W}$ a compact Riemannian manifold and $F : \mathcal{P}(\mathcal{W}) \to \mathbb{R}$ such that*

$$
\forall\eta\in\mathcal{P}(\mathcal{W}), \forall w\in\mathcal{W}, \ \|\nabla F'[\eta](w)\|_w \leq B.
$$

*Then*

$$
\forall\eta, \eta'\in\mathcal{P}(\mathcal{W}), \ |F(\eta) - F(\eta')| \leq B W_1(\eta, \eta') \leq B W_2(\eta, \eta').
$$

*Proof.* For any $x, y \in \mathcal{W}$, pose $(\Sigma_\theta(x, y))_{\theta \in [0,1]}$ the constant-speed length-minimizing geodesic in $\mathcal{W}$ interpolating between $x$ and $y$. Also pose $\Sigma'_\theta(x, y) = \frac{d}{d\theta} \Sigma_\theta(x, y) \in T_{\Sigma_\theta(x,y)}\mathcal{W}$ for any $\theta$. For example if $\mathcal{W} = \mathbb{R}^d$, $\Sigma_\theta(x, y) = x + \theta(y - x)$ and $\Sigma'_\theta(x, y) = y - x$ for all $\theta$.

Let $\gamma$ the optimal coupling between $\eta, \eta'$ in the $W_1$ sense, and for all $\theta \in [0, 1]$, $\eta_\theta = (\Sigma_\theta)_\sharp \gamma$ the pushforward measure of $\gamma$ by $\Sigma_\theta$. Note that for any $\theta \in [0, 1]$,

$$\frac{d}{d\theta} F(\eta_\theta) = \int_{\mathcal{W}} F'[\eta_\theta] \mathrm{d}\, (\partial_\theta \eta_\theta)$$

and that

$$\forall \varphi : \mathcal{W} \to \mathbb{R}, \ \frac{d}{d\theta} \int_{\mathcal{W}} \varphi \mathrm{d}\eta_\theta = \frac{d}{d\theta} \iint_{\mathcal{W} \times \mathcal{W}} \varphi(\Sigma_\theta(x, y)) \mathrm{d}\gamma(x, y)$$

$$= \iint_{\mathcal{W} \times \mathcal{W}} \frac{d}{d\theta} \varphi(\Sigma_\theta(x, y)) \mathrm{d}\gamma(x, y)$$

$$= \iint_{\mathcal{W} \times \mathcal{W}} \langle \Sigma'_\theta(x, y), \nabla\varphi(\Sigma_\theta(x, y)) \rangle_{\Sigma_\theta(x,y)} \mathrm{d}\gamma(x, y).$$

(The interchange of $\frac{d}{d\theta}$ and $\iint_{\mathcal{W} \times \mathcal{W}}$ on the second line can be justified by the dominated convergence theorem assuming that $\varphi$ has bounded $\mathcal{C}^1$ norm, which is the case of $F'[\eta_\theta]$ by assumption.) So by Cauchy-Schwarz inequality,

$$\frac{d}{d\theta} F(\eta_\theta) = \iint_{\mathcal{W} \times \mathcal{W}} \langle \Sigma'_\theta(x, y), \nabla F'[\eta_\theta](\Sigma_\theta(x, y)) \rangle_{\Sigma_\theta(x,y)} \mathrm{d}\gamma(x, y)$$

$$\left| \frac{d}{d\theta} F(\eta_\theta) \right| \le \iint_{\mathcal{W} \times \mathcal{W}} \| \Sigma'_\theta(x, y) \|_{\Sigma_\theta(x,y)} \cdot \| \nabla F'[\eta_\theta](\Sigma_\theta(x, y)) \|_{\Sigma_\theta(x,y)} \mathrm{d}\gamma(x, y)$$

$$\le \sup_{w \in \mathcal{W}} \sup_{\eta' \in \mathcal{P}(\mathcal{W})} \| \nabla F'[\eta](w) \|_w \cdot \iint_{\mathcal{W} \times \mathcal{W}} \| \Sigma'_\theta(x, y) \|_{\Sigma_\theta(x,y)} \mathrm{d}\gamma(x, y)$$

$$\le B \cdot \iint_{\mathcal{W} \times \mathcal{W}} \mathrm{dist}(x, y) \mathrm{d}\gamma(x, y) = B W_1(\eta, \eta')$$

by definition of the geodesic $(\Sigma_\theta(x, y))_{\theta \in [0,1]}$ and by definition of the optimal coupling $\gamma$. Finally,

$$|F(\eta) - F(\eta')| = \left| \int_0^1 \frac{d}{d\theta} F(\eta_\theta) \, \mathrm{d}\theta \right| \le \sup_{\theta \in [0,1]} \left| \frac{d}{d\theta} F(\eta_\theta) \right| \le B W_1(\eta, \eta'). \qquad \square$$

**Proof of the Proposition.**

*Proof of Prop. 3.4.* **We first check** (P0). The fact that $J_\lambda$ is convex is given by Prop. 3.3. Moreover, let any $\beta > 0$ and let us check that $J_{\lambda,\beta} := J_\lambda + \beta^{-1} H \left( \cdot | \tau \right)$ has a minimizer. Indeed, $J_{\lambda,\beta}$ is weakly continuous as shown in Lem. D.5, and non-negative so lower-bounded. Since $\mathcal{W}$ is compact then any set of probability measures on $\mathcal{W}$ is tight, i.e., any sequence in $\mathcal{P}(\mathcal{W})$ has a weakly convergent subsequence. So we conclude by the direct method of calculus of variations: let a sequence $(\eta_n)_n$ such that $J_{\lambda,\beta}(\eta_n) \to \inf_{\mathcal{P}(\mathcal{W})} J_{\lambda,\beta}$ and extract a weakly convergent subsequence with limit $\eta_\infty$; then by weak continuity $\eta_\infty$ is a minimizer of $J_{\lambda,\beta}$.

**We now show that** $J_\lambda$ **satisfies** (P1). Recall from (D.2) that $J'_\lambda[\eta](w) = -\frac{\lambda}{2} |f_\eta|^2 (w)$ with $f_\eta = -\frac{1}{\lambda} G'[f_\eta \eta]$ over $\mathcal{W}$. Let us show the first condition for (P1):

$$\forall \eta \in \mathcal{P}_2(\mathcal{W}), \ \forall w \in \mathcal{W}, \ \max_{\substack{s \in T_w \mathcal{W} \\ \|s\|_w \le 1}} \left| \nabla^2 J'_\lambda[\eta](s, s) \right| \le \Lambda$$

for some $\Lambda < \infty$, where $\nabla^2$ denotes the Riemannian Hessian. We have

$$\nabla J'_\lambda[\eta](w) = -\lambda f_\eta(w) \nabla f_\eta(w)$$

$$\nabla^2 J'_\lambda[\eta](w) = -\lambda f_\eta(w) \nabla^2 f_\eta(w) - \lambda \nabla f_\eta(w) \nabla^\top f_\eta(w)$$

and so, for all $s \in T_w\mathcal{W}$ such that $\|s\|_w \leq 1$,

$$
\begin{aligned}
\left| \nabla^2 J'_\lambda[\eta](s,s) \right| &\leq \lambda |f_\eta| \left\| \nabla^2 f_\eta \right\| + \lambda \left\| \nabla f_\eta \right\|^2 \\
&\leq \sqrt{2L_0 G(0)} \left( \frac{L_2}{\lambda^2} \sqrt{2L_0 G(0)} + \frac{B_2}{\lambda} \right) + \lambda \left( \frac{L_1}{\lambda^2} \sqrt{2L_0 G(0)} + \frac{B_1}{\lambda} \right)^2
\end{aligned}
$$

by Lem. D.4.

Let us now check the second condition for (P1), namely that

$$
\forall w \in \mathcal{W}, \ \forall \eta, \eta' \in \mathcal{P}_2(\mathcal{W}), \ \left\| \nabla J'_\lambda[\eta] - \nabla J'_\lambda[\eta'] \right\|_w \leq \Lambda\, W_2(\eta, \eta')
$$

for some $\Lambda < \infty$. Indeed,

$$
\begin{aligned}
&\left\| \nabla J'_\lambda[\eta] - \nabla J'_\lambda[\eta'] \right\|_w \\
&= \lambda \left\| f_\eta \nabla f_\eta - f_{\eta'} \nabla f_{\eta'} \right\| \leq \lambda \left( \left\| f_\eta (\nabla f_\eta - \nabla f_{\eta'}) \right\| + \left\| (f_\eta - f_{\eta'}) \nabla f_{\eta'} \right\| \right) \\
&\leq \lambda \left( \sup_{\eta''} \sup_{\mathcal{W}} |f_{\eta''}| \cdot \sup_{\mathcal{W}} \left\| \nabla f_\eta - \nabla f_{\eta'} \right\| + \sup_{\eta''} \sup_{\mathcal{W}} \left\| \nabla f_{\eta''} \right\| \cdot \sup_{\mathcal{W}} |f_\eta - f_{\eta'}| \right) \\
&\leq \lambda \left( \frac{1}{\lambda} \sqrt{2L_0 G(0)} \cdot H' W_2(\eta, \eta') + \left( \frac{L_1}{\lambda^2} \sqrt{2L_0 G(0)} + \frac{B_1}{\lambda} \right) \cdot H W_2(\eta, \eta') \right) =: \Lambda W_2(\eta, \eta')
\end{aligned}
$$

by Lem. D.4 and Lem. D.7.

**We now turn to the proof of** (P2) with the quantitative bound on the local LSI constant. Let $\eta \in \mathcal{P}(\mathcal{W})$. By the first part of Lem. D.4, we directly have that

$$
|J'_\lambda[\eta](w)| = \frac{\lambda}{2} |f_\eta|^2(w) \leq \frac{L_0}{\lambda} J_\lambda(\eta).
$$

In particular, by the Holley-Stroock bounded perturbation argument [HS86], the proximal Gibbs measure $\hat{\eta} := e^{-\beta J'_\lambda[\eta]} \tau / Z$ satisfies LSI with constant $\alpha_{\hat{\eta}} = \alpha_\tau \exp\left( -\frac{1}{\lambda} L_0 \beta J_\lambda(\eta) \right)$.

Finally, we turn to the proof of the bound on the uniform LSI constant along the MFLD trajectory $(\eta_t)_{t \geq 0}$. Given the bound on the local LSI constants, it suffices to show that

$$
\forall \eta \in \mathcal{P}(\mathcal{W}), \ J_\lambda(\eta) \leq G(0) \quad \text{and} \quad \forall t \geq 0, \ J_\lambda(\eta_t) \leq J_\lambda(\eta_0) + \beta^{-1} H(\eta_0|\tau).
$$

The first bound was shown in Lem. D.4. For the second bound, note that $J_\lambda(\eta_t) + \beta^{-1} H(\eta_t|\tau)$ decreases with $t$, since MFLD is precisely the Wasserstein gradient flow for $\eta \mapsto J_\lambda(\eta) + \beta^{-1} H(\eta)$ and $H(\eta)$ and $H(\eta|\tau)$ differ by a constant. So, since relative entropy is non-negative,

$$
J_\lambda(\eta_t) \leq J_\lambda(\eta_t) + \beta^{-1} H(\eta_t|\tau) \leq J_\lambda(\eta_0) + \beta^{-1} H(\eta_0|\tau)
$$

for all $t \geq 0$, as desired. $\qquad\square$

# E  Details for Sec. 4 (global convergence by annealing)

The following preliminary lemma allows to control the effect of entropic regularization, using a box-kernel smoothing technique similar to [Chi22a].

**Lemma E.1.** *Let $\mathcal{W}$ a $d$-dimensional compact Riemannian manifold and denote by $\tau$ the uniform probability measure over $\mathcal{W}$. Let $\mathcal{J} : \mathcal{P}(\mathcal{W}) \to \mathbb{R}$ and $\eta^* \in \mathcal{P}(\mathcal{W})$, and suppose that there exist constants $A, B > 0$ such that*

$$
\forall \eta \ \text{s.t.} \ W_1(\eta, \eta^*) \leq A, \quad \mathcal{J}(\eta) - \mathcal{J}(\eta^*) \leq B W_\infty(\eta, \eta^*).
$$

*Denote $\mathcal{J}_\beta = \mathcal{J} + \beta^{-1} H(\cdot|\tau)$, for any $\beta > 0$. Then*

$$
\min_{\eta : W_1(\eta, \eta^*) \leq A} \mathcal{J}_\beta(\eta) \ \leq \ \mathcal{J}(\eta^*) + \inf_{0 < \epsilon \leq \min\{1, A\}} \left[ B\epsilon + \frac{d}{\beta} \log\left( \frac{1}{\epsilon} \right) + \frac{\log C}{\beta} \right]
$$

*where $C := \left[ \inf_{w \in \mathcal{W}} \inf_{0 < \epsilon \leq 1} \epsilon^{-d} \cdot \tau\left( \{ w' ; \text{dist}_{\mathcal{W}}(w, w') \leq \epsilon \} \right) \right]^{-1}$.*

*Proof.* The proof is adapted from [Chi22a]. It is based on constructing an $\epsilon$-smoothed version of $\eta^*$, i.e. a measure $\eta_\epsilon$ which admits a density w.r.t. $\tau$ while being close to $\eta^*$ in an appropriate sense.

Let any $0 < \epsilon \leq \min\{1, A\}$. Given $w \in \mathcal{W}$, define the probability measure $\gamma_{\epsilon,w}(\mathrm{d}w')$ as the uniform probability measure over the geodesic ball $B_\epsilon(w) := \{w \in \mathcal{W}; \mathrm{dist}(w, w') \leq \epsilon\}$. In other words, $\frac{\mathrm{d}\gamma_{\epsilon,w}}{\mathrm{d}\tau}(w') := \frac{\mathbb{1}(w' \in B_\epsilon(w))}{\tau(B_\epsilon(w))}$. Then, let $\gamma_\epsilon(\mathrm{d}w, \mathrm{d}w') = \eta^*(\mathrm{d}w)\gamma_{\epsilon,w}(\mathrm{d}w') \in \mathcal{P}(\mathcal{W} \times \mathcal{W})$, and let $\eta_\epsilon(\mathrm{d}w') = \int_{w \in \mathcal{W}} \gamma_\epsilon(\mathrm{d}w, \mathrm{d}w')$ its second marginal.

One can then verify that

$$\frac{\mathrm{d}\eta_\epsilon}{\mathrm{d}\tau}(w') = \int_{w \in \mathcal{W}} \frac{\mathrm{d}\gamma_{\epsilon,w}}{\mathrm{d}\tau}(w')\eta^*(\mathrm{d}w) = \int_{w \in \mathcal{W}} \frac{\mathbb{1}(w' \in B_\epsilon(w))}{\tau(B_\epsilon(w))}\eta^*(\mathrm{d}w).$$

Moreover there exists a positive constant $C$ such that $\tau(B_\epsilon(w)) \geq C^{-1}\epsilon^d$ for all $\epsilon \leq 1$ [GV79, Theorem 3.3]. As a consequence,

$$H\left(\eta_\epsilon|\tau\right) = \int \mathrm{d}\eta_\epsilon(w') \log \frac{\mathrm{d}\eta_\epsilon}{\mathrm{d}\tau}(w') \leq \sup_{w \in \mathcal{W}} -\log\tau(B_\varepsilon(w)) \leq d\log(1/\epsilon) + \log C.$$

Furthermore, by definition of the coupling $\gamma_\epsilon$, we have $W_1(\eta_\epsilon, \eta^*) \leq W_\infty(\eta_\epsilon, \eta^*) \leq \epsilon \leq A$. Therefore, by assumption $\mathcal{J}(\eta_\epsilon) - \mathcal{J}(\eta^*) \leq BW_\infty(\eta_\epsilon, \eta^*) \leq B\epsilon$, and so

$$\min_{\eta: W_1(\eta, \eta^*) \leq A} \mathcal{J}_\beta(\eta) \leq \mathcal{J}_\beta(\eta_\epsilon) = \mathcal{J}(\eta_\epsilon) + \beta^{-1}H\left(\eta_\epsilon|\tau\right)$$

$$\leq \mathcal{J}(\eta^*) + B\epsilon + \beta^{-1}\left(d\log(1/\epsilon) + \log C\right),$$

and the inequality of the lemma follows by taking the infimum over $\epsilon$. $\qquad\square$

## E.1 Proof of Prop. 4.1

We state and prove a more precise version of Prop. 4.1 below.

**Proposition E.2.** *Under Assumption 1, let $\Delta > 0$ and assume that $\Delta \leq \frac{2L_0 L_1 G(0)}{\lambda^2 J_\lambda^*}$. Then MFLD-Bilevel with the temperature schedule $\forall t, \beta_t = \frac{4d}{\Delta J_\lambda^*}\log\left(\frac{4C^{1/d}B}{\Delta J_\lambda^*}\right)$ converges to $(1 + \Delta)$-multiplicative accuracy in time*

$$T_\Delta \leq \frac{2d}{\alpha_\tau \Delta J_\lambda^*}\log\left(\frac{4C^{1/d}B}{\Delta J_\lambda^*}\right) \cdot \exp\left(\frac{4dL_0 G(0)}{\lambda \Delta J_\lambda^*}\log\left(\frac{4C^{1/d}B}{\Delta J_\lambda^*}\right)\right) \cdot \log\left(\frac{2J_\lambda(\eta_0)}{\Delta J_\lambda^*} + \frac{H\left(\eta_0|\tau\right)}{2\log C}\right)$$

*where $C = \max\left\{1, \left[\inf_{w \in \mathcal{W}} \inf_{0 < \epsilon \leq 1} \epsilon^{-d} \cdot \tau\left(\{w'; \mathrm{dist}_\mathcal{W}(w, w') \leq \epsilon\}\right)\right]^{-1}\right\}$.*

*Proof of Prop. E.2.* Let $(\eta)_t$ the MFLD-Bilevel trajectory with constant inverse temperature parameter $\beta$ to be chosen. Denote $J_{\lambda,\beta} = J_\lambda + \beta^{-1}H\left(\cdot|\tau\right)$. Recall that by Prop. 3.4, $J_{\lambda,\beta}$ satisfies $\alpha_\beta$-LSI uniformly along the MFLD trajectory with $\alpha_\beta = \alpha_\tau \exp\left(-\frac{1}{\lambda}L_0\beta G(0)\right)$. So by Thm. 2.1, for all $t$,

$$J_\lambda(\eta_t) \leq J_{\lambda,\beta}(\eta_t) \leq \inf J_{\lambda,\beta} + e^{-2\beta^{-1}\alpha_\beta t}\left(J_{\lambda,\beta}(\eta_0) - \inf J_{\lambda,\beta}\right) \leq \inf J_{\lambda,\beta} + e^{-2\beta^{-1}\alpha_\beta t}J_{\lambda,\beta}(\eta_0),$$

where in the first inequality we used that $J_{\lambda,\beta} - J_\lambda = \beta^{-1}H\left(\cdot|\tau\right) \geq 0$.

Furthermore, by applying Lem. E.1 to $\mathcal{J} = J_\lambda$, $\eta^* = \arg\min J_\lambda$, $A = \infty$ and $B = \sqrt{2L_0 G(0)} \cdot \left(\frac{L_1}{\lambda^2}\sqrt{2L_0 G(0)} + \frac{B_1}{\lambda}\right)$ the constant from Lem. D.5, we find that

$$\inf J_{\lambda,\beta} \leq \inf J_\lambda + \inf_{0 < \epsilon \leq 1}\left[B\epsilon + \frac{d}{\beta}\log\frac{1}{\epsilon} + \frac{\log C}{\beta}\right].$$

Taking $\beta = \frac{d}{B}s$ for some $s \geq 1$ to be chosen, and evaluating at the infimum at $\epsilon = \frac{d}{\beta B}$, we get

$$\inf J_{\lambda,\beta} \leq J_\lambda^* + \frac{d + \log C'}{\beta} - \frac{d}{\beta}\log\left(\frac{d}{\beta B}\right).$$

where $C' = \max\{1, C\}$. So in order to guarantee that $J_\lambda(\eta_t) \leq (1 + \Delta)J_\lambda^*$, it suffices to take $t$ such that

$$J_\lambda^* + \frac{d + \log C'}{\beta} - \frac{d}{\beta} \log\left(\frac{d}{\beta B}\right) + e^{-2\beta^{-1}\alpha_\beta t}\left(J_\lambda(\eta_0) + \beta^{-1}H(\eta_0|\tau)\right) \leq (1 + \Delta)J_\lambda^*$$

$$\text{i.e.} \quad t \geq \frac{\beta}{2\alpha_\beta} \log\left(\frac{J_\lambda(\eta_0) + \beta^{-1}H(\eta_0|\tau)}{\Delta J_\lambda^* - \left(\frac{d + \log C'}{\beta} - \frac{d}{\beta}\log\left(\frac{d}{\beta B}\right)\right)}\right) =: T_s,$$

assuming that $\Delta$ is large enough so that the above expression is well-defined. More explicitly, substituting the value of $\alpha_\beta$ and of $\beta = \frac{d}{B}s$, we have

$$T_s = \frac{\beta}{2\alpha_\tau} \cdot \exp\left(\frac{1}{\lambda}L_0\beta G(0)\right) \cdot \log\left(\frac{J_\lambda(\eta_0) + \beta^{-1}H(\eta_0|\tau)}{\Delta J_\lambda^* - \left(\frac{d + \log C'}{\beta} - \frac{d}{\beta}\log\left(\frac{d}{\beta B}\right)\right)}\right)$$

$$= \frac{sd/B}{2\alpha_\tau} \cdot \exp\left(s\frac{1}{\lambda B}L_0 dG(0)\right) \cdot \log\left(\frac{J_\lambda(\eta_0) + \frac{B}{sd}H(\eta_0|\tau)}{\Delta J_\lambda^* - \frac{B}{s}(1 + d^{-1}\log C' + \log s)}\right).$$

Noting that

$$\log\frac{s\Delta J_\lambda^*}{4B} = \log s - \log\frac{4B}{\Delta J_\lambda^*} \leq \frac{s\Delta J_\lambda^*}{4B} - 1$$

$$\text{so} \quad \frac{B}{s}(1 + d^{-1}\log C' + \log s) \leq \frac{B}{s}\left(d^{-1}\log C' + \log\frac{4B}{\Delta J_\lambda^*} + \frac{s\Delta J_\lambda^*}{4B}\right)$$

$$= \frac{B}{s}\left(d^{-1}\log C' + \log\frac{4B}{\Delta J_\lambda^*}\right) + \frac{\Delta J_\lambda^*}{4},$$

choose henceforth $s = \max\left\{1, \frac{4B}{\Delta J_\lambda^*}\left(d^{-1}\log C' + \log\frac{4B}{\Delta J_\lambda^*}\right)\right\}$, so that

$$\Delta J_\lambda^* - \frac{B}{s}(1 + d^{-1}\log C' + \log s) \geq \frac{\Delta J_\lambda^*}{2}.$$

To simplify the final statement, we make the assumption that $\Delta$ is small enough so that $1 \leq \frac{4B}{\Delta J_\lambda^*}\left(d^{-1}\log C' + \log\frac{4B}{\Delta J_\lambda^*}\right)$. More explicitly, since we were careful to choose $C' \geq 1$,

$$1 \leq \frac{4B}{\Delta J_\lambda^*}\left(d^{-1}\log C' + \log\frac{4B}{\Delta J_\lambda^*}\right) \iff \frac{\Delta J_\lambda^*}{4B} + \log\frac{\Delta J_\lambda^*}{4B} \leq d^{-1}\log C'$$

$$\impliedby \frac{\Delta J_\lambda^*}{4B} \leq 1 \quad \text{and} \quad \log\frac{\Delta J_\lambda^*}{4B} \leq -1 \iff \frac{\Delta J_\lambda^*}{4B} \leq \min\{1, e^{-1}\} = e^{-1}$$

$$\iff \Delta \leq \frac{4Be^{-1}}{J_\lambda^*} = \frac{4e^{-1}}{J_\lambda^*} \cdot \sqrt{2L_0 G(0)}\left(\frac{L_1}{\lambda^2}\sqrt{2L_0 G(0)} + \frac{B_1}{\lambda}\right)$$

$$\impliedby \Delta \leq \frac{4e^{-1}}{J_\lambda^*} \cdot \frac{2L_0 L_1 G(0)}{\lambda^2} \impliedby \Delta \leq \frac{1}{J_\lambda^*} \cdot \frac{2L_0 L_1 G(0)}{\lambda^2}.$$

Then $s = \frac{4B}{\Delta J_\lambda^*}\left(d^{-1}\log C' + \log\frac{4B}{\Delta J_\lambda^*}\right)$, $\beta = \frac{4d}{\Delta J_\lambda^*}\left(d^{-1}\log C' + \log\frac{4B}{\Delta J_\lambda^*}\right) \geq \frac{4}{\Delta J_\lambda^*}\log C'$, and

$$T_s \leq \frac{\beta}{2\alpha_\tau} \cdot \exp\left(\frac{1}{\lambda}L_0\beta G(0)\right) \cdot \log\left(\frac{J_\lambda(\eta_0) + \beta^{-1}H(\eta_0|\tau)}{\Delta J_\lambda^*/2}\right)$$

$$\leq \frac{2d}{\alpha_\tau\Delta J_\lambda^*}\log\left(\frac{4C'^{1/d}B}{\Delta J_\lambda^*}\right) \cdot \exp\left(\frac{4dL_0 G(0)}{\lambda\Delta J_\lambda^*}\log\left(\frac{4C'^{1/d}B}{\Delta J_\lambda^*}\right)\right) \cdot \log\left(\frac{2J_\lambda(\eta_0)}{\Delta J_\lambda^*} + \frac{H(\eta_0|\tau)}{2\log C'}\right) =: \overline{T}_\Delta.$$

Hence the time-complexity upper bound of $\overline{T}_\Delta$ for reaching $(1 + \Delta)$-multiplicative accuracy. $\quad\square$

**Algorithm 1** Annealing of the MFLD.

---

**Require:** Functional $\mathcal{J} : \mathcal{P}(\mathcal{W}) \to \mathbb{R}$. Initialization $\eta_0, \beta_0 > 0$. Schedule $K$, $(T_k)_{k=0}^K$.
  1: $\eta_0^0 = \eta_0$
  2: **for** $k = 0, \dots, K$ **do**
  3:    $\beta_k = 2^k \beta_0$
  4:    Run the MFLD with $\beta_k$ initialized from $\eta_0^k$ up to $T_k$,
$$\partial_t \eta_t^k = \mathrm{div}(\eta_t^k \nabla \mathcal{J}'[\eta_t^k]) + \frac{1}{\beta_k} \Delta \eta_t^k.$$
  5:    $\eta_0^{k+1} = \eta_{T_k}^k$
  6: **end for**
  7: **return** $\eta_{T_K}^K$.

---

## E.2 General annealing procedure and its convergence guarantee

The following theorem builds upon and generalizes the idea of [SWON23, Sec. 4.1] to objective functionals $\mathcal{J}$ that have a positive optimal value. It ensures fast convergence to a fixed multiplicative accuracy.

**Theorem E.3.** *Let $\mathcal{W}$ a $d$-dimensional compact Riemannian manifold, so in particular the uniform measure $\tau$ over $\mathcal{W}$ satisfies $\alpha_\tau$-LSI for some $\alpha_\tau > 0$. Let $\mathcal{J} : \mathcal{P}(\mathcal{W}) \to \mathbb{R}_+$ convex, suppose that $\mathcal{J}^* := \min \mathcal{J} > 0$ and that there exists a minimizer $\eta^*$. Suppose that there exist constants $\kappa_1, C_L, A > 0$ such that*

*1. $\|\mathcal{J}'[\eta]\|_\infty \leq \kappa_1 \mathcal{J}(\eta)$ for all $\eta \in \mathcal{P}(\mathcal{W})$.*

*2. $\mathcal{J}(\eta) - \mathcal{J}(\eta^*) \leq C_L W_\infty(\eta, \eta^*)$ for all $\eta \in \mathcal{P}(\mathcal{W})$ such that $W_1(\eta, \eta^*) \leq A$.*

*Fix $0 < \delta \leq \frac{C_L \min\{1, A\}}{\mathcal{J}^*}$. Let $\eta_t^k$ the iterates of the annealing procedure of Algorithm 1 with initialization $\beta_0 = d$ and with the schedule $K = \lceil \log_2(1/(\delta \mathcal{J}^*)) \rceil$ and*

$$T_k = 2^{k-1} d \log\left(2^k \mathcal{J}_{\beta_0}(\eta_0)\right) \cdot \alpha_\tau^{-1} \exp\left(2\kappa_1 d \left(\delta^{-1} + \log\left(\frac{C_L C^{1/d}}{\delta \mathcal{J}^*}\right) + 2 + \frac{\mathcal{J}_{\beta_0}(\eta_0)}{2}\right)\right) \quad \text{(E.1)}$$

*where $C := \left[\inf_{w \in \mathcal{W}} \inf_{0 < \epsilon \leq 1} \epsilon^{-d} \cdot \tau\left(\{w'; \mathrm{dist}_\mathcal{W}(w, w') \leq \epsilon\}\right)\right]^{-1}$.*

*Then $\mathcal{J}(\eta_{T_K}^K) \leq \mathcal{J}^* \left(1 + 3\delta + 2\delta \log\left(\frac{C_L C^{1/d}}{\delta \mathcal{J}^*}\right)\right)$, and the total time-complexity is given by*

$$\sum_{k=0}^K T_k \leq \frac{d}{\delta \mathcal{J}^*} \log\left(\frac{\mathcal{J}_{\beta_0}(\eta_0)}{\delta \mathcal{J}^*}\right) \cdot \alpha_\tau^{-1} \exp\left(2\kappa_1 d \left(\delta^{-1} + \log\left(\frac{C_L C^{1/d}}{\delta \mathcal{J}^*}\right) + 2 + \frac{\mathcal{J}_{\beta_0}(\eta_0)}{2}\right)\right).$$

Let us discuss the assumptions of Thm. E.3 and possible generalizations.

- Note that the condition 2. of the theorem holds as soon as $\mathcal{J}'[\eta] : \mathcal{W} \to \mathbb{R}$ is $C_L$-Lipschitz for all $\eta \in \mathcal{P}(\mathcal{W})$, as shown in Lem. D.8, since $W_1 \leq W_2 \leq W_\infty$.

- The annealing procedure and its convergence guarantee can be generalized to a non-compact manifold $\mathcal{W}$ by modifying MFLD to include a confining potential term, as discussed in Sec. A.2.

- Condition 1. of the theorem actually holds for any $\mathcal{J}$ such that $\sup_{\eta, w, w'} |\mathcal{J}''[\eta](w, w')| \leq L < \infty$ and $\mathcal{J}^* > 0$, with the constant $\kappa_1 = \sqrt{\frac{2L}{\mathcal{J}^*}}$. Indeed, one can then show similarly to Lem. D.3 that

$$\|\mathcal{J}'[\eta]\|_\infty^2 \leq 2L \left(\mathcal{J}(\eta) - \mathcal{J}^*\right) \leq 2L \mathcal{J}(\eta) \leq 2L \frac{\mathcal{J}(\eta)^2}{\mathcal{J}^*}.$$

  However, when plugging in $\kappa_1 = \sqrt{2L/\mathcal{J}^*}$ into the bounds of the theorem, one obtains a less favorable dependency of the total time-complexity in $\mathcal{J}^*$. In particular, note that the total time-complexity guaranteed by the theorem scales exponentially in $\kappa_1$ and polynomially in $1/\mathcal{J}^*$.

- The way that the condition 1. of the theorem comes into the proof, is that it allows to guarantee a local LSI constant of $\mathcal{J} + \beta_t^{-1} H$ at $\eta_t$ of $\alpha_{\hat{\eta}_t} = \text{cst} \cdot e^{-\kappa_1 \beta_t \mathcal{J}(\eta_t)}$. One could similarly formulate an annealing procedure, and state convergence guarantees, tailored to objectives $\mathcal{J}$ that satisfy different criteria for LSI, such as the Bakry-Emery curvature-dimension criterion.

The remainder of this subsection is dedicated to proving Thm. E.3.

*Proof of Thm. E.3.* Fix any $0 < \delta \leq \frac{C_L \min\{1, A\}}{\mathcal{J}^*}$. Let, for any $\beta > 0$, $\mathcal{J}_\beta = \mathcal{J} + \beta^{-1} H(\cdot | \tau)$.

By condition 1. of the theorem and the Holley-Stroock bounded perturbation argument, for any $t, k$, the proximal Gibbs measure $\widehat{\eta_t^k} \propto e^{-\beta_k \mathcal{J}'[\eta_t^k]} \tau$ satisfies LSI with the constant

$$\alpha_\tau \exp\left(-\beta_k \kappa_1 \mathcal{J}(\eta_t^k)\right) \geq \inf_{t' \geq 0} \alpha_\tau \exp\left(-\beta_k \kappa_1 \mathcal{J}(\eta_{t'}^k)\right) =: \alpha(k).$$

That is, for any $k$, $\mathcal{J}_{\beta_k}$ satisfies $\alpha(k)$-LSI at $\eta_t^k$ for all $t \geq 0$. (To see that $\alpha(k) > 0$, note that for any $k, t$, $\mathcal{J}(\eta_t^k) \leq \mathcal{J}_{\beta_k}(\eta_t^k) \leq \mathcal{J}_{\beta_k}(\eta_0^k)$, since $H(\cdot|\tau)$ is non-negative and $(\eta_t^k)_t$ is a Wasserstein gradient flow of $\mathcal{J}_{\beta_k}$, and so $\alpha(k) = \inf_{t \geq 0} \alpha_\tau \exp\left(-\beta_k \kappa_1 \mathcal{J}(\eta_t^k)\right) \geq \alpha_\tau \exp\left(-\beta_k \kappa_1 \mathcal{J}_{\beta_k}(\eta_0^k)\right) > 0$; but we will not make use of this rough bound in the sequel.)

Now let

$$T_k = \frac{\beta_k}{2\underline{\alpha}(k)} \log\left(\frac{\beta_k}{d} \overline{c}_k\right)$$

for some $\underline{\alpha}(k) \leq \alpha(k)$ and $\overline{c}_k \geq \mathcal{J}_{\beta_k}(\eta_0^k) - \min \mathcal{J}_{\beta_k}$ to be chosen. Then by Thm. 2.1 applied to $\mathcal{J}_{\beta_k}$, we obtain

$$\mathcal{J}_{\beta_k}(\eta_{T_k}^k) \leq \min \mathcal{J}_{\beta_k} + \exp\left(-2\beta_k^{-1} \alpha(k) T_k\right) \cdot \left(\mathcal{J}_{\beta_k}(\eta_0^k) - \min \mathcal{J}_{\beta_k}\right)$$

$$\leq \min \mathcal{J}_{\beta_k} + \left[\frac{\beta_k}{d}\left(\mathcal{J}_{\beta_k}(\eta_0^k) - \min \mathcal{J}_{\beta_k}\right)\right]^{-1} \cdot \left(\mathcal{J}_{\beta_k}(\eta_0^k) - \min \mathcal{J}_{\beta_k}\right)$$

$$= \min \mathcal{J}_{\beta_k} + \frac{d}{\beta_k}.$$

Further, by Lem. E.1,

$$\mathcal{J}_{\beta_k}(\eta_{T_k}^k) \leq \mathcal{J}^* + \inf_{0 < \epsilon \leq \min\{1, A\}} \left[C_L \epsilon + \frac{d}{\beta_k} \log\left(\frac{1}{\epsilon}\right) + \frac{\log C}{\beta_k}\right] + \frac{d}{\beta_k}$$

$$\leq \mathcal{J}^*(1 + \delta) + \frac{d}{\beta_k} \log\left(\frac{C_L}{\delta \mathcal{J}^*}\right) + \frac{d + \log C}{\beta_k}, \tag{E.2}$$

where the last inequality follows by choosing $\epsilon = \frac{\delta \mathcal{J}^*}{C_L} \leq \min\{1, A\}$ since $\delta \leq \frac{C_L \min\{1, A\}}{\mathcal{J}^*}$.

Then, for all $k \geq 1$ and $t \geq 0$,

$$\beta_k \mathcal{J}(\eta_t^k) \leq \beta_k \mathcal{J}_{\beta_k}(\eta_t^k) \leq \beta_k \mathcal{J}_{\beta_k}(\eta_0^k) = \beta_k \mathcal{J}_{\beta_k}(\eta_{T_{k-1}}^{k-1}) \leq \beta_k \mathcal{J}_{\beta_{k-1}}(\eta_{T_{k-1}}^{k-1}) = 2\beta_{k-1} \mathcal{J}_{\beta_{k-1}}(\eta_{T_{k-1}}^{k-1}),$$

where we used successively that $\mathcal{J}_{\beta_k} - \mathcal{J} = \beta_k^{-1} H(\cdot|\tau) \geq 0$, that $(\eta_t^k)_t$ is a Wasserstein gradient flow for $\mathcal{J}_{\beta_k}$, that $\mathcal{J}_{\beta_{k-1}} - \mathcal{J}_{\beta_k} = (\beta_{k-1}^{-1} - \beta_k^{-1}) H(\cdot|\tau) \geq 0$ since $(\beta_k)_k$ is increasing, and that by definition $\beta_k = 2^k \beta_0$. So by (E.2),

$$\beta_k \mathcal{J}(\eta_t^k) \leq 2\beta_{k-1} \mathcal{J}_{\beta_{k-1}}(\eta_{T_{k-1}}^{k-1}) \leq 2\beta_{k-1} \mathcal{J}^*(1 + \delta) + 2d \log\left(\frac{C_L}{\delta \mathcal{J}^*}\right) + 2d + 2\log C$$

$$\leq 2\frac{d}{\delta}(1 + \delta) + 2d \log\left(\frac{C_L}{\delta \mathcal{J}^*}\right) + 2d + 2\log C$$

$$= 2d\left(\delta^{-1} + \log\left(\frac{C_L}{\delta \mathcal{J}^*}\right) + 2 + \frac{\log C}{d}\right)$$

since our choice of $\beta_0 = d$ and $K = \lceil \log_2(1/(\delta \mathcal{J}^*)) \rceil$ ensures that $\beta_{k-1} \leq \beta_K = 2^K \beta_0 \leq \frac{d}{\delta \mathcal{J}^*}$. For $k = 0$ and all $t \geq 0$, we have more simply $\beta_0 \mathcal{J}(\eta_t^0) \leq \beta_0 \mathcal{J}_{\beta_0}(\eta_t^0) \leq \beta_0 \mathcal{J}_{\beta_0}(\eta_0) = d \mathcal{J}_{\beta_0}(\eta_0)$. As a result, for all $k \geq 0$ we have

$$\forall t \geq 0, \ \beta_k \mathcal{J}(\eta_t^k) \leq 2d \left( \delta^{-1} + \log \left( \frac{C_L}{\delta \mathcal{J}^*} \right) + 2 + \frac{\log C}{d} + \frac{1}{2} \mathcal{J}_{\beta_0}(\eta_0) \right)$$

and so

$$\alpha(k) = \inf_{t \geq 0} \alpha_\tau \exp \left( -\kappa_1 \beta_k \mathcal{J}(\eta_t^k) \right)$$

$$\geq \alpha_\tau \exp \left( -2\kappa_1 d \left( \delta^{-1} + \log \left( \frac{C_L}{\delta \mathcal{J}^*} \right) + 2 + \frac{\log C}{d} + \frac{1}{2} \mathcal{J}_{\beta_0}(\eta_0) \right) \right) =: \underline{\alpha}(k).$$

Moreover, we can choose $\bar{c}_k$ as

$$\mathcal{J}_{\beta_k}(\eta_0^k) = \mathcal{J}_{\beta_k}(\eta_{T_{k-1}}^{k-1}) \leq \mathcal{J}_{\beta_{k-1}}(\eta_{T_{k-1}}^{k-1}) \leq \mathcal{J}_{\beta_{k-1}}(\eta_0^{k-1}) \leq \dots \leq \mathcal{J}_{\beta_0}(\eta_0) \quad \text{by induction,}$$

$$\text{so} \quad \mathcal{J}_{\beta_k}(\eta_0^k) - \min \mathcal{J}_{\beta_k} \leq \mathcal{J}_{\beta_0}(\eta_0) =: \bar{c}_k.$$

Therefore, more explicitly,

$$T_k = \frac{\beta_k}{2\underline{\alpha}(k)} \log \left( \frac{\beta_k}{d} \bar{c}_k \right)$$

$$= \frac{\beta_k}{2} \log \left( \frac{\beta_k}{d} \mathcal{J}_{\beta_0}(\eta_0) \right) \cdot \alpha_\tau^{-1} \exp \left( 2\kappa_1 d \left( \delta^{-1} + \log \left( \frac{C_L}{\delta \mathcal{J}^*} \right) + 2 + \frac{\log C}{d} + \frac{1}{2} \mathcal{J}_{\beta_0}(\eta_0) \right) \right)$$

$$= 2^{k-1} d \cdot \log \left( 2^k \mathcal{J}_{\beta_0}(\eta_0) \right) \cdot \alpha_\tau^{-1} \exp \left( 2\kappa_1 d \left( \delta^{-1} + \log \left( \frac{C_L}{\delta \mathcal{J}^*} \right) + 2 + \frac{\log C}{d} + \frac{1}{2} \mathcal{J}_{\beta_0}(\eta_0) \right) \right)$$

since $\beta_k = 2^k \beta_0 = 2^k d$. Note that

$$\sum_{k=0}^{K} 2^{k-1} \log \left( 2^k \mathcal{J}_{\beta_0}(\eta_0) \right) = \sum_{k=0}^{K} 2^k \frac{\log J_{\beta_0}(\eta_0)}{2} + \sum_{k=0}^{K} k 2^{k-1} \log(2)$$

$$= (2^{K+1} - 1) \frac{\log J_{\beta_0}(\eta_0)}{2} + \log(2) \left( (K-1) 2^K + 1 \right)$$

$$\leq 2^K \log J_{\beta_0}(\eta_0) + \log(2) K 2^K$$

$$\leq \frac{1}{\delta \mathcal{J}^*} \log J_{\beta_0}(\eta_0) + \frac{1}{\delta \mathcal{J}^*} \log \left( \frac{1}{\delta \mathcal{J}^*} \right)$$

$$= \frac{1}{\delta \mathcal{J}^*} \log \left( \frac{J_{\beta_0}(\eta_0)}{\delta \mathcal{J}^*} \right)$$

since $K = \lceil \log_2(1/(\delta \mathcal{J}^*)) \rceil$, hence the announced bound on the total time-complexity $\sum_{k=0}^{K} T_k$.

Finally, at round $K = \lceil \log_2(1/(\delta \mathcal{J}^*)) \rceil$, then $\beta_K = 2^K \beta_0 = 2^K d \in \left[ \frac{1}{2} \frac{d}{\delta \mathcal{J}^*}, \frac{d}{\delta \mathcal{J}^*} \right]$, so by (E.2),

$$\mathcal{J}(\eta_{T_K}^K) \leq \mathcal{J}_{\beta_K}(\eta_{T_K}^K) \leq \mathcal{J}^*(1 + \delta) + \frac{d}{\beta_K} \log \left( \frac{C_L}{\delta \mathcal{J}^*} \right) + \frac{d + \log C}{\beta_K}$$

$$\leq \mathcal{J}^* \left( 1 + 3\delta + 2\delta \frac{\log(C)}{d} + 2\delta \log \left( \frac{C_L}{\delta \mathcal{J}^*} \right) \right),$$

which completes the proof. $\qquad \square$

### E.3 Proof of Thm. 4.2

We state a slightly more precise version of Thm. 4.2 below, and prove it as a corollary of the more general Thm. E.3. Then Thm. 4.2 follows by choosing $\delta = \Theta(\frac{\Delta}{\log(B/(\Delta J_\lambda^*))})$, gathering the constants appearing in the bounds, noting that $J_{\lambda,\beta_0}(\eta_0) \leq J_\lambda(\eta_0) + dH(\eta_0|\tau) \leq G(0) + dH(\eta_0) + d \log \text{vol}(\mathcal{W})$.

**Theorem E.4.** *Under Assumption 1, there exists constants $B = \mathrm{poly}(L_i, B_i, G(0), \lambda^{-1})$ and $C$ dependent only on $\mathcal{W}$ such that the following holds. For any $\delta \leq \frac{B}{J_\lambda^*}$, MFLD-Bilevel with the temperature schedule $(\beta_t)_{t \geq 0}$ defined by $\forall k \leq K, \forall t \in [t_k, t_{k+1}], \beta_t = 2^k d$ where $t_0 = 0$ and $K = \lceil \log_2(1/(\delta \mathcal{J}^*)) \rceil$ and*

$$t_{k+1} - t_k = 2^{k-1} d \log \left( 2^k J_{\lambda, \beta_0}(\eta_0) \right) \cdot \alpha_\tau^{-1} \exp \left( \frac{2L_0 d}{\lambda} \left( \delta^{-1} + \log \left( \frac{BC^{1/d}}{\delta J_\lambda^*} \right) + 2 + \frac{J_{\lambda, \beta_0}(\eta_0)}{2} \right) \right),$$

*achieves $(1 + \Delta)$-multiplicative accuracy, where $\Delta = 3\delta + 2\delta \log \left( \frac{BC^{1/d}}{\delta J_\lambda^*} \right)$, with time-complexity*

$$T_\Delta \leq t_{K+1} \leq \frac{d}{\delta J_\lambda^*} \log \left( \frac{J_{\lambda, \beta_0}(\eta_0)}{\delta J_\lambda^*} \right) \cdot \alpha_\tau^{-1} \exp \left( \frac{2L_0 d}{\lambda} \left( \delta^{-1} + \log \left( \frac{BC^{1/d}}{\delta J_\lambda^*} \right) + 2 + \frac{J_{\lambda, \beta_0}(\eta_0)}{2} \right) \right).$$

*Proof of Thm. 4.2 .* Let us show that the conditions of Thm. E.3 are satisfied, under Assumption 1, for $\mathcal{J} = J_\lambda$. $J_\lambda$ is convex and non-negative, and it is implied throughout Sec. 4.1 that $\inf J_\lambda > 0$, for the notion of convergence to a fixed multiplicative accuracy to apply (Def. 4.1). The existence of a minimizer $\eta^*$ is ensured by the weak convexity of $J_\lambda$, by a similar argument as the proof of (P0) in Sec. D.3. We have the condition 1. with $\kappa_1 = \frac{L_0}{\lambda}$, i.e. $\|J_\lambda'[\eta]\|_\infty \leq \frac{L_0}{\lambda} J_\lambda(\eta)$, by the first part of Lem. D.4. We also have condition 2. with $A = \infty$ and $C_L = B := \sqrt{2L_0 G(0)} \cdot \left( \frac{L_1}{\lambda^2} \sqrt{2L_0 G(0)} + \frac{B_1}{\lambda} \right)$, as shown in Lem. D.5, since $W_1 \leq W_2 \leq W_\infty$.

Note that annealed MFLD-Bilevel with the announced temperature annealing schedule $(\beta_t)_t$, precisely corresponds to Algorithm 1 with the schedule (E.1) applied to $\mathcal{J} = J_\lambda$. So the announced time-complexity bound follows directly from the application of Thm. E.3. $\qquad\square$

## F   Details for Sec. 5 (estimates of the local LSI constant)

We begin by presenting the proof of Prop. 5.1, which states that bounding the LSI constant of $\eta_{\lambda, \beta}$ leads to a local convergence rate.

*Proof of Prop. 5.1.* For any $\eta \in \mathcal{P}(\mathcal{W})$, we denote $\hat{\eta}(\mathrm{d}w) = e^{-\beta J_\lambda'[\eta](w)} \tau(\mathrm{d}w) / Z_\eta$ where $Z_\eta = \int e^{-\beta J_\lambda'[\eta]} \mathrm{d}\tau$. First note that for any $\eta, \eta' \in \mathcal{P}(\mathcal{W})$,

$$\left| \log \frac{\mathrm{d}\hat{\eta}}{\mathrm{d}\hat{\eta}'}(w) + (\log Z_\eta - \log Z_{\eta'}) \right| = \beta \left| J_\lambda'[\eta](w) - J_\lambda'[\eta'](w) \right|$$

$$= \beta \frac{\lambda}{2} \left| f_\eta(w)^2 - f_{\eta'}(w)^2 \right|$$

$$\leq \beta \frac{\lambda}{2} \left( |f_\eta| + |f_{\eta'}| \right)(w) \cdot |f_\eta - f_{\eta'}|(w)$$

$$\leq \beta \frac{\lambda}{2} \cdot 2 \frac{1}{\lambda} \sqrt{2L_0 G(0)} \cdot H W_2(\eta, \eta') =: \widetilde{H} W_2(\eta, \eta')$$

by Lem. D.4 and Lem. D.7, where $H$ is a constant dependent only on $\lambda^{-1}, G(0), L_0, L_1, B_1, \widetilde{L}_2$.

Now suppose that $\eta_{\lambda, \beta} = \arg \min J_{\lambda, \beta} = \widehat{\eta_{\lambda, \beta}}$ satisfies $\alpha^*$-LSI. Let $\varepsilon > 0$ and $\eta_0$ in the $\delta$-sublevel set of $J_{\lambda, \beta}$, i.e., $\eta_0 \in S_\delta := J_{\lambda, \beta}^{-1}((-\infty, \inf J_{\lambda, \beta} + \delta])$, for some $\delta > 0$ to be chosen. Denote by $(\eta_t)_t$ the MFLD trajectory for $J_{\lambda, \beta}$ initialized at $\eta_0$. Note that $S_\delta$ is stable by MFLD since $J_{\lambda, \beta}(\eta_t)$ decreases with $t$. So it suffices to show that $J_{\lambda, \beta}$ satisfies $(\alpha^* - \varepsilon)$-LSI uniformly over $S_\delta$.

Choose any $\eta \in S_\delta$, i.e., such that $J_{\lambda, \beta}(\eta) - \inf J_{\lambda, \beta} \leq \delta$. In particular by Thm. 2.1, it holds

$$\beta^{-1} H (\eta | \eta_{\lambda, \beta}) \leq J_{\lambda, \beta}(\eta) - \inf J_{\lambda, \beta} \leq \delta.$$

Furthermore, since $\eta_{\lambda, \beta}$ satisfies LSI with constant $\alpha^*$ then it also satisfies the following Talagrand inequality, as shown in [OV00]:

$$\forall \eta', \; W_2(\eta', \eta_{\lambda, \beta}) \leq \sqrt{\frac{2}{\alpha^*} H (\eta' | \eta_{\lambda, \beta})}.$$

Then by the inequality noted above, we have

$$\left|\log \frac{\mathrm{d}\hat{\eta}}{\mathrm{d}\eta_{\lambda,\beta}}(w) + c\right| \leq \widetilde{H} W_2(\eta, \eta_{\lambda,\beta}) \leq \widetilde{H}\sqrt{\frac{2}{\alpha^*} H(\eta|\eta_{\lambda,\beta})} \leq \widetilde{H}\sqrt{\frac{2}{\alpha^*}} \cdot \sqrt{\beta\delta} =: M\sqrt{\delta}$$

for some $c \in \mathbb{R}$, and so by the Holley-Stroock bounded perturbation argument, $\hat{\eta}$ satisfies LSI with constant $\alpha^* e^{-M\sqrt{\delta}} \geq \alpha^* - \varepsilon$ for $\delta$ small enough. $\qquad\square$

## F.1 Preliminary estimates for $J_\lambda$ under Assumption 2

Throughout the remainder of this appendix, in the context of Assumption 2, we will use the notations

- the Hilbert space $\mathcal{H} = L^2_\rho(\mathbb{R}^{d+1})$ with the inner product $\langle f, g\rangle_\mathcal{H} = \mathbb{E}_{x\sim\rho} f(x)g(x)$,
- the feature map $\phi : \mathcal{W} \to \mathcal{H}$ given by $\phi(w)(x) = \varphi(\langle w, x\rangle)$,
- the symmetric positive-semi-definite operator in $\mathcal{H}$: $K_\eta = \int \phi(w)\phi(w)^*\mathrm{d}\eta(w)$, where $*$ denotes adjoint in $\mathcal{H}$.
- For any $h \in \mathcal{H}$, we denote by $\langle h, \nabla\phi(w)\rangle_\mathcal{H}$ (resp. $\langle h, \nabla^2\phi(w)\rangle_\mathcal{H}$) the gradient (resp. Hessian) at $w$ of $w \mapsto \langle h, \phi(w)\rangle_\mathcal{H}$.

The usefuless of these notations is justified by Prop. F.1 below, which gives a simplified expression for $J_\lambda$ and $J'_\lambda$.

**Proposition F.1.** *Under Assumption 2, letting the Hilbert space $\mathcal{H} = L^2_\rho(\mathbb{R}^{d+1})$ and the feature map $\phi : \mathcal{W} \to \mathcal{H}$ given by $\phi(w)(x) = \varphi(\langle w, x\rangle)$, we have*

$$J_\lambda(\eta) = \frac{\lambda}{2}\langle y, (K_\eta + \lambda\,\mathrm{id})^{-1}y\rangle_\mathcal{H}, \qquad J'_\lambda[\eta](w) = -\frac{\lambda}{2}\langle \phi(w), (K_\eta + \lambda\,\mathrm{id})^{-1}y\rangle^2_\mathcal{H},$$

*with $K_\eta = \int \phi(w)\phi(w)^*\mathrm{d}\eta(w)$, where $*$ denotes adjoint in $\mathcal{H}$. More explicitly, $K_\eta$ is the integral operator of the kernel $k_\eta(x, x') = \int \varphi(\langle w, x\rangle)\varphi(\langle w, x'\rangle)\mathrm{d}\eta(w)$ with respect to the distribution $x \sim \rho$, i.e.,*

$$\forall h \in \mathcal{H} = L^2_\rho(\mathbb{R}^{d+1}), \quad (K_\eta h)(x) = \mathbb{E}_{x'\sim\rho}[k_\eta(x, x')h(x')] \quad in\ L^2_\rho.$$

*Proof.* Under Assumption 2 we have

$$G(\nu) = \frac{1}{2}\mathbb{E}_{x\sim\rho}\left|\int_\mathcal{W} \varphi(\langle w, x\rangle)\mathrm{d}\nu(w) - y(x)\right|^2 = \frac{1}{2}\left\|\int_\mathcal{W} \phi(w)\mathrm{d}\nu(w) - y\right\|^2_\mathcal{H},$$

so the optimization problem (3.3) defining $J_\lambda(\eta)$, for a fixed $\eta$, writes

$$\min_{f\in L^2_\eta(\mathcal{W})} \frac{1}{2}\left\|\int_\mathcal{W} \phi(w)f(w)\mathrm{d}\eta(w) - y\right\|^2_\mathcal{H} + \frac{\lambda}{2}\int_\mathcal{W} |f|^2(w)\mathrm{d}\eta(w).$$

This problem is strictly convex thanks to the term in $\lambda$, and the FOC is $\forall w,\ \langle\int \phi f\mathrm{d}\eta - y, \phi(w)\eta(\mathrm{d}w)\rangle_\mathcal{H} + \lambda f(w)\eta(\mathrm{d}w) = 0$. So the unique minimum $f_\eta$ is a solution of the fixed point equation $f(w) = -\frac{1}{\lambda}\langle\int \phi f\mathrm{d}\eta - y, \phi(w)\rangle_\mathcal{H}$ in $L^2_\eta(\mathcal{W})$. In particular, denoting $\hat{h}_\eta = -\frac{1}{\lambda}(\int \phi f_\eta\mathrm{d}\eta - y)$, then $f_\eta(w) = \langle\hat{h}_\eta, \phi(w)\rangle_\mathcal{H}$ and, integrating against $\phi\eta$,

$$\int_\mathcal{W} f_\eta(w)\phi(w)\mathrm{d}\eta(w) = \int_\mathcal{W} \phi(w)\,\phi(w)^*\hat{h}_\eta\,\mathrm{d}\eta(w)$$

$$\iff -\lambda\hat{h}_\eta + y = K_\eta\hat{h}_\eta \iff (K_\eta + \lambda\,\mathrm{id})\hat{h}_\eta = y \iff \hat{h}_\eta = (K_\eta + \lambda\,\mathrm{id})^{-1}y,$$

where $a^*b = \langle a, b\rangle_\mathcal{H}$ and $K_\eta = \int_\mathcal{W} \phi(w)\phi(w)^*\mathrm{d}\eta(w)$. So the optimal value $J_\lambda(\eta)$ is

$$J_\lambda(\eta) = \frac{1}{2}\left\|\int_\mathcal{W} \phi(w)f_\eta(w)\mathrm{d}\eta(w) - y\right\|^2_\mathcal{H} + \frac{\lambda}{2}\int_\mathcal{W} |f_\eta|^2(w)\mathrm{d}\eta(w) \qquad\text{(F.1)}$$

$$= \frac{1}{2}\left\|\lambda\hat{h}_\eta\right\|^2_\mathcal{H} + \frac{\lambda}{2}\int_\mathcal{W} \hat{h}_\eta^*\phi(w)\,\phi(w)^*\hat{h}_\eta\,\mathrm{d}\eta(w)$$

$$= \frac{1}{2}\langle\lambda\hat{h}_\eta, \lambda\hat{h}_\eta\rangle_\mathcal{H} + \frac{\lambda}{2}\langle\hat{h}_\eta, K_\eta\hat{h}_\eta\rangle_\mathcal{H} = \frac{1}{2}\langle\lambda\hat{h}_\eta, \lambda\hat{h}_\eta + K_\eta\hat{h}_\eta\rangle_\mathcal{H}$$

$$= \frac{1}{2}\langle\lambda\hat{h}_\eta, y\rangle_\mathcal{H} = \frac{\lambda}{2}\langle y, (K_\eta + \lambda\,\mathrm{id})^{-1}y\rangle_\mathcal{H}.$$

Further, by applying the envelope theorem on (F.1) (and reasoning similarly to the proof of Prop. D.2 to deal with $w \notin \mathrm{supp}(\eta)$, by extending $f_\eta \in L^2_\eta(\mathcal{W})$ into a function $\mathcal{W} \to \mathbb{R}$), we then have

$$\forall w \in \mathcal{W}, \ J'_\lambda[\eta](w) = \left\langle \int \phi f_\eta \mathrm{d}\eta - y, \phi(w) f_\eta(w) \right\rangle_\mathcal{H} + \frac{\lambda}{2} |f_\eta|^2 (w)$$

$$= f_\eta(w) \left\langle -\lambda \hat{h}_\eta, \phi(w) \right\rangle_\mathcal{H} + \frac{\lambda}{2} |f_\eta|^2 (w)$$

$$= -\lambda |f_\eta|^2 (w) + \frac{\lambda}{2} |f_\eta|^2 (w) \ = -\frac{\lambda}{2} |f_\eta|^2 (w) = -\frac{\lambda}{2} \left\langle \hat{h}_\eta, \phi(w) \right\rangle_\mathcal{H}^2.$$

The characterization of $K_\eta$ as the integral operator in $L^2_\rho(\mathbb{R}^{d+1})$ of the kernel $k_\eta(x, x') = \int_\mathcal{W} \phi(w)(x) \, \phi(w)(x') \mathrm{d}\eta(w)$ follows directly from the definition $K_\eta = \int_\mathcal{W} \phi(w)\phi(w)^* \mathrm{d}\eta(w)$, since

$$\forall h \in \mathcal{H}, \ K_\eta h = \int_\mathcal{W} \phi(w) \left\langle \phi(w), h \right\rangle_\mathcal{H} \mathrm{d}\eta(w),$$

$$(K_\eta h)(x) = \int_\mathcal{W} \phi(w)(x) \, \mathbb{E}_{x' \sim \rho} [\phi(w)(x')h(x')] \ \mathrm{d}\eta(w)$$

$$= \mathbb{E}_{x' \sim \rho} \left[ \int_\mathcal{W} \phi(w)(x)\phi(w)(x') \, h(x') \, \mathrm{d}\eta(w) \right]$$

$$= \mathbb{E}_{x' \sim \rho} [k_\eta(x, x')h(x')]. \qquad \square$$

We have the following Wasserstein Lipschitz-continuity properties for the bilevel objective functional $J_\lambda$.

**Proposition F.2.** *Under Assumption 2, suppose furthermore that* $\sup_w \left\| \nabla^i \phi(w) \right\|_\mathcal{H} \le B_i < \infty$ *for* $i \in \{0, 1, 2\}$. *Then for any* $w \in \mathcal{W} = \mathbb{S}^d$ *and any* $\eta, \eta' \in \mathcal{P}(\mathcal{W})$, *it holds*

$$|J_\lambda(\eta) - J_\lambda(\eta')| \le \frac{B_0 B_1}{\lambda} \|y\|_\mathcal{H}^2 \cdot W_1(\eta, \eta')$$

$$and \quad |J'_\lambda[\eta](w) - J'_\lambda[\eta'](w)| \le \frac{2B_0^3 B_1}{\lambda^2} \|y\|_\mathcal{H}^2 \cdot W_1(\eta, \eta')$$

$$and \quad \|\nabla J'_\lambda[\eta](w) - \nabla J'_\lambda[\eta'](w)\|_w \le \frac{4B_0^2 B_1^2}{\lambda^2} \|y\|_\mathcal{H}^2 \cdot W_1(\eta, \eta')$$

$$and \quad \left\| \nabla^2 J'_\lambda[\eta](w) - \nabla^2 J'_\lambda[\eta'](w) \right\|_{\mathrm{op} \ w} \le \frac{4B_0 B_1 (B_0 B_2 + B_1^2)}{\lambda^2} \|y\|_\mathcal{H}^2 \cdot W_1(\eta, \eta').$$

*Proof.* By Prop. F.1,

$$J'_\lambda[\eta](w) = -\frac{\lambda}{2} \left\langle \phi(w), (K_\eta + \lambda \, \mathrm{id})^{-1} y \right\rangle_\mathcal{H}^2 \quad \text{where} \ K_\eta = \int_\mathcal{W} \phi(w'')\phi(w'')^* \, \mathrm{d}\eta(w'')$$

$$\text{so} \quad \nabla J'_\lambda[\eta](w) = -\lambda \left\langle \phi(w), (K_\eta + \lambda \, \mathrm{id})^{-1} y \right\rangle_\mathcal{H} \left\langle \nabla \phi(w), (K_\eta + \lambda \, \mathrm{id})^{-1} y \right\rangle_\mathcal{H} \qquad \text{(F.2)}$$

$$\|\nabla J'_\lambda[\eta]\|_w \le \lambda \|\phi(w)\|_\mathcal{H} \|\nabla \phi(w)\|_w \left\| (K_\eta + \lambda \, \mathrm{id})^{-1} y \right\|_\mathcal{H}^2$$

$$\le \lambda B_0 B_1 \|y\|_\mathcal{H}^2 \left\| (K_\eta + \lambda)^{-1} \right\|_{\mathrm{op}}^2$$

$$\le \frac{1}{\lambda} B_0 B_1 \|y\|_\mathcal{H}^2$$

since $K_\eta$ is positive-semi-definite by definition and so $\left\| (K_\eta + \lambda)^{-1} \right\|_{\mathrm{op}} = \sigma_{\max}((K_\eta + \lambda \, \mathrm{id})^{-1}) = [\sigma_{\min}(K_\eta + \lambda \, \mathrm{id})]^{-1} \le \lambda^{-1}$. So by applying Lem. D.8, this shows the first inequality.

Moreover, the first variation of $K_\eta$ at any $\eta$ is $w' \mapsto \phi(w')\phi(w')^*$, thus by the formula $\partial(X^{-1}) = -X^{-1}(\partial X)X^{-1}$ for the derivative of a matrix inverse,

$$\frac{\delta}{\delta \eta(w')}(K_\eta + \lambda \, \mathrm{id})^{-1} = -(K_\eta + \lambda \, \mathrm{id})^{-1} \cdot \phi(w')\phi(w')^* \cdot (K_\eta + \lambda \, \mathrm{id})^{-1},$$

and so, letting for concision $M = (K_\eta + \lambda\,\mathrm{id})^{-1}$,

$$J_\lambda''[\eta](w, w')$$
$$= -\lambda \left\langle \phi(w), (K_\eta + \lambda\,\mathrm{id})^{-1} y \right\rangle_\mathcal{H} \left\langle \phi(w), -(K_\eta + \lambda\,\mathrm{id})^{-1} \cdot \phi(w')\phi(w')^* \cdot (K_\eta + \lambda\,\mathrm{id})^{-1} y \right\rangle_\mathcal{H}$$
$$= -\lambda \left\langle \phi(w), My \right\rangle_\mathcal{H} \left\langle \phi(w), -M \cdot \phi(w')\phi(w')^* \cdot My \right\rangle_\mathcal{H}$$
$$= \lambda \left\langle \phi(w), My \right\rangle_\mathcal{H} \ \left\langle \phi(w), M\phi(w') \right\rangle_\mathcal{H} \ \left\langle \phi(w'), My \right\rangle_\mathcal{H}.$$

As a result,

$$\nabla_w J_\lambda''[\eta](w, w') = \lambda \left\langle \phi(w'), My \right\rangle_\mathcal{H} \cdot$$
$$\left( \left\langle \nabla\phi(w), My \right\rangle \cdot \left\langle \phi(w), M\phi(w') \right\rangle_\mathcal{H} + \left\langle \phi(w), My \right\rangle \cdot \left\langle \nabla\phi(w), M\phi(w') \right\rangle_\mathcal{H} \right)$$

and, using again that $\|M\|_{\mathrm{op}} = \left\| (K_\eta + \lambda)^{-1} \right\|_{\mathrm{op}} \le \lambda^{-1}$,

$$\left\| \nabla_w J_\lambda''(w, w') \right\|_w \le \lambda B_0 \lambda^{-1} \|y\|_\mathcal{H} \cdot 2 B_0^2 B_1 \lambda^{-2} \|y\|_\mathcal{H} = 2\lambda^{-2} B_0^3 B_1 \|y\|_\mathcal{H}^2.$$

Then applying Lem. D.8 shows the second inequality.

Furthermore, for a fixed $w \in \mathcal{W}$, continuing from the expression of $\nabla_w J_\lambda''[\eta](w, w')$ derived above,

$$\nabla_{w'} \nabla_w J_\lambda''[\eta](w, w')$$
$$= \lambda \left\langle \nabla\phi(w'), My \right\rangle_\mathcal{H} \cdot \left( \left\langle \nabla\phi(w), My \right\rangle \cdot \left\langle \phi(w), M\phi(w') \right\rangle_\mathcal{H} + \left\langle \phi(w), My \right\rangle \cdot \left\langle \nabla\phi(w), M\phi(w') \right\rangle_\mathcal{H} \right)$$
$$+ \lambda \left\langle \phi(w'), My \right\rangle_\mathcal{H} \cdot \left( \left\langle \nabla\phi(w), My \right\rangle \cdot \left\langle \phi(w), M\nabla\phi(w') \right\rangle_\mathcal{H} + \left\langle \phi(w), My \right\rangle \cdot \left\langle \nabla\phi(w), M\nabla\phi(w') \right\rangle_\mathcal{H} \right),$$

so $\left\| \nabla_{w'} \nabla_w J_\lambda''[\eta](w, w') \right\| \le 4\lambda^{-2} B_0^2 B_1^2 \|y\|_\mathcal{H}^2$, and the third inequality follows by applying Lem. D.8 to $\eta \mapsto \left\langle s, \nabla J_\lambda'[\eta](w) \right\rangle_w$ for $s \in T_w \mathcal{W}$ arbitrary.

Finally, by differentiating the expression of $\nabla_{w'} \nabla_w J_\lambda''[\eta](w, w')$ once more with respect to $w$ we get that, for any fixed $w \in \mathcal{W}$,

$$\nabla_{w'} \nabla_w^2 J_\lambda''[\eta](w, w')$$
$$= \lambda \left\langle \nabla\phi(w'), My \right\rangle_\mathcal{H} \cdot \left( \left\langle \nabla^2\phi(w), My \right\rangle \cdot \left\langle \phi(w), M\phi(w') \right\rangle_\mathcal{H} + \left\langle \nabla\phi(w), My \right\rangle \cdot \left\langle \nabla\phi(w), M\phi(w') \right\rangle_\mathcal{H} \right)$$
$$+ \lambda \left\langle \nabla\phi(w'), My \right\rangle_\mathcal{H} \cdot \left( \left\langle \nabla\phi(w), My \right\rangle \cdot \left\langle \nabla\phi(w), M\phi(w') \right\rangle_\mathcal{H} + \left\langle \phi(w), My \right\rangle \cdot \left\langle \nabla^2\phi(w), M\phi(w') \right\rangle_\mathcal{H} \right)$$
$$+ \lambda \left\langle \phi(w'), My \right\rangle_\mathcal{H} \cdot \left( \left\langle \nabla^2\phi(w), My \right\rangle \cdot \left\langle \phi(w), M\nabla\phi(w') \right\rangle_\mathcal{H} + \left\langle \nabla\phi(w), My \right\rangle \cdot \left\langle \nabla\phi(w), M\nabla\phi(w') \right\rangle_\mathcal{H} \right)$$
$$+ \lambda \left\langle \phi(w'), My \right\rangle_\mathcal{H} \cdot \left( \left\langle \nabla\phi(w), My \right\rangle \cdot \left\langle \nabla\phi(w), M\nabla\phi(w') \right\rangle_\mathcal{H} + \left\langle \phi(w), My \right\rangle \cdot \left\langle \nabla^2\phi(w), M\nabla\phi(w') \right\rangle_\mathcal{H} \right),$$

hence $\left\| \nabla_{w'} \nabla_w^2 J_\lambda''[\eta](w, w') \right\| \le \lambda^{-2} \|y\|^2 B_0 B_1 (4B_2 B_0 + 4B_1^2)$, and the fourth inequality follows by applying Lem. D.8 to $\eta \mapsto \left\langle s, \nabla^2 J_\lambda'[\eta](w) \cdot s \right\rangle_w$ for $s \in T_w \mathcal{W}$ arbitrary. $\qquad\square$

The following lemma provides explicit upper estimates of the regularity constants $B_0, B_1, B_2$ of $\phi$ appearing in Prop. F.2, in terms of the activation function $\varphi$ and the data distribution $\rho$.

**Lemma F.3.** *Under Assumption 2, recall that $\phi : \mathcal{W} \to \mathcal{H} = L_\rho^2(\mathbb{R}^{d+1})$ is defined by $\phi(w)(x) = \varphi(\langle w, x \rangle)$, and that $\varphi : \mathbb{R} \to \mathbb{R}$ is $\mathcal{C}^2$. There exists a universal constant $c > 0$ such that*

$$\sup_{w \in \mathbb{S}^d} \|\phi(w)\|_\mathcal{H} \le \|\varphi\|_{L^2(\rho)},$$
$$\sup_{w \in \mathbb{S}^d} \|\nabla\phi(w)\|_\mathcal{H} \le \|\varphi'\|_{L^4(\rho)} N_4(\rho),$$
$$\sup_{w \in \mathbb{S}^d} \left\| \nabla^2\phi(w) \right\|_\mathcal{H} \le \left( \|\varphi''\|_{L^4(\rho)} + \|\varphi'\|_{L^4(\rho)} \right) N_4(\rho)$$

*where*

$$N_4(\rho) := \sup_{\|u\|_2 \le 1} \left( \mathbb{E}_{x \sim \rho} \langle u, x \rangle^4 \right)^{1/4} \quad \text{and} \quad \forall f : \mathbb{R} \to \mathbb{R}, \ \|f\|_{L^p(\rho)} := \sup_{w \in \mathbb{S}^d} \left( \mathbb{E}_{x \sim \rho} |f(\langle w, x \rangle)|^p \right)^{1/p}.$$

*Note that if $\rho$ is rotationally invariant, then $\mathbb{E}_{x \sim \rho} |f(\langle w, x \rangle)|^p$ is independent of $w$, and there exists a universal constant $c$ such that $N_4(\rho) \le c d^{-1/2} \left( \mathbb{E}_{x \sim \rho} \|x\|^4 \right)^{1/4}$.*

*Proof.* For the first inequality, we have by definition

$$\sup_w \|\varphi(w)\|_{\mathcal{H}} = \sup_w \sqrt{\mathbb{E}_{x\sim\rho} |\varphi(\langle w, x\rangle)|^2} = \|\varphi\|_{L^2(\rho)}.$$

For the second inequality, define the orthogonal projector $\Pi_w = I_{d+1} - ww^\top : \mathbb{R}^{d+1} \to T_w\mathbb{S}^d = \{w\}^\perp$ for any $w \in \mathbb{S}^d$. Then $[\nabla\phi(w)](x) = \varphi'(\langle w, x\rangle)\Pi_w x$, so by Cauchy-Schwarz inequalities,

$$\|\nabla\phi(w)\|_{\mathcal{H}} = \left( \sup_{\|f\|_{L^2(\rho)} \le 1} \sup_{\substack{s\in T_w\mathbb{S}^d \\ \|s\|_w = 1}} \mathbb{E}_{x\sim\rho} \left[ f(x) \langle s, \nabla\phi(w)(x)\rangle_w \right] \right)$$

$$= \sup_{\substack{s\in T_w\mathbb{S}^d \\ \|s\|_w = 1}} \mathbb{E}_{x\sim\rho} \left[ \langle s, \nabla\phi(w)(x)\rangle_w^2 \right]^{1/2} = \sup_{\substack{s\in T_w\mathbb{S}^d \\ \|s\|_w = 1}} \mathbb{E}_{x\sim\rho} \left[ |\varphi'(\langle w, x\rangle)|^2 \langle \Pi_w s, x\rangle^2 \right]^{1/2}$$

$$\le \left( \mathbb{E}_{x\sim\rho} \left[ |\varphi'(\langle w, x\rangle)|^4 \right] \right)^{1/4} \cdot \sup_{\|u\|_2 = 1} \left( \mathbb{E}_{x\sim\rho} \left[ \langle u, x\rangle^4 \right] \right)^{1/4}$$

since $\|s\|_w = \|\Pi_w s\|_2$.

For the third inequality, the Riemannian Hessian of $\phi(w) = \varphi(\langle w, \cdot\rangle) : \mathbb{S}^d \to \mathbb{R}$ is given by

$$\left[ \nabla^2\phi(w)\right](x) = \nabla_w^2\varphi(\langle w, x\rangle) = \nabla_w^\top \left[ \varphi'(\langle w, x\rangle)\Pi_w x\right] = \Pi_w \left( \varphi''(\langle w, x\rangle)xx^\top - \varphi'(\langle w, x\rangle)\langle w, x\rangle\right)\Pi_w,$$

so similarly by Cauchy-Schwarz inequalities,

$$\left\|\nabla^2\phi(w)\right\|_{\mathcal{H}} \le \sup_{\substack{s\in T_w\mathbb{S}^d \\ \|s\|_w = 1}} \mathbb{E}_{x\sim\rho} \left[ |\varphi''(\langle w, x\rangle)|^2 \langle s, \Pi_w x\rangle^2 \right]^{1/2} + \mathbb{E}_{x\sim\rho} \left[ |\varphi'(\langle w, x\rangle)|^2 \langle w, x\rangle^2 \right]^{1/2}$$

$$\le \left( \mathbb{E}_{x\sim\rho} \left[ |\varphi''(\langle w, x\rangle)|^4 \right] \right)^{1/4} \cdot \sup_{\substack{s\in T_w\mathbb{S}^d \\ \|s\|_w = 1}} \left( \mathbb{E}_{x\sim\rho} \left[ \langle \Pi_w s, x\rangle^4 \right] \right)^{1/4}$$

$$+ \left( \mathbb{E}_{x\sim\rho} \left[ |\varphi'(\langle w, x\rangle)|^4 \right] \right)^{1/4} \left( \mathbb{E}_{x\sim\rho} \left[ \langle w, x\rangle^4 \right] \right)^{1/4}.$$

Finally, suppose that $\rho$ is rotationally invariant, and let us show that $N_4(\rho) \le cd^{-1/2}\left(\mathbb{E}_{x\sim\rho}\|x\|^4\right)^{1/4}$ for some universal constant $c$. Indeed, for $x \sim \rho$, we have that $x$ and $\bar{x} = x/\|x\|$ are independent and that $\bar{x} \sim \tau$. Therefore,

$$N_4^4(\rho) = \sup_{\|u\|_2 \le 1} \mathbb{E}_{x\sim\rho} \|x\|^4 \langle u, x/\|x\|\rangle^4 = \sup_{\|u\|_2 \le 1} \left( \mathbb{E}_{x\sim\rho} \|x\|^4 \right) \cdot \left( \mathbb{E}_{\bar{x}\sim\tau} \langle u, \bar{x}\rangle^4 \right),$$

and $\sup_{\|u\|_2 \le 1} \mathbb{E}_{\bar{x}\sim\tau} \langle u, \bar{x}\rangle^4 \le \frac{c}{(d+1)^2}$ for some universal constant $c$, which is a direct consequence of the fact that $\langle u, \bar{x}\rangle$ is sub-Gaussian with sub-Gaussian norm $\tilde{c}/\sqrt{d+1}$ for some universal constant $\tilde{c}$ [Ver18, Theorem 3.4.6], along with the moment bound for sub-Gaussian random variables [Ver18, Proposition 2.5.2] $\qquad\square$

Finally, we check rigorously in the following proposition that Assumption 2 with proper additional regularity assumptions on $\varphi$ and $\rho$, is a special case of Assumption 1.

**Proposition F.4.** *Consider $\mathcal{W} = \mathbb{S}^d$ and $G : \mathcal{M}(\mathcal{W}) \to \mathbb{R}$ defined as in Assumption 2. Suppose furthermore that $N_4(\rho), \|\varphi\|_{L^2(\rho)}, \|\varphi'\|_{L^4(\rho)}, \|\varphi''\|_{L^4(\rho)} < \infty$, where $N_4(\rho)$ and $\|\cdot\|_{L^p(\rho)}$ are defined in Lem. F.3. Then, $G$ and $\mathcal{W}$ satisfy Assumption 1.*

*Proof.* The fact that $\mathbb{S}^d$ satisfies $\alpha_\tau$-LSI with $\alpha_\tau = d - 1$ is classical and can be found in [BGL14, Sec. 5.7].

By definition, $G(\nu) = \frac{1}{2} \left\| \int_{\mathcal{W}} \phi(w) \mathrm{d}\nu(w) - y \right\|_{\mathcal{H}}^2$, so $G$ is non-negative and admits second variations: for any $\nu \in \mathcal{M}(\mathcal{W})$ and $w, w' \in \mathbb{S}^d$,

$$G'[\nu](w) = \left\langle \phi(w), \int_{\mathcal{W}} \phi(w') \mathrm{d}\nu(w') - y \right\rangle_{\mathcal{H}}$$

$$G''[\nu](w, w') = \langle \phi(w), \phi(w') \rangle_{\mathcal{H}}$$

$$\text{and} \quad \nabla_w G''[\nu](w, w') = \langle \nabla \phi(w), \phi(w') \rangle_{\mathcal{H}}$$

$$\nabla_w^2 G''[\nu](w, w') = \left\langle \nabla^2 \phi(w), \phi(w') \right\rangle_{\mathcal{H}}$$

$$\nabla_w \nabla_{w'} G''[\nu](w, w') = \langle \nabla \phi(w), \nabla \phi(w') \rangle_{\mathcal{H}} .$$

Consequently, denoting $C_i = \sup_{w \in \mathbb{S}^d} \left\| \nabla^i \phi \right\|_{\mathcal{H}}$ for $i \in \{0, 1, 2\}$, which are all finite by Lem. F.3,

$$|G''[\nu](w, w')| \le C_0^2 =: L_0$$

$$\|\nabla_w G''[\nu](w, w')\|_w \le C_0 C_1 =: L_1$$

$$\left\| \nabla_w^2 G''[\nu](w, w') \right\|_w \le C_0 C_2 =: L_2$$

$$\|\nabla_w \nabla_{w'} G''[\nu](w, w')\| \le C_1^2 =: \widetilde{L}_2 .$$

Now for each $i \in \{0, 1, 2\}$,

$$\forall (\nu, w, w'), \left\| \nabla_w^i G''[\nu](w, w') \right\|_w \le L_i \implies \forall (\nu, \nu', w), \left\| \nabla^i G'[\nu] - \nabla^i G'[\nu'] \right\|_w \le L_i \|\nu - \nu'\|_{TV} .$$

Indeed, the right-hand side can be shown by applying the mean-value theorem to $g(\theta) = \left\langle s, \nabla^i G'[\nu + \theta(\nu' - \nu)](w) \right\rangle_w$ over $\theta \in [0, 1]$ for each $s \in (T_w \mathcal{W})^{\otimes i}$. Thus, to show the existence of $B_i < \infty$ such that $\forall (\nu, w, w'), \left\| \nabla^i G'[\nu] \right\|_w \le L_i \|\nu\|_{TV} + B_i$, it suffices to check that there exists $\nu_0$ such that $\|\nu_0\|_{TV}$ and $\sup_w \left\| \nabla^i G'[\nu_0] \right\|_w < \infty$. Note that for any $\nu$ and $w$,

$$\nabla^i G'[\nu](w) = \left\langle \nabla^i \phi(w), \int_{\mathcal{W}} \phi(w') \mathrm{d}\nu(w') - y \right\rangle_{\mathcal{H}} ,$$

$$\text{thus} \quad \nabla^i G'[0](w) = - \left\langle \nabla^i \phi(w), y \right\rangle_{\mathcal{H}}$$

$$\text{and} \quad \sup_w \left\| \nabla^i G'[0](w) \right\|_w \le C_i \|y\|_{\mathcal{H}} < \infty.$$

Hence the existence of the $B_i < \infty$ is verified. This finishes the verification of Assumption 1. $\quad\square$

## F.2 Proof of Thm. 5.2

In the single-index setting of Assumption 3, it is intuitive that $\delta_v$ is a minimizer of $J_\lambda$, for any $\lambda \ge 0$, and that $\eta_{\lambda, \beta}$ and $\delta_v$ are close in certain regimes of $\beta$ and $\lambda$. For this reason, we will first investigate the properties of $J'_\lambda[\delta_v]$ as a proxy of $J'_\lambda[\eta_{\lambda, \beta}]$, to show that it is amenable to a refined analysis for proving LSI, in Sec. F.2.1. This step uses a Lyapunov approach inspired by [MS14; LE23]. We will then prove that these properties carry from $J'_\lambda[\delta_v]$ over to $J'_\lambda[\eta_{\lambda, \beta}]$, in Sec. F.2.2, thanks to a quantitative bound on $W_2(\eta_{\lambda, \beta}, \delta_v)$ proved in Sec. F.2.3.

**Lemma F.5.** *Under Assumptions 2 and 3, we have*

$$\forall w \in \mathbb{S}^d, J'_\lambda[\delta_v](w) = -\frac{\lambda}{2} \left( \lambda + \|\phi(v)\|_{\mathcal{H}}^2 \right)^{-2} \langle \phi(v), \phi(w) \rangle_{\mathcal{H}}^2$$

$$= -\frac{\lambda}{2} \left( \lambda + \|\varphi\|_{L^2(\rho)}^2 \right)^{-2} \left| \mathbb{E}_{x \sim \rho} \varphi(\langle x, v \rangle) \varphi(\langle x, w \rangle) \right|^2$$

$$= -\lambda g(\langle v, w \rangle)$$

*for some $g : [-1, +1] \to \mathbb{R}$.*

*Proof.* By Prop. F.1, since $y = \phi(v)$,

$$J'_\lambda[\delta_v] = -\frac{\lambda}{2} \left\langle \phi(w), (K_{\delta_v} + \lambda \mathrm{id})^{-1} \phi(v) \right\rangle_{\mathcal{H}}^2 .$$

Since $\phi(v)$ is an eigenvector of $K_{\delta_v} = \int_{\mathcal{W}} \phi(w')\phi(w')^* \mathrm{d}\delta_v = \phi(v)\phi(v)^*$ with eigenvalue $\|\phi(v)\|_{\mathcal{H}}^2 = \mathbb{E}_{x\sim\rho}\varphi(\langle x,v\rangle)^2 = \|\varphi\|_{L^2(\rho)}^2$, it is also an eigenvector of $(K_{\delta_v} + \lambda\,\mathrm{id})^{-1}$ with eigenvalue $(\|\varphi(v)\|_{\mathcal{H}}^2 + \lambda)^{-1}$, whence the expression of $J_\lambda'[\delta_v]$ follows.

Moreover, by rotational invariance of $\rho$, $\mathbb{E}_{x\sim\rho}\varphi(\langle x,v\rangle)\varphi(\langle x,w\rangle)$ depends only on $\langle v,w\rangle$, for all $w \in \mathbb{S}^d$. In other words, there exists $g$ such that $J_\lambda'[\delta_v] = -\lambda g(\langle v,\cdot\rangle)$. $\qquad\square$

### F.2.1 Lyapunov function analysis for bounding the LSI constant of $\widehat{\delta}_v \propto e^{-\beta J_\lambda'[\delta_v]}\tau$

Observe that by the assumption $g' \geq c_1 > 0$ of Thm. 5.2, $J_\lambda'[\delta_v] = -\lambda g(\langle v,\cdot\rangle)$ has a unique global minimum at $v$. Moreover, our other assumptions on $g$ will imply that the Riemannian Hessian at optimum $\nabla^2 J_\lambda'[\delta_v](v)$ is positive definite. This motivates us to follow the strategy of [LE23, Thm. 3.4] for proving LSI for $\widehat{\delta}_v \propto e^{-\beta J_\lambda'[\delta_v]}\tau$. Let us first outline the strategy and recall some useful classical notions.

The generator of the Langevin diffusion with invariant measure $\exp(-\beta f)\tau/Z$ is

$$\mathcal{L} = \Delta - \beta\langle\nabla f,\nabla\rangle. \tag{F.3}$$

Define $U = \{w : \mathrm{dist}_{\mathcal{W}}(w,v) \leq r\}$ for some $v \in \mathbb{S}^d$, with $r > 0$ to be chosen later. We say $W : \mathbb{S}^d \to [1,\infty)$ is a Lyapunov function if $\frac{\mathcal{L}W}{W} \leq -\theta + b\mathbf{1}_U$, for constants $\theta > 0$ and $b \geq 0$. When proving functional inequalities for a Gibbs measure $\exp(-\beta f)\tau/Z$, a typical choice of Lyapunov function is $W = \exp(\beta(f - \min f)/2)$, for which the Lyapunov condition writes

$$\frac{\beta\Delta f}{2} - \frac{\beta^2\|\nabla f\|^2}{4} \leq -\theta + b\mathbf{1}_U. \tag{F.4}$$

Further, we say a probability measure $\nu \in \mathcal{P}(\mathbb{S}^d)$ satisfies a local Poincaré inequality on $U$ with constant $\kappa_U$ if

$$\int_U f^2 \mathrm{d}\nu \leq \frac{1}{\kappa_U}\int_U \|\nabla f\|^2\,\mathrm{d}\nu, \quad \text{for all smooth } f : U \to \mathbb{R} \text{ such that } \int_U f\mathrm{d}\nu = 0.$$

Notice that $U$ has a convex boundary, thus we can use the Bakry-Émery criterion as adapted to manifolds with convex boundaries by [LE23, Proposition B.11] to prove a local Poincaré inequality on $U$. Specifically, it suffices to have $\inf_{w\in U} \lambda_{\min}(\nabla^2 f(w)) > 0$.

In summary, a Lyapunov condition of the form (F.4), along with a control on the eigenspectrum of $\nabla^2 f(w)$, implies an LSI for $e^{-\beta f}\tau/Z$. We record this fact in the theorem below, working out the proper dependence on problem parameters for future use.

**Theorem F.6.** *Let $v \in \mathbb{S}^d$, $0 < \lambda \leq 1$ and $f : \mathbb{S}^d \to \mathbb{R}$ of the form $f(w) = -\lambda g(\langle w,v\rangle)$ for some increasing function $g : [-1,1] \to \mathbb{R}$. Suppose there exist constants $D_0, D_1, D_2, D_3, D_4 > 0$, and $r \in (0,\pi/2)$ such that if $\beta \geq D_0 d\lambda^{-1}$ then*

$$\forall w \in \mathbb{S}^d, \ \frac{1}{2}\Delta f - \frac{\beta}{4}\|\nabla f\|^2 \leq D_1\lambda d \tag{$\mathrm{L}_{\mathbb{S}^d}$}$$

$$\forall w \in \mathbb{S}^d \setminus U, \ \frac{1}{2}\Delta f - \frac{\beta}{4}\|\nabla f\|^2 \leq -D_2\beta\lambda^2 \tag{$\mathrm{L}_U$}$$

$$\forall w \in \mathbb{S}^d, \ \lambda_{\min}(\nabla^2 f(w)) \geq -D_3\lambda \tag{$\mathrm{C}_{\mathbb{S}^d}$}$$

$$\forall w \in U, \ \lambda_{\min}(\nabla^2 f(w)) \geq D_4\lambda \tag{$\mathrm{C}_U$}$$

*where $U = \{w \in \mathbb{S}^d; \ \mathrm{dist}_{\mathcal{W}}(w,v) \leq r\}$. Then (provided that $\beta \geq D_0 d\lambda^{-1}$) the probability measure $\nu = \exp(-\beta f)\tau/Z$ satisfies $\alpha$-LSI for a constant $\alpha$ dependent only on the $D_i$ and on $r$.*

*Furthermore, if the condition on $\beta$ is replaced by $\beta \geq D_0'd^4\lambda^{-4}$ and if ($\mathrm{L}_{\mathbb{S}^d}$) is replaced by*

$$\forall w \in \mathbb{S}^d, \ \frac{1}{2}\Delta f - \frac{\beta}{4}\|\nabla f\|^2 \leq D_1'\lambda d\beta^{3/4}, \tag{$\mathrm{L}_{\mathbb{S}^d}'$}$$

*then (provided that $\beta \geq D_0'd^4\lambda^{-4}$) $\nu$ satisfies $\alpha'$-LSI for a constant $\alpha'$ dependent only on $D_0', D_1', D_2, D_3, D_4$ and on $r$.*

*Proof.* By the Lyapunov criterion for Poincaré inequality [BGL14, Thm. 4.6.2], if the generator $\mathcal{L}$ given by (F.3) satisfies the Lyapunov condition $\frac{\mathcal{L}W}{W} \leq -\theta + b\mathbf{1}_U$ for some $\theta > 0$, $b \geq 0$, $U \subset \mathbb{S}^d$ and $W : \mathbb{S}^d \to \mathbb{R}$, and if $\nu$ satisfies a local Poincaré inequality on $U$ with constant $\kappa_U$, then $\nu$ satisfies a Poincaré inequality on $\mathbb{S}^d$ with constant $\kappa \geq \frac{\theta}{1 + \frac{b}{\kappa_U}}$.

Let us apply this to $W = \exp(\beta(f - \min f)/2)$. By (L$_{\mathbb{S}^d}$) and (L$_U$), the Lyapunov condition holds with $\theta = D_2\beta^2\lambda^2$ and $b = D_1\lambda\beta(d-1) + D_2\beta^2\lambda^2$. Moreover, since $U$ has a convex boundary (the geodesic in $\mathbb{S}^d$ between any two points in $U$ remains in $U$ for $r < \pi/2$), by [LE23, Propostion B.11] $\nu$ satisfies a local Poincaré inequality on $U$ with constant

$$\kappa_U \geq \mathrm{Ric}_g + \beta\lambda_{\min}(\nabla^2 f(w)) \geq d - 1 + \beta\lambda D_4$$

where $\mathrm{Ric}_g$ denotes the Ricci curvature of $\mathbb{S}^d$. As a result, $\nu$ satisfies Poincaré inequality with constant

$$\kappa \geq \frac{D_2\beta^2\lambda^2}{1 + \frac{D_1\lambda\beta d + D_2\beta^2\lambda^2}{d-1+\beta\lambda D_4}} \geq C\beta\lambda, \tag{F.5}$$

for some constant $C$ depending only on the $D_i$, where we used that $\beta \geq D_0 d\lambda^{-1}$.

Moreover, by [LE23, Proposition 9.17], if $\nu \in \mathcal{P}(\mathbb{S}^d)$ satisfies the Poincaré inequality with constant $\kappa$, and $\beta\nabla^2 f + \mathrm{Ric}_g \succcurlyeq -\beta K$ for some $K > 0$ on $\mathbb{S}^d$, then for $\beta \geq 1$, $\nu$ satisfies the LSI with constant $\alpha = \frac{\kappa}{11\beta K}$. By the assumptions of the theorem, this indeed holds with $K = D_3\lambda$. Consequently, $\nu$ satisfies LSI with constant $\alpha = C/(11D_3)$, which finishes the proof of the first part of the theorem.

The second part, with (L$'_{\mathbb{S}^d}$) instead of (L$_{\mathbb{S}^d}$), follows by a similar reasoning, except that "$D_1$" should be replaced by "$D'_1\beta^{3/4}$" in the calculation of (F.5). This still leads to a bound of the form $\kappa \geq C'\beta\lambda$ provided that $\beta \geq D'_0 d^4\lambda^{-4}$, and the rest of the proof follows without change. $\qquad\square$

We now verify that $J'_\lambda[\delta_v]$ satisfies the conditions of Thm. F.6.

**Proposition F.7.** *Under the assumptions of Thm. 5.2, $f_0 := J'_\lambda[\delta_v]$ satisfies the conditions of Thm. F.6 with $D_0, ..., D_4, r$ dependent only on $c_1, C_1, C_2, C_3$.*

*Proof.* The Riemannian gradient and Hessian of $f_0 = J'_\lambda[\delta_v] = -\lambda g(\langle v, \cdot \rangle)$ are given by

$$\nabla f_0(w) = -\lambda g'(\langle w, v \rangle)\Pi_w v$$
$$\text{and} \quad \nabla^2 f_0(w) = -\lambda\Pi_w \left( g''(\langle w, v \rangle)vv^\top - g'(\langle w, v \rangle)\langle w, v \rangle I_{d+1} \right)\Pi_w$$

where $\Pi_w = I_{d+1} - ww^\top : \mathbb{R}^{d+1} \to T_w\mathbb{S}^d = \{w\}^\perp$ for any $w \in \mathbb{S}^d$. This can be shown by considering the smooth extension of $f_0$ to $\mathbb{R}^{d+1} \to \mathbb{R}$ defined by $x \mapsto -\lambda g(\langle v, x \rangle)$ and using that $\mathbb{S}^d$ is a sub-Riemannian manifold of $\mathbb{R}^{d+1}$ [Bou23, Chap. 5]. In particular since $v^\top\Pi_w\Pi_w v = 1 - \langle w, v \rangle^2$ and $\mathrm{Tr}\,\Pi_w = d$,

$$\|\nabla f_0(w)\|^2 = \lambda^2 g'(\langle w, v \rangle)^2(1 - \langle w, v \rangle^2)$$
$$\text{and} \quad \Delta f_0(w) = \mathrm{Tr}\,\nabla^2 f_0(w) = -\lambda\left( g''(\langle w, v \rangle)(1 - \langle w, v \rangle^2) - g'(\langle w, v \rangle)\langle w, v \rangle d \right).$$

Pose $U = \left\{ w \in \mathbb{S}^d : \mathrm{dist}_{\mathbb{S}^d}(w, v) \leq r \right\}$ for some $r > 0$ to be chosen.

Let us verify (L$_{\mathbb{S}^d}$). We have for all $w \in \mathbb{S}^d$

$$\frac{1}{2}\Delta f_0 - \frac{\beta}{4}\|\nabla f_0\|^2 = -\frac{\lambda}{4}\left(2g''(\langle w, v \rangle) + \beta\lambda g'(\langle w, v \rangle)^2\right)(1 - \langle w, v \rangle^2) + \frac{\lambda}{2}g'(\langle w, v \rangle)\langle w, v \rangle d. \tag{F.6}$$

The second term is bounded by $\frac{\lambda}{2}C_1 d$. We can ensure that the first term is non-positive by appropriately restricting $\beta$ as follows:

$$\inf_{[-1,1]}\left[2g'' + \beta\lambda(g')^2\right] \geq 0 \impliedby 2(\inf g'') + \beta\lambda(\inf g')^2 \geq 0$$

$$\impliedby -2C_2 + \beta\lambda c_1^2 \geq 0 \iff \beta \geq \frac{2C_2}{c_1^2}\lambda^{-1}.$$

Let us verify ($L_U$). We can upper-bound the first term in (F.6) by a negative quantity by restricting $\beta$ further: by a similar calculation as just above,

$$\beta \geq \frac{4C_2}{c_1^2 \lambda} \implies \inf_{[-1,1]} \left[ 2g'' + \frac{\beta}{2}\lambda(g')^2 \right] \geq 0 \implies 2g'' + \beta\lambda(g')^2 \geq \frac{\beta}{2}\lambda(g')^2 \text{ over } [-1,1].$$

Then for all $w \in \mathbb{S}^d \setminus U$, we have $r \leq \mathrm{dist}_{\mathcal{W}}(w,v) = \arccos(\langle w,v\rangle) \leq \frac{\pi}{2}\sqrt{1 - \langle w,v\rangle^2}$, and so

$$\frac{1}{2}\Delta f_0 - \frac{\beta}{4}\|\nabla f_0\|^2 \leq -\frac{\lambda}{4}\left(\frac{1}{2}\beta\lambda g'(\langle w,v\rangle)^2\right)(1 - \langle w,v\rangle^2) + \frac{\lambda}{2}g'(\langle w,v\rangle)\langle w,v\rangle d$$

$$= \frac{\lambda}{4}g'(\langle w,v\rangle)\left\{ -\frac{\beta\lambda}{2}g'(\langle w,v\rangle)(1 - \langle w,v\rangle^2) + 2\langle w,v\rangle d \right\}$$

$$\leq \frac{\lambda}{4}g'(\langle w,v\rangle)\left\{ -\frac{2\beta\lambda c_1 r^2}{\pi^2} + 2\langle w,v\rangle d \right\}$$

$$\leq -\frac{\lambda}{4}g'(\langle w,v\rangle)\cdot\frac{\beta\lambda c_1 r^2}{\pi^2} \leq -\frac{c_1^2}{4\pi^2}\beta\lambda^2 r^2$$

provided that $\beta \geq \frac{2\pi^2 d}{\lambda c_1 r^2}$.

To verify ($C_{\mathbb{S}^d}$), simply note that, since $\left\|\Pi_w vv^\top \Pi_w\right\|_{\mathrm{op}} = \|\Pi_w v\|^2 = 1 - \langle w,v\rangle^2$,

$$\forall w, \ \left\|\nabla^2 f_0(w)\right\|_{\mathrm{op}} \leq \lambda g''(\langle w,v\rangle)(1 - \langle w,v\rangle^2) + \lambda C_1$$

$$\leq \lambda\left[ \sup_{s\in[-1,1]} g''(s)(1 - s^2) \right] + \lambda C_1 \leq (C_3 + C_1)\lambda,$$

and therefore, $\inf_{w\in\mathbb{S}^d}\lambda_{\min}(\nabla^2 f_0(w)) \geq -\left(\sup_w \left\|\nabla^2 f_0(w)\right\|_{\mathrm{op}}\right) \geq -(C_3 + C_1)\lambda$.

Finally, let us verify ($C_U$). Indeed, for any $w \in U$,

$$\lambda_{\min}(\nabla^2 f_0(w)) = \min_{\|u\|^2=1, \langle u,w\rangle=0} -\lambda g''(\langle w,v\rangle)\langle u,v\rangle^2 + \lambda g'(\langle w,v\rangle)\langle w,v\rangle$$

$$\geq -\lambda\left|g''(\langle w,v\rangle)\right| \max_{\|u\|^2=1, \langle u,w\rangle=0} \langle u,v\rangle^2 + \lambda c_1\langle w,v\rangle$$

$$= -\lambda\left|g''(\langle w,v\rangle)\right|(1 - \langle w,v\rangle^2) + \lambda c_1\langle w,v\rangle,$$

where the bound of the second term follows from $\langle w,v\rangle \geq 0$, which can be ensured by taking $r \leq \frac{\pi}{2}$. Since $w \in U \iff \langle w,v\rangle \geq \cos(r) \geq 1 - r^2$, it follows that

$$\lambda_{\min}(\nabla^2 f_0(w)) \geq -\lambda\left[ \sup_{\cos r\leq s\leq 1} \left|g''(s)\right|(1 - s^2) \right] + \lambda c_1\cos r$$

$$\geq -\lambda C_3\left[ \sup_{\cos r\leq s\leq 1} \sqrt{1 - s^2} \right] + \lambda c_1\cos r$$

$$= \lambda\left(-C_3\sin r + c_1\cos r\right) \geq \lambda\frac{c_1}{2}$$

for a certain choice of $r$ small enough, dependent only on $c_1$ and $C_3$. $\qquad\square$

### F.2.2 Lyapunov function analysis for bounding the LSI constant of $\eta_{\lambda,\beta}$

To prove Thm. 5.2, it only remains to show that the conditions of Thm. F.6 are satisfied for $J'_\lambda[\eta_{\lambda,\beta}]$ instead of $J'_\lambda[\delta_v]$.

**Lemma F.8.** *Under the setting of Assumptions 2 and 3, $\eta_{\lambda,\beta}$ is rotationally invariant except for the direction $v$, or formally $Rv = v \implies R_\sharp\eta_{\lambda,\beta} = \eta_{\lambda,\beta}$ for orthonormal matrices $R$, where $R_\sharp\eta$ denotes the pushforward measure. Moreover, there exists $g_\eta : [-1,1] \to \mathbb{R}$ such that for all $w \in \mathbb{S}^d$, $J'_\lambda[\eta_{\lambda,\beta}](w) = -\lambda g_\eta(\langle w,v\rangle)$.*

*Proof.* The lemma follows directly from the fact that $\rho$ is rotationally invariant and that $y = \phi(v)$. $\qquad\square$

**Lemma F.9.** *Under Assumption 2, suppose furthermore that* $\sup_w \left\| \nabla^i \phi(w) \right\|_{\mathcal{H}} \leq B_i < \infty$ *for* $i \in \{0, 1, 2\}$. *Then we have, for any* $\eta, \eta' \in \mathcal{P}(\mathcal{W})$,

$$\forall w \in \mathbb{S}^d, \ \left| \frac{1}{2} \Delta J'_\lambda[\eta] - \frac{\beta}{4} \left\| \nabla J'_\lambda[\eta] \right\|^2 - \frac{1}{2} \Delta J'_\lambda[\eta'] + \frac{\beta}{4} \left\| \nabla J'_\lambda[\eta'] \right\|^2 \right|$$

$$\leq \left( d \frac{2 B_0 B_1 (B_0 B_2 + B_1^2)}{\lambda^2} \left\| y \right\|_{\mathcal{H}}^2 + \beta \frac{2 B_0^3 B_1^3}{\lambda^3} \left\| y \right\|_{\mathcal{H}}^4 \right) W_1(\eta, \eta')$$

*and* $\quad \left| \lambda_{\min}(\nabla^2 J'_\lambda[\eta]) - \lambda_{\min}(\nabla^2 J'_\lambda[\eta']) \right| \leq \dfrac{4 B_0 B_1 (B_0 B_2 + B_1^2)}{\lambda^2} \left\| y \right\|_{\mathcal{H}}^2 W_1(\eta, \eta').$

*Proof.* By Prop. F.2,

$$\left\| \nabla^2 J'_\lambda[\eta](w) - \nabla^2 J'_\lambda[\eta'](w) \right\|_{\mathrm{op}} \leq \frac{4 B_0 B_1 (B_0 B_2 + B_1^2)}{\lambda^2} \left\| y \right\|_{\mathcal{H}}^2 W_1(\eta, \eta')$$

and $\left| \lambda_{\min}(\nabla^2 J'_\lambda[\eta](w)) - \lambda_{\min}(\nabla^2 J'_\lambda[\eta'](w)) \right| \leq \left\| \nabla^2 J'_\lambda[\eta](w) - \nabla^2 J'_\lambda[\eta'](w) \right\|_{\mathrm{op}}$ by Weyl's inequality. This shows the second inequality of the lemma.

For the first inequality, we have $\Delta J'_\lambda[\eta](w) = \mathrm{Tr}\, \nabla^2 J'_\lambda[\eta](W)$ and so

$$\left| \frac{1}{2} \Delta J'_\lambda[\eta] - \frac{1}{2} \Delta J'_\lambda[\eta'] \right| \leq \frac{d}{2} \left\| \nabla^2 J'_\lambda[\eta](w) - \nabla^2 J'_\lambda[\eta'](w) \right\|_{\mathrm{op}} \leq \frac{d}{2} \frac{4 B_0 B_1 (B_0 B_2 + B_1^2)}{\lambda^2} \left\| y \right\|_{\mathcal{H}}^2 W_1(\eta, \eta').$$

Moreover, we showed in (F.2) resp. in Prop. F.2 that

$$\left\| \nabla J'_\lambda[\eta] \right\| \leq \frac{B_0 B_1}{\lambda} \left\| y \right\|_{\mathcal{H}}^2 \quad \text{and} \quad \left\| \nabla J'_\lambda[\eta] - \nabla J'_\lambda[\eta'] \right\| \leq \frac{4 B_0^2 B_1^2}{\lambda^2} \left\| y \right\|_{\mathcal{H}}^2 W_1(\eta, \eta'),$$

so

$$\left| \frac{\beta}{4} \left\| \nabla J'_\lambda \eta \right\|^2 - \frac{\beta}{4} \left\| \nabla J'_\lambda[\eta'] \right\|^2 \right| \leq \frac{\beta}{4} \cdot 2 \frac{B_0 B_1 \left\| y \right\|_{\mathcal{H}}^2}{\lambda} \cdot \frac{4 B_0^2 B_1^2 \left\| y \right\|_{\mathcal{H}}^2}{\lambda^2} W_1(\eta, \eta')$$

$$= \beta \frac{2 B_0^3 B_1^3 \left\| y \right\|_{\mathcal{H}}^4}{\lambda^3} W_1(\eta, \eta'),$$

which implies the first inequality of the lemma by triangle inequality. $\qquad \square$

We can now proceed to the proof of Thm. 5.2, thanks to a bound on $W_2(\eta_{\lambda, \beta}, \delta_v)$ under Assumption 3 proved in the next section.

*Proof of Thm. 5.2.* For concision, in this proof, we will use the notations $O(\cdot), \Omega(\cdot), \Theta(\cdot), \lesssim$ to hide constants dependent only on $\left\| \varphi \right\|_{L^2(\rho)}, \left\| \varphi' \right\|_{L^4(\rho)}, \left\| \varphi'' \right\|_{L^4(\rho)}, \mathbb{E}_{x \sim \rho} \left\| x \right\|^4 / d^2, c_1, C_1, C_2, C_3$ and $C_4$.

We established in Prop. F.7 that $f_0 := J'_\lambda[\delta_v]$ satisfies the conditions $(L_{\mathbb{S}^d})$ $(L_U)$ $(C_{\mathbb{S}^d})$ $(C_U)$ of Thm. F.6 with some constants $D_i, r = O(1)$ (in fact only dependent on $c_1, C_1, C_2, C_3$) provided that $\beta \geq D_0 d \lambda^{-1}$. Thus, the first part of the theorem concerning the LSI of $\widehat{\delta}_v \propto e^{-\beta J'_\lambda[\delta_v]} \tau$, follows from Thm. F.6. To prove the second part of the theorem, it suffices to show that $f^* := J'_\lambda[\eta_{\lambda, \beta}]$ satisfies the conditions $(L'_{\mathbb{S}^d})$ $(L_U)$ $(C_{\mathbb{S}^d})$ $(C_U)$ of Thm. F.6 with some constants $\widetilde{D}'_0, \widetilde{D}'_1, \widetilde{D}_2, \widetilde{D}_3, \widetilde{D}_4, r = \Theta(1)$.

By Lem. F.3, there exist constants $B_i = O(1)$ such that $\sup_w \left\| \nabla^i \phi(w) \right\|_{\mathcal{H}} \leq B_i$, for $i \in \{0, 1, 2\}$. Moreover, by Lem. F.12 below, provided that $\beta \geq \Omega(d\lambda)$, one has

$$W_2(\eta_{\lambda, \beta, \delta_v}) \lesssim \sqrt{\beta^{-1} d \lambda^{-1} \cdot \log(\beta d^{-1} \lambda^{-1})} \ =: \overline{W}.$$

Now by the conditions $(L_{\mathbb{S}^d})$ $(L_U)$ $(C_{\mathbb{S}^d})$ $(C_U)$ for $f = f_0$ and $D_i = \Theta(1)$ (by Prop. F.7), from Lem. F.9 along with the triangle inequality we have

$$\forall w \in \mathbb{S}^d, \ \frac{1}{2} \Delta f^* - \frac{\beta}{4} \left\| \nabla f^* \right\|^2 \lesssim \lambda d + (d\lambda^{-2} + \beta \lambda^{-3}) \overline{W}$$

$$\forall w \in \mathbb{S}^d \setminus U, \ \frac{1}{2} \Delta f^* - \frac{\beta}{4} \left\| \nabla f^* \right\|^2 \leq -D_2 \beta \lambda^2 + E_2 \cdot (d\lambda^{-2} + \beta \lambda^{-3}) \overline{W}$$

$$\forall w \in \mathbb{S}^d, \ \lambda_{\min}(\nabla^2 f^*(w)) \gtrsim -\lambda - \lambda^{-2} \overline{W}$$

$$\forall w \in U, \ \lambda_{\min}(\nabla^2 f^*(w)) \geq D_4 \lambda - E_4 \cdot \lambda^{-2} \overline{W}$$

for some constants $E_2, E_4 = O(1)$. So,

- ($\text{L}'_{\mathbb{S}^d}$) for $f^*$ can be ensured with $\widetilde{D}'_1 = O(1)$ provided that $(d\lambda^{-2} + \beta\lambda^{-3})\overline{W} = (\beta^{-1}d\lambda + 1)\beta\lambda^{-3}\overline{W} = O(\lambda d\beta^{3/4})$. Since we already assume that $\beta \geq \Omega(d\lambda)$, this is equivalent to $\beta\lambda^{-3}\overline{W} = O(\lambda d\beta^{3/4})$, i.e., $\beta^{1/4}\lambda^{-4}d^{-1}\overline{W} = O(1)$.

- ($\text{L}_U$) can be ensured with $\widetilde{D}_2 = \frac{D_2}{2}$ if $\beta$ is such that $E_2(d\lambda^{-2} + \beta\lambda^{-3})\overline{W} \leq \frac{D_2}{2}\beta\lambda^2$, i.e., $(\beta^{-1}d\lambda + 1)\lambda^{-5}\overline{W} \leq \frac{D_2}{2E_2}$. Since we already assume that $\beta \geq \Omega(d\lambda)$, this is equivalent to $\lambda^{-5}\overline{W} \leq F_2$ for a certain $F_2 = \Theta(1)$.

- ($\text{C}_{\mathbb{S}^d}$) can be ensured with $\widetilde{D}_3 = O(1)$ provided that $\lambda^{-2}\overline{W} = O(\lambda)$, i.e., $\lambda^{-3}\overline{W} = O(1)$.

- ($\text{C}_U$) can be ensured with $\widetilde{D}_4 = \frac{D_4}{4}$ if $E_4\lambda^{-2}\overline{W} \leq \frac{D_4}{2}\lambda$, i.e., $\lambda^{-3}\overline{W} \leq \frac{D_4}{2E_4} =: F_4 = \Theta(1)$.

In summary, since we assume $\lambda \leq 1$, we have $\lambda^{-3} \leq \lambda^{-5}$ and $\lambda^{-4}d^{-1} \leq \lambda^{-5}$. Hence we will choose $\beta$ such that $\beta^{1/4}d^{-1}\lambda^{-4}\overline{\overline{W}} = O(1)$ and $\lambda^{-5}\overline{W} \leq F_2$ for a certain $F_2 = \Theta(1)$, and this will ensure all four conditions with constants $\widetilde{D}'_1, \widetilde{D}_2, \widetilde{D}_3, \widetilde{D}_4 = \Theta(1)$. For choices of $\beta$ such that $\beta \geq d^4\lambda^{-4}$, it suffices to have $\beta^{1/4}d^{-1}\lambda^{-4}\overline{W} \leq F_2$. Now substituting the definition of $\overline{W}$, this sufficient condition rewrites

$$\beta^{1/4}d^{-1}\lambda^{-4}\overline{W} \leq F_2 \iff \beta^{1/2}d^{-2}\lambda^{-8} \cdot \beta^{-1}d\lambda^{-1}\log\left(\frac{\beta}{d\lambda}\right) = \beta^{-1/2}\lambda^{-9}d^{-1}\log\left(\frac{\beta}{d\lambda}\right) \leq F_2^2.$$

Since $\forall \varepsilon, x > 0, \varepsilon\log x = \log x^\varepsilon \leq x^\varepsilon$, then for any $\varepsilon > 0$ it suffices to choose $\beta$ such that

$$\beta^{-1/2}\lambda^{-9}d^{-1}\left(\frac{\beta}{d\lambda}\right)^\varepsilon \leq \varepsilon F_2^2 \iff \beta^{1/2-\varepsilon} \geq \varepsilon^{-1}F_2^{-2}\lambda^{-9-\varepsilon}d^{-1-\varepsilon}.$$

Choosing e.g. $\varepsilon = \frac{1}{4}$, we get that a sufficient condition is $\beta \geq \Omega(\text{poly}(\lambda^{-1}, d))$.

Hence we may apply the second part of Thm. F.6 to $f^* = J'_\lambda[\eta_{\lambda,\beta}]$ with constants $\widetilde{D}'_1, \widetilde{D}_2, \widetilde{D}_3, \widetilde{D}_4 = O(1)$, provided that $\beta \geq \Omega(\text{poly}(\lambda^{-1}, d))$. This concludes the proof of the second part of the theorem. $\qquad\square$

### F.2.3 Bound on $W_1(\eta_{\lambda,\beta}, \delta_v)$

The following lemma shows a form of weak coercivity of $J_\lambda$.

**Lemma F.10.** *Under Assumptions 2 and 3, if furthermore there exist $c_1, C_1, C_3, C_4 > 0$ such that*

$$\forall r \in [-1, +1], \quad c_1 \leq g'(r) \leq C_1, \quad \left|g''(r)(1-r^2)^{1/2}\right| \leq C_3, \quad \left|g'''(r)(1-r^2)^{3/2}\right| \leq C_4,$$

*then there exists a constant $\alpha_g$ dependent only on $c_1, C_1, C_3, C_4$ such that*

$$\forall \eta, \ J_\lambda(\eta) - J_\lambda(\delta_v) \geq \lambda\alpha_g W_2^2(\eta, \delta_v).$$

*Proof.* Since $J_\lambda$ is convex,

$$J_\lambda(\eta) - J_\lambda(\delta_v) \geq \int_{\mathbb{S}^d} J'_\lambda[\delta_v]d(\eta - \delta_v) = -\lambda\int_{\mathbb{S}^d} g(\langle v, w \rangle)d(\eta - \delta_v)(w)$$

$$= \lambda\int_{\mathbb{S}^d}[g(1) - g(\langle v, w \rangle)]\,d\eta(w).$$

Now let $U_r = \left\{w \in \mathbb{S}^d; \text{dist}_{\mathbb{S}^d}(w, v) \leq r\right\}$ for some $r > 0$ to be chosen. We will compute the integral separately on $U_r$ and on $\mathbb{S}^d \setminus U_r$.

For the part $\int_{U_r}$, we proceed by a second-order Taylor expansion. Namely, for any $w \in U_r \setminus \{v\}$, let $e \perp v$ such that $w = \cos(\theta)v + \sin(\theta)e$ for some $0 < \theta \leq r$, since $\text{dist}_{\mathbb{S}^d}(w, v) = \arccos(\langle w, v \rangle) = \theta$. Then $g(\langle v, w \rangle) = g(\cos\theta)$, and

$$\frac{d}{d\theta}g(\cos\theta) = -\sin(\theta)g'(\cos\theta)$$

$$\frac{d^2}{d\theta^2}g(\cos\theta) = \sin(\theta)^2 g''(\cos\theta) - \cos(\theta)g'(\cos\theta)$$

$$\frac{d^3}{d\theta^3}g(\cos\theta) = -\sin(\theta)^3 g'''(\cos\theta) + 3\sin(\theta)\cos(\theta)g''(\cos\theta) + \sin(\theta)g'(\cos\theta).$$

Notice that by our assumptions on $g$, it is smooth enough at 1 so that $\sin(\theta)g'(\cos\theta) \to 0$ and $\sin(\theta)^2 g''(\cos\theta) \to 0$ as $\theta \to 0$. Further,

$$\sup_\theta \frac{d^3}{d\theta^3} g(\cos\theta) \leq C_4 + 3C_3 + C_1 =: 6M_{3,g}.$$

Consequently, by a univariate Taylor expansion with remainder in Langrange form around $\theta = 0$, for all $0 < \theta \leq r$, provided that we choose $r \leq \frac{g'(1)}{2M_{3,g}}$, we have

$$g(\cos\theta) = g(1) + 0 + \frac{1}{2}(0 - g'(1))\theta^2 + \frac{1}{6}(g \circ \cos)^{(3)}(u)\theta^3 \quad \text{for some } u \in [0, r]$$

$$\leq g(1) - \frac{1}{2}g'(1)\theta^2 + \frac{1}{6}\left[\sup_{[0,r]}(g\circ\cos)^{(3)}\right]\theta^3$$

$$\leq g(1) - \frac{1}{2}g'(1)\theta^2 + M_{3,g}\theta^3 = g(1) - \left(\frac{1}{2}g'(1) - M_{3,g}\theta\right)\theta^2$$

$$\leq g(1) - \frac{1}{4}g'(1)\theta^2. \tag{F.7}$$

In other words,

$$\forall w \in U_r, \quad g(1) - g(\langle v, w\rangle) \geq \frac{1}{4}g'(1)\operatorname{dist}_{\mathbb{S}^d}(w, v)^2,$$

$$\text{and so,} \quad \int_{U_r} [g(1) - g(\langle v, w\rangle)]\, d\eta(w) \geq \frac{1}{4}g'(1)\int_{U_r} \operatorname{dist}_{\mathbb{S}^d}(w, v)^2\, d\eta(w).$$

For the part $\int_{\mathbb{S}^d \setminus U_r}$, since $g$ is increasing on $[-1, 1]$ since $g' \geq c_1 > 0$, we have

$$\int_{\mathbb{S}^d \setminus U_r} [g(1) - g(\langle v, w\rangle)]\, d\eta(w) \geq [g(1) - g(\cos(r))]\,[1 - \eta(U_r)]$$

$$\geq \left[\frac{1}{4}g'(1)r^2\right][1 - \eta(U_r)]$$

where the second inequality follows from the Taylor expansion (F.7) above applied to $\theta = r$. Thus we showed

$$J_\lambda(\eta) - J_\lambda(\delta_v) \geq \lambda\left\{\left[\frac{1}{4}g'(1)r^2\right][1 - \eta(U_r)] + \frac{g'(1)}{4}\int_{U_r}\operatorname{dist}_{\mathbb{S}^d}(w, v)^2 d\eta(w)\right\}$$

$$= \frac{\lambda g'(1)}{4}\left\{r^2[1 - \eta(U_r)] + \int_{U_r}\operatorname{dist}_{\mathbb{S}^d}(w, v)^2 d\eta(w)\right\}.$$

On the other hand, since $\operatorname{dist}_{\mathbb{S}^d}(v, w) = \arccos(\langle v, w\rangle)$,

$$W_2^2(\eta, \delta_v) = \int_{\mathbb{S}^d \setminus U_r}\operatorname{dist}_{\mathbb{S}^d}(v, w)^2 d\eta(w) + \int_{U_r}\operatorname{dist}_{\mathbb{S}^d}(v, w)^2 d\eta(w)$$

$$\leq \pi^2[1 - \eta(U_r)] + \int_{U_r}\operatorname{dist}_{\mathbb{S}^d}(v, w)^2 d\eta(w).$$

Hence

$$J_\lambda(\eta) - J_\lambda(\delta_v) \geq \frac{\lambda g'(1)}{4} \cdot \sup_{0 \leq r \leq \frac{g'(1)}{2M_{3,g}}} \min\left[\frac{r^2}{\pi^2}, 1\right] W_2^2(\eta, \delta_v)$$

$$= \lambda \cdot \frac{g'(1)}{4}\min\left[\left(\frac{g'(1)}{2M_{3,g}}\right)^2/\pi^2, 1\right] \cdot W_2^2(\eta, \delta_v)$$

$$\geq \lambda \cdot \frac{c_1}{4}\min\left[\left(\frac{c_1}{2M_{3,g}}\right)^2/\pi^2, 1\right] \cdot W_2^2(\eta, \delta_v) =: \lambda\alpha_g W_2^2(\eta, \delta_v).$$

Notice that $\alpha_g$ only depends on $c_1, C_1, C_3, C_4$. $\qquad\square$

We will use the following fact about the surface area of a small hyperspherical cap around a pole for bounding $W_1(\eta_{\lambda,\beta}, \delta_v)$. It essentially shows that, for $\mathcal{W} = \mathbb{S}^d$, the constant called $C$ in the statement of Lem. E.1 scales with dimension as $2^{-d} \lesssim C^{-1} \lesssim 1/\sqrt{d}$.

**Lemma F.11.** *Fix $d \geq 2$ and $v \in \mathbb{S}^d$ and denote by $\tau$ the uniform measure on $\mathbb{S}^d$. For any $\epsilon > 0$, let $S_\epsilon = \{w \in \mathbb{S}^d : \mathrm{dist}_{\mathbb{S}^d}(w, v) \leq \epsilon\}$. There exist universal constants $C_-, C_+ > 0$ such that*

$$\forall 0 < \epsilon \leq \frac{\pi}{4}, \quad C_-^{-1} (\epsilon/2)^d \leq \tau(S_\epsilon) \leq C_+ \, \epsilon^d / \sqrt{d}.$$

*Proof.* For $w \sim \tau$, the distribution of $\langle w, v \rangle$ admits a probability density function $h(z) = (1 - z^2)^{d/2-1}/Z$, where

$$Z = \int_{-1}^1 (1 - z^2)^{d/2-1} \mathrm{d}z = B\left(\frac{d}{2}, \frac{1}{2}\right) = \frac{\Gamma\left(\frac{d}{2}\right)\sqrt{\pi}}{\Gamma\left(\frac{d+1}{2}\right)}.$$

Note that by Gautschi's inequality $\forall s \in (0, 1), \forall x > 0, \; x^{1-s} < \frac{\Gamma(x+1)}{\Gamma(x+s)} < (x+1)^{1-s}$ applied to $s = \frac{1}{2}$ and $x = \frac{d-1}{2}$, we have $\sqrt{\frac{d-1}{2}} < \frac{\Gamma\left(\frac{d+1}{2}\right)}{\Gamma\left(\frac{d}{2}\right)} < \sqrt{\frac{d+1}{2}}$, so

$$\sqrt{\frac{2\pi}{d+1}} \leq Z \leq \sqrt{\frac{2\pi}{d-1}}.$$

By definition, since $\mathrm{dist}_{\mathbb{S}^d}(w, v) = \arccos(\langle w, v \rangle)$, $\tau(S_\epsilon) = \int_{\cos(\epsilon)}^1 h(z)\mathrm{d}z$. One can verify

$$\forall 0 < \epsilon \leq \frac{\pi}{4}, \; \sqrt{1 - \epsilon^2} \leq \cos(\epsilon) \leq \sqrt{1 - \frac{\epsilon^2}{4}}.$$

So for all $0 < \epsilon \leq \frac{\pi}{4}$,

$$\tau(S_\epsilon) = \int_{\cos(\epsilon)}^1 h(z)\mathrm{d}z \leq \int_{\sqrt{1-\epsilon^2}}^1 h(z)\mathrm{d}z$$

$$= Z^{-1} \int_{\sqrt{1-\epsilon^2}}^1 (1-z^2)^{d/2-1}\mathrm{d}z = Z^{-1} \int_{1-\epsilon^2}^1 (1-t)^{d/2-1} \frac{\mathrm{d}t}{2\sqrt{t}}$$

$$\leq Z^{-1} \frac{1}{2\sqrt{1-\epsilon^2}} \int_{1-\epsilon^2}^1 (1-t)^{d/2-1}\mathrm{d}t = Z^{-1} \frac{1}{2\sqrt{1-\epsilon^2}} \int_0^{\epsilon^2} t^{d/2-1}\mathrm{d}t$$

$$= Z^{-1} \frac{1}{2\sqrt{1-\epsilon^2}} \cdot \frac{2}{d}[\epsilon^2]^{d/2} \leq Z^{-1} \frac{1}{d\sqrt{1-(\pi/4)^2}}\epsilon^d \; \leq C_+ \epsilon^d/\sqrt{d}$$

for some universal constant $C_+$. In the other direction,

$$\tau(S_\epsilon) \geq \int_{\sqrt{1-\epsilon^2/4}}^1 h(z)\mathrm{d}z = Z^{-1} \int_{\sqrt{1-\epsilon^2/4}}^1 (1-z^2)^{d/2-1}\mathrm{d}z = Z^{-1} \int_{1-\epsilon^2/4}^1 (1-t)^{d/2-1}\frac{\mathrm{d}t}{2\sqrt{t}}$$

$$\geq Z^{-1} \frac{1}{2} \int_{1-\epsilon^2/4}^1 (1-t)^{d/2-1}\mathrm{d}t = Z^{-1} \frac{1}{2} \int_0^{\epsilon^2/4} t^{d/2-1}\mathrm{d}t$$

$$= Z^{-1} \frac{1}{2} \frac{2}{d}[\epsilon^2/4]^{d/2} = Z^{-1} \frac{1}{d}(\epsilon/2)^d \; \geq c(\epsilon/2)^d/\sqrt{d}.$$

for some universal constants $c$. By repeating the same argument with $\sqrt{1 - \frac{\epsilon^2}{4}}$ replaced by $\sqrt{1 - \frac{\epsilon^2}{3.9}}$, we get $\tau(S_\epsilon) \geq c'(\epsilon/1.99)^d/\sqrt{d} \geq C_-^{-1}(\epsilon/2)^d$ for some universal constants $c', C_-$. $\square$

The following lemma combines the weak coercivity and weak Lipschitz-continuity of $J_\lambda$ by a $\Gamma$-convergence type argument, to show an explicit bound on $W_1(\eta_{\lambda,\beta}, \delta_v)$. It quantifies the intuitive fact that $\eta_{\lambda,\beta}$ converges weakly to $\delta_v$ when $\beta^{-1} \to 0$ or $\lambda \to +\infty$.

**Lemma F.12.** *Under Assumptions [2] and [3], if $\sup_w \left\| \nabla^i \phi(w) \right\|_{\mathcal{H}} \leq B_i < \infty$ for $i \in \{0, 1\}$, and if $\beta \geq \frac{4d\lambda}{\pi} \left( B_0 B_1 \left\| y \right\|_{\mathcal{H}}^2 \right)^{-1}$, then*

$$W_2(\eta_{\lambda,\beta}, \delta_v) \leq \sqrt{\frac{1}{\alpha_g} \frac{\beta^{-1} d}{\lambda} \left( \widetilde{C} + \log \left( B_0 B_1 \left\| y \right\|_{\mathcal{H}}^2 \right) - \log \left( \beta^{-1} d\lambda \right) \right)}$$

*where $\widetilde{C}$ is a universal constant and $\alpha_g$ is the constant from [Lem. F.10].*

*Proof.* Since $\eta_{\lambda,\beta} = \arg\min J_{\lambda,\beta}$ and $J_{\lambda,\beta} = J + \beta^{-1} H\left( \cdot | \tau \right)$, then for any $\eta^\sigma \in \mathcal{P}(\mathcal{W})$,

$$J_\lambda(\eta_{\lambda,\beta}) \leq J_\lambda(\eta_{\lambda,\beta}) + \beta^{-1} H\left(\eta_{\lambda,\beta} | \tau\right) = J_{\lambda,\beta}(\eta_{\lambda,\beta}) \leq J_{\lambda,\beta}(\eta^\sigma) = J_\lambda(\eta^\sigma) + \beta^{-1} H\left(\eta^\sigma | \tau\right).$$

Further, we showed in [Lem. F.10] that $\forall \eta, \; J_\lambda(\eta) - J_\lambda(\delta_v) \geq \lambda \alpha_g \cdot W_2^2(\eta, \delta_v)$, so

$$\lambda \alpha_g \cdot W_2^2(\eta_{\lambda,\beta}, \delta_v) \leq J_\lambda(\eta_{\lambda,\beta}) - J_\lambda(\delta_v) \leq J_\lambda(\eta^\sigma) - J_\lambda(\delta_v) + \beta^{-1} H\left(\eta^\sigma | \tau\right).$$

It remains to upper-bound the right-hand side, which we do by choosing as $\eta^\sigma$ a box-kernel smoothed version of $\delta_v$ (this part the proof is essentially an instantiation of [Lem. E.1]). Specifically, let $\eta^\sigma$ be the uniform measure over the spherical cap $S_\sigma = \left\{ w \in \mathbb{S}^d; \operatorname{dist}_{\mathbb{S}^d}(w, v) \leq \sigma \right\}$ for $\sigma$ to be chosen. We showed in [Prop. F.2] that

$$J_\lambda(\eta^\sigma) - J_\lambda(\delta_v) \leq \frac{B_0 B_1 \left\| y \right\|_{\mathcal{H}}^2}{\lambda} \cdot W_1(\eta^\sigma, \delta_v)$$

where $\sup_w \left\| \nabla^i \phi(w) \right\|_{\mathcal{H}} \leq B_i$, and by definition

$$W_1(\eta^\sigma, \delta_v) = \int \operatorname{dist}_{\mathbb{S}^d}(w, v) \, d\eta^\sigma(w) = \frac{1}{\operatorname{vol}(S_\sigma)} \int_{S_\sigma} \operatorname{dist}_{\mathbb{S}^d}(w, v) \, d\operatorname{vol}(w) \leq \sigma.$$

Moreover by [Lem. F.11], provided that $0 < \sigma \leq \frac{\pi}{4}$,

$$H\left(\eta^\sigma | \tau\right) = \int d\eta_\sigma \log \frac{d\eta_\sigma}{d\tau} = \log \frac{\operatorname{vol}(\mathbb{S}^d)}{\operatorname{vol}(S_\sigma)} = -\log \tau(S_\sigma) \leq \log C - d \log \frac{\sigma}{2}$$

for some universal constant $C$, and let us assume w.l.o.g. that $C > 1$, so that $\log C \leq d \log C$. Thus

$$J_\lambda(\eta^\sigma) - J_\lambda(\delta_v) + \beta^{-1} H\left(\eta^\sigma | \tau\right) \leq \frac{B_0 B_1 \left\| y \right\|_{\mathcal{H}}^2}{\lambda} \sigma - \beta^{-1} d \log \sigma + \beta^{-1} d \log 2C.$$

Therefore, taking the infimum over $0 < \sigma \leq \frac{\pi}{4}$,

$$\begin{aligned}
\lambda \alpha_g \cdot W_2^2(\eta_{\lambda,\beta}, \delta_v) &\leq \inf_{0 < \sigma \leq \frac{\pi}{4}} \frac{B_0 B_1 \left\| y \right\|_{\mathcal{H}}^2}{\lambda} \sigma - \beta^{-1} d \log \sigma + \beta^{-1} d \log 2C \\
&= \beta^{-1} d - \beta^{-1} d \log \frac{\beta^{-1} d\lambda}{B_0 B_1 \left\| y \right\|_{\mathcal{H}}^2} + \beta^{-1} d \log 2C \\
&= \beta^{-1} d \left( 1 + \log(2C) - \log(\beta^{-1} d\lambda) + \log \left( B_0 B_1 \left\| y \right\|_{\mathcal{H}}^2 \right) \right),
\end{aligned}$$

where on the second line we used that the unconstrained infimum of the right-hand side over $\sigma > 0$ is attained at $\sigma = \frac{\beta^{-1} d\lambda}{B_0 B_1 \left\| y \right\|_{\mathcal{H}}^2}$, which is indeed less than $\frac{\pi}{4}$ by assumption. This shows the bound

$$W_2(\eta_{\lambda,\beta}, \delta_v) \leq \sqrt{\frac{1}{\lambda \alpha_g} \beta^{-1} d \left( 1 + \log(2C) - \log(\beta^{-1} d\lambda) + \log \left( B_0 B_1 \left\| y \right\|_{\mathcal{H}}^2 \right) \right)},$$

and the bound announced in the proposition follows by gathering some universal constants into $\widetilde{C}$. $\qquad\square$

## F.3 Proof of Prop. 5.3 (examples of activations satisfying the assumptions)

Before presenting the proof, we recall a few concepts from the theory of spherical harmonics, and refer to [AH12; FE12] for more details. Let $\tau$ be the uniform probability measure on $\mathbb{S}^d$. The spherical harmonics in dimension $d+1$ form an orthonormal basis of $L^2(\tau)$. We denote them by $\{Y_{kj}\}_{k,j}$, where $k \geq 0$ and $1 \leq j \leq N(d,k)$, where $N(d,0) = 1$ and $N(d,k) = \frac{2k+d-1}{k}\binom{k+d-2}{d-1}$ for $k \geq 1$ (for $k = 0$ we have $Y_{01} = 1$). Consequently, any $\phi \in L^2(\tau)$ can be written as

$$\phi = \sum_{k=0}^{\infty} \sum_{j=1}^{N(d,k)} \langle \phi, Y_{kj}\rangle_{L^2(\tau)} Y_{kj}.$$

Let $P_{k,d}$ be the Legendre polynomial (a.k.a. Gegenbauer polynomial) of degree $k$ in dimension $d+1$, normalized such that $P_{k,d}(1) = 1$. Thanks to Rodrigues' formula [AH12, Theorem 2.23], we can express Legendre polynomials as,

$$P_{k,d}(t) = \frac{(-1)^k \Gamma(d/2)}{2^k \Gamma(k+d/2)}(1-t^2)^{(2-d)/2}\left(\frac{\mathrm{d}}{\mathrm{d}t}\right)^k(1-t^2)^{k+(d-2)/2}.$$

We now go over some useful properties of spherical harmonics and Legendre polynomials.

- **(Addition Formula)** We have the following formula which relates Legendre polynomials to spherical harmonics [AH12, Theorem 2.9],

$$\sum_{j=1}^{N(d,k)} Y_{kj}(w)Y_{kj}(v) = N(d,k)P_{k,d}(\langle w,v\rangle), \quad \forall w,v \in \mathbb{S}^d.$$

- **(Hecke-Funk Formula)** Suppose $\phi \in L^2(\tau)$ is given by $\phi(\cdot) = \varphi(\langle w, \cdot\rangle)$ for some $w \in \mathbb{S}^d$. Then [AH12, Theorem 2.22],

$$\langle \phi, Y_{kj}\rangle_{L^2(\tau)} = \frac{\Gamma((d+1)/2)}{\Gamma(d/2)\sqrt{\pi}}Y_{kj}(w)\int_{-1}^{1}\varphi(t)P_k(t)(1-t^2)^{(d-2)/2}\mathrm{d}t.$$

- **(Orthogonality of Legendre Polynomials)** Using the addition formula and orthonormality of spherical harmonics, for every $k, k' \geq 0$ we have,

$$\langle P_{k,d}(\langle w,\cdot\rangle), P_{k',d}(\langle v,\cdot\rangle)\rangle_{L^2(\tau)} = \frac{\delta_{kk'}P_{k,d}(\langle w,v\rangle)}{N(d,k)}.$$

- **(Derivative of Legendre Polynomials)** For every $k \geq j$, we have the following identity for derivatives of Legendre polynomials [AH12, Equation (2.89)],

$$P_{k,d}^{(j)}(t) = c_{j,k,d}P_{k-j,d+2j}(t),$$

where $P_{k,d}^{(j)}$ denotes the $j$th derivative of $P_{k,d}$, and

$$c_{j,k,d} = \frac{k(k-1)\ldots(k-j+1)(k+d-1)(k+d)\ldots(k+d+j-2)}{d(d+2)\ldots(d+2j-2)}. \tag{F.8}$$

Notice that for $j > k$ we have $P_{k,d}^{(j)} = 0$.

We use the tools introduced above to prove the following lemma.

**Lemma F.13.** *Suppose $\rho$ is a spherically symmetric probability measure on $\mathbb{R}^{d+1}$. Define $q : [-1,1] \to \mathbb{R}$ via $q(\langle w,v\rangle) = \int \varphi(\langle w,x\rangle)\varphi(\langle v,x\rangle)\mathrm{d}\rho(x)$ for $w, v \in \mathbb{S}^d$. Then, for every $j \geq 1$,*

$$q^{(j)}(\langle w,v\rangle) = \frac{1}{(d+1)(d+3)\ldots(d+2j-1)}\int \|x\|^{2j}\varphi^{(j)}(\langle w,x\rangle)\varphi^{(j)}(\langle v,x\rangle)\mathrm{d}\rho(x),$$

*where $\varphi^{(j)}$ denotes the $j$th derivative of $\varphi$.*

*Proof.* We being by introducing the notation $\varphi_r(\langle w, x \rangle) = \varphi(r \langle w, x \rangle)$. Doing so allows us to only consider functions on $\mathbb{S}^d$ by conditioning on the norm of input $\|x\|$. Notice that

$$q(\langle w, v \rangle) = \mathbb{E}\left[\mathbb{E}\left[\varphi\left(\|x\| \left\langle w, \frac{x}{\|x\|}\right\rangle\right)\varphi\left(\|x\| \left\langle v, \frac{x}{\|x\|}\right\rangle\right) \mid \|x\|\right]\right] = \mathbb{E}_{\|x\|}\left[q_{\|x\|}(\langle w, v \rangle)\right], \quad \text{(F.9)}$$

where

$$q_r(\langle w, v \rangle) := \int \varphi(r\langle w, x \rangle)\varphi(r\langle v, x \rangle)\mathrm{d}\tau(x) = \langle \varphi_r(\langle w, \cdot \rangle), \varphi_r(\langle v, \cdot \rangle)\rangle_{L^2(\tau)}.$$

By the Hecke-Funk formula,

$$\langle \varphi_r(\langle w, \cdot \rangle), Y_{kj}(\cdot)\rangle_{L^2(\tau)} = \bar{\alpha}_{k,r} Y_{kj}(w) := \frac{\alpha_{k,r}}{\sqrt{N(d,k)}} Y_{kj}(w),$$

where

$$\bar{\alpha}_{k,r} := \frac{\Gamma((d+1)/2)}{\Gamma(d/2)\sqrt{\pi}} \int_{-1}^{1} \varphi(rt)P_k(t)(1-t^2)^{(d-2)/2}\mathrm{d}t.$$

Then, by the expansion of $\varphi_r(\langle w, \cdot \rangle)$ in the basis of spherical harmonics,

$$\varphi_r(\langle w, \cdot \rangle) = \sum_{k=0}^{\infty} \sum_{j=1}^{N(d,k)} \frac{\alpha_{k,r}}{\sqrt{N(d,k)}} Y_{kj}(w)Y_{kj}(\cdot) = \sum_{k=0}^{\infty} \sqrt{N(d,k)}\alpha_{k,r}P_{k,d}(\langle w, \cdot \rangle). \quad \text{(F.10)}$$

Via the formula for inner products of Legendre polynomials, we obtain

$$q_r(\langle w, v \rangle) = \sum_{k=0}^{\infty} \alpha_{k,r}^2 N(d,k)\langle P_{k,d}(\langle w, \cdot \rangle), P_{k,d}(\langle v, \cdot \rangle)\rangle_{L^2(\tau)} = \sum_{k=0}^{\infty} \alpha_{k,r}^2 P_{k,d}(\langle w, v \rangle).$$

As a result,

$$q_r^{(j)}(\langle w, v \rangle) = \sum_{k=0}^{\infty} \alpha_{k,r}^2 P_{k,d}^{(j)}(\langle w, v \rangle) = \sum_{k=j}^{\infty} \alpha_{k,r}^2 c_{j,k,d}P_{k-j,d+2j}(\langle w, v \rangle), \quad \text{(F.11)}$$

where $c_{j,k,d}$ is given by (F.8). On the other hand, we can directly obtain from (F.10),

$$\varphi_r^{(j)}(\langle w, x \rangle) = \sum_{k=0}^{\infty} \sqrt{N(d,k)}\alpha_{k,r}P_{k,d}^{(j)}(\langle w, x \rangle) = \sum_{k=j}^{\infty} \sqrt{N(d,k)}\alpha_{k,r}c_{j,k,d}P_{k-j,d+2j}(\langle w, x \rangle).$$

Therefore,

$$\langle \varphi_r^{(j)}(\langle w, \cdot \rangle), \varphi_r^{(j)}(\langle v, \cdot \rangle)\rangle_{L^2(\tau)} = \sum_{k=j}^{\infty} \frac{\alpha_{k,r}^2 c_{j,k,d}^2 N(d,k)}{N(d+2j,k-j)} P_{k-j,d+2j}(\langle w, v \rangle).$$

Moreover, it is straightforward to verify that

$$\frac{c_{j,k,d}N(d,k)}{N(d+2j,k-j)} = (d+1)(d+3)\dots(d+2j-1)$$

for $k \geq j$. Therefore,

$$\langle \varphi_r^{(j)}(\langle w, \cdot \rangle), \varphi_r^{(j)}(\langle v, \cdot \rangle)\rangle_{L^2(\tau)} = (d+1)(d+3)\dots(d+2j-1)\sum_{k=j}^{\infty} \alpha_{k,r}^2 c_{j,k,d}P_{k-j,d+2j}(\langle w, v \rangle)$$

$$= (d+1)(d+3)\dots(d+2j-1)q_r^{(j)}(\langle w, v \rangle),$$

where the last identity follows from (F.11). We can now use the fact that $\varphi_r^{(j)} = r\varphi^{(j)}$, and plug the above back into (F.9) to obtain

$$q^{(j)}(\langle w, v \rangle) = \mathbb{E}_{\|x\|}\left[q_{\|x\|}^{(j)}(\langle w, v \rangle)\right] = \mathbb{E}_{\|x\|}\int \frac{\|x\|^{2j}}{(d+1)(d+3)\dots(d+2j-1)}\varphi^{(j)}(\|x\| \langle w, \bar{x} \rangle)\varphi^{(j)}(\|x\| \langle v, \bar{x} \rangle)\mathrm{d}\tau(\bar{x})$$

$$= \int \frac{\|x\|^{2j}}{(d+1)(d+3)\dots(d+2j-1)}\varphi^{(j)}(\langle w, x \rangle)\varphi^{(j)}(\langle v, x \rangle)\mathrm{d}\rho(x),$$

which concludes the proof. $\qquad\square$

We are now ready to state the proof of Prop. 5.3

*Proof of Prop. 5.3.* Recall $g(\langle w, v \rangle) = \frac{\langle \phi(w), \phi(v) \rangle_{\mathcal{H}}^2}{2(\lambda + \|\phi(v)\|_{\mathcal{H}}^2)^2}$. Let $q(\langle w, v \rangle) = \langle \phi(w), \phi(v) \rangle_{\mathcal{H}}$. Consequently,

$$g' = \frac{qq'}{(\lambda + \|\phi(v)\|_{\mathcal{H}}^2)^2}, \quad g'' = \frac{qq'' + q'^2}{(\lambda + \|\phi(v)\|_{\mathcal{H}}^2)^2}, \quad g''' = \frac{3q'q'' + qq'''}{(\lambda + \|\phi(v)\|_{\mathcal{H}}^2)^2}.$$

We proceed to bound each term separately. By non-negativity of $\phi$, for any $r > 0$, we have

$$\begin{aligned}
q(\langle w, v \rangle) &= \mathbb{E}\left[\varphi(\langle w, x \rangle)\varphi(\langle v, x \rangle)\right] \\
&\geq \mathbb{E}\left[\varphi(\langle w, x \rangle)\phi(\langle v, x \rangle)\mathbf{1}\left(|\langle w, x \rangle| \leq r, |\langle v, x \rangle| \leq r\right)\right] \\
&\geq (\inf_{|z| \leq r} \varphi(z))^2 \mathbb{P}\left[\{|\langle w, x \rangle| \leq r\} \cap \{|\langle v, x \rangle| \leq r\}\right] \\
&\geq (\inf_{|z| \leq r} \varphi(z))^2 \left(1 - \mathbb{P}[\langle w, x \rangle^2 > r^2] - \mathbb{P}[\langle v, x \rangle^2 > r^2]\right) \\
&\geq (\inf_{|z| \leq r} \varphi(z))^2 \left(1 - \frac{2\mathbb{E}[\|x\|^2]}{(d+1)r^2}\right),
\end{aligned}$$

where the last inequality follows from Markov inequality along with the fact that $\mathbb{E}[xx^\top] = \frac{\mathbb{E}[\|x\|^2]}{d+1}I_{d+1}$ for spherically symmetric distributions. Thus, by choosing $r = m = \frac{2b_2\sqrt{b_2}}{b_1}$, we have $q(z) \geq \frac{1}{2}(\inf_{|z| \leq m} \varphi(z))$. Furthermore, by the Cauchy-Schwartz inequality, $q(\langle w, v \rangle) \leq \mathbb{E}[\varphi(\langle w, x \rangle)^2] = \|\varphi\|_{L^2(\rho)}^2$. Next, we move on to bounding $q'$. Let $\bar{x} \sim \tau$ be a uniform random vector on $\mathbb{S}^d$. Then, for any $r > 0$, by Lem. F.13,

$$\begin{aligned}
q'(\langle w, v \rangle) &= \frac{1}{d+1}\mathbb{E}\left[\|x\|^2 \varphi'(\langle w, x \rangle)\varphi'(\langle v, x \rangle)\right] \\
&= \frac{1}{d+1}\mathbb{E}\left[\|x\|^2 \mathbb{E}\left[\varphi'(\langle w, x \rangle)\varphi'(\langle v, x \rangle) \mid \|x\|\right]\right] \\
&\geq \frac{(\inf_{|z| \leq r} \varphi'(z))^2}{d+1}\mathbb{E}\left[\|x\|^2 \mathbb{P}\left[\left\{|\langle w, \bar{x} \rangle| \leq \frac{r}{\|x\|}\right\} \cap \left\{|\langle v, \bar{x} \rangle| \leq \frac{r}{\|x\|}\right\} \mid \|x\|\right]\right] \\
&\geq \frac{(\inf_{|z| \leq r} \varphi'(z))^2}{d+1}\mathbb{E}\left[\|x\|^2 \left(1 - \mathbb{P}\left[\langle w, \bar{x} \rangle^2 > \frac{r^2}{\|x\|^2} \mid \|x\|\right] - \mathbb{P}\left[\langle v, \bar{x} \rangle^2 > \frac{r^2}{\|x\|^2} \mid \|x\|\right]\right)\right] \\
&\geq \frac{(\inf_{|z| \leq r} \varphi'(z))^2}{d+1}\mathbb{E}\left[\|x\|^2 \left(1 - \frac{2\|x\|^2}{r^2(d+1)}\right)\right].
\end{aligned}$$

Consequently, by choosing $r = m = \frac{2b_2\sqrt{b_2}}{b_1}$, we obtain $q' \geq \frac{b_1}{2}(\inf_{|z| \leq m} \phi'(z))^2$. Moreover, by the Cauchy-Schwartz inequality, $q' \leq b_2 \|\varphi'\|_{L^4(\rho)}^2$. As a result,

$$\frac{b_1(\inf_{|z| \leq m} \varphi(z))^2(\inf_{|z| \leq m} \varphi'(z))^2}{(\lambda + \|\varphi\|_{L^2(\rho)}^2)^2} \leq g' \leq \frac{b_2 \|\varphi\|_{L^2(\rho)}^2 \|\varphi'\|_{L^4(\rho)}^2}{(\lambda + \|\varphi\|_{L^2(\rho)}^2)^2}.$$

Furthermore, by Lem. F.13 and the Cauchy-Schwartz inequality,

$$|q''| \leq \frac{b_2^2(d+1)}{d+3}\|\varphi''\|_{L^4(\rho)}^2, \quad |q'''| \leq \frac{b_2^3(d+1)^2}{(d+3)(d+5)}\|\phi'''\|_{L^4(\rho)}^2.$$

Hence,

$$\frac{-b_2^2 \|\varphi''\|_{L^4(\rho)}^2 \|\varphi\|_{L^4(\rho)}^2}{(\lambda + \|\varphi\|_{L^2(\rho)}^2)^2} \leq g'' \leq \frac{b_2^2 \|\varphi''\|_{L^4(\rho)}^2 \|\varphi\|_{L^2(\rho)}^2 + b_2^2 \|\varphi'\|_{L^4(\rho)}^4}{(\lambda + \|\varphi\|_{L^2(\rho)}^2)^2},$$

and

$$|g'''| \leq \frac{3b_2^3 \|\phi'\|_{L^4(\rho)}^2 \|\phi''\|_{L^4(\rho)}^2 + b_2^3 \|\phi\|_{L^2(\rho)}^2 \|\phi'''\|_{L^4(\rho)}^2}{(\lambda + \|\varphi\|_{L^2(\rho)}^2)^2},$$

which completes the proof. □

### F.4 Implementation details for Fig. 1

We consider the problem (1.1) where $\mathcal{W} = \mathbb{S}^d$ and $G$ is defined as in Assumption 2, where $d = 10$, $\lambda = 10^{-3}$ and

- $y : \mathbb{R}^{d+1} \to \mathbb{R}$ is given by a teacher 2NN with 5 neurons defined as follows. The first-layer weights are orthonormal, drawn from the Haar measure, and the second layer weights are drawn i.i.d. from $\mathcal{N}(0, 1.8 I_d)$. Its activation is $\varphi_{\text{teacher}}(z) = \frac{z^4 - 6z^2 + 3}{\sqrt{24}}$, which is the normalized 4th degree Hermite polynomial.
- $\rho$ is the empirical distribution of a (covariate) dataset $(x_i)_{i \leq n}$ of $n = 100$ training samples, sampled i.i.d. from $\mathcal{N}\left( \begin{pmatrix} 0_d \\ 1 \end{pmatrix}, \begin{pmatrix} I_d & 0 \\ 0 & 0 \end{pmatrix} \right)$, with the last coordinate representing bias.
- The activation function $\varphi$ of the student 2NN $\hat{y}_\nu$ is the ReLU, $\varphi(z) = \max(0, z)$.

We performed 5 different runs, each corresponding to a different teacher network ($y$) and training dataset ($\rho$), and tested all the algorithms considered at each run. So the objective functional $G_\lambda$ is different for each run, which is why the values shown on the $y$-axis are offset by $G_\lambda^*$, the best value achieved by any of the algorithms considered for each run.

For the algorithms using the bilevel formulation, we computed the values and the Wasserstein gradients of $J_\lambda$ explicitly by the formulas from Prop. F.1 and (F.2) (the matrix $K_\eta + \lambda \, \text{id}$ in $L_\rho^2 \simeq \mathbb{R}^n$ is inverted explicitly).

For the algorithms using MFLD, we used $\beta^{-1} = 10^{-3}$. We ran the Euler-Maruyama discretization of the noisy particle gradient flow SDE described in Sec. 2 (with an inexact simulation of the Brownian increments described below), using $N = 1000$ particles – corresponding to the width of the student 2NN –, and a step size of $10^{-2}$ for (1a) and $10^{-3}$ for (1b). For Wasserstein GF without noise, we used the same discretization but with $\beta^{-1} = 0$.

Concerning the initialization of the particles $(r^i, w^i)_{i \leq N}$ – corresponding to the second resp. first-layer weights of the student network –, the $w_0^i$ are drawn i.i.d. uniformly on $\mathbb{S}^d$, and for the algorithms using the lifting formulation, the $r_0^i$ are drawn i.i.d. from $\mathcal{N}(0, 1)$.

Note that our simulations of Brownian motion are not exact. To implement MFLD on $\mathbb{S}^d$, we simply took gradient steps in $\mathbb{R}^{d+1}$ with added Gaussian noise, and projected the weights back to the sphere.

The code to reproduce this experiment can be found at https://github.com/mousavih/2024-MFLD-bilevel.

