# OpenReview forum: "Mean-Field Langevin Dynamics for Signed Measures via a Bilevel Approach"
_NeurIPS.cc/2024/Conference — NeurIPS 2024 spotlight_

### Official Review · Reviewer_Hr7F · 2024-06-25

**Soundness:** 4
**Presentation:** 3
**Contribution:** 4
**Rating:** 7
**Confidence:** 3

**Summary:**

Mean-Field Langevin Dynamics (MFLD) framework is used to solve optimization problem over manifold. The main contribution appears to be reducing a general optimization problem over signed measures to probability measures using lifting or bilevel approaches. Convergence rate of MFLD, when applied to both approaches, are investigated.

**Strengths:**

It sounds interesting to use lifting or bilevel ideas to reduce MFLD to solve general optimization problems over signed measures.

**Weaknesses:**

The scope of lifting or bilevel ideas should be made clear. For instance, lifting idea seems to be applicable when the signed measure can be represented as projections of probability measures; does the projection representation always exist?
Also, reduction of the optimization problem over probability measure may change to a more difficult optimization problem, e.g., the objective functions (3.1) and (3.3) which might be more difficult to deal with. Some discussions on how to deal with these functions empirically will be helpful.

**Questions:**

See *weakness*

---

> ### Author Rebuttal · Authors · 2024-08-05
>
> We thank the reviewer for their positive evaluation, and address their questions and comments below.
> - Thank you for pointing out the potential confusion about the scope of these ideas. In the final version, we will further clarify that the lifting and bilevel ideas are both always applicable for optimization over signed measures. In particular, any signed measure $\nu \in \mathcal{M}(\mathcal{W})$ can be represented as a projection of a probability measure $\mu(\mathrm{d} r, \mathrm{d} w) = \delta\_{\\|\nu\\|\_{TV} \frac{\mathrm{d}\nu}{\mathrm{d}|\nu|}(w)}(\mathrm{d} r) \frac{|\nu|(\mathrm{d} w)}{\\|\nu\\|\_{TV}}$.
> - The objective functions $F_{\lambda,b}$ (3.1) and $J_\lambda$ (3.3) are amenable to optimization using Wasserstein gradient flow (WGF) or MFLD, because they are defined over sets of probability measures, whereas the original optimization problem (1.1) is not. To solve the optimization problems (3.1) or (3.3), we rely on the WGF and MFLD literature which have studied the problem of optimizing convex functionals over probability measures in detail. We will further emphasize this fact in the final version.

---

### Official Review · Reviewer_3p9w · 2024-07-05

**Soundness:** 4
**Presentation:** 3
**Contribution:** 3
**Rating:** 7
**Confidence:** 3

**Summary:**

The paper studies extension of mean-field Langevin dynamics (MFLD) to perform convex optimization over the space of signed measures. The paper considers the lifting and bilevel approaches and shows that the latter guarantees better convergence properties at a wider range of hyperparameters. MFLD-bilevel is shown to be amenable to a previously studied improved temperature annealing schedule over the standard $1/\log t$ rate. Moreover when learning a single index model, MFLD-bilevel is shown to achieve a local convergence rate which depends polynomially on the dimension and inverse temperature via an improved LSI constant.

**Strengths:**

* The paper rigorously compares the lifting and bilevel approaches, and the result that the latter leads to stronger guarantees (at least in the low temperature regime) is quite surprising to me as many influential works on shallow NNs have built on the lifted formulation.
* The paper is overall well written and detailed, and the analysis utilizes novel techniques (namely local LSI bounds) to go beyond the standard results in MFLD.
* While the paper has some similarities with [Takakura, Suzuki, 2024], the setting is more general with a focus on more abstract optimization as explained in Appendix A.

**Weaknesses:**

* The parameter space $\mathcal{W}$ is assumed to be a compact manifold. Does this mean compact without boundary, which precludes subspaces of Euclidean space? Can this assumption be removed by e.g. adding a confinement term and considering relative entropy regularization with a log-concave distribution? While the non-necessity of these elements are presented as advantages in Section 1.1, studying $\mathcal{W} =\mathbb{R}^d$ is simply necessary for certain optimization problems and it would be very nice to fill out the details.

**Questions:**

* In Theorem 4.2, can the time complexity to achieve $(1+\Delta)$-accuracy be rewritten in terms of $\Delta$ and compared with the usual logarithmic annealing? The current rate expression in terms of $\delta$ makes this unclear. For example, it seems $\beta_t\approx O(1/t)$.
* Besides the established positive results, are there any theoretical/empirical results showing the difference in optimization dynamics between using squared and un-squared TV regularizer?
* What is the main issue when trying to extend the results of Section 5 to multi-index models or general target functions?

**Limitations:**

Discussed in corresponding sections.

---

> ### Author Rebuttal · Authors · 2024-08-05
>
> We thank the reviewer for their thoughtful assessment, and address their questions and comments below.
> - **On compactness**: $\mathcal{W}$ is assumed to be without boundaries. This assumption is missing on line 20. Thank you for pointing this out.
>
>     The techniques presented in our work can be used without significant modifications for dynamics over $\mathbb{R}^d$ using additional confinement. We chose to focus on compact Riemannian manifold without boundaries because it is the natural setting for 2-layer neural networks with homogeneous activations (on the sphere), or for sparse deconvolution (on the torus). The absence of additional confinement is not an advantage in general; the phrasing in Section 1.1 can indeed be misleading and will be corrected.
> - **On the statement of Theorem 4.2**: To state the time complexity of this theorem in terms of $\Delta$, we notice that $\delta = \Delta / \log(C/\Delta)$ for some constant $C$ that is polynomial in problem parameters (including $\lambda^{-1}$ and ${J^*_\lambda}^{-1}$). The time complexity would roughly read $T = \exp\big(O\big(\frac{\log(C/\Delta)}{\lambda\Delta}\big)\big)$. This can be compared with the classical logarithmic annealing procedure, which results in $T = \exp\big(O\big(\frac{1}{\lambda\Delta J^*_\lambda}\big)\big)$. Further, following the discussion on line 271, we can take $J^*_\lambda = O(\lambda)$. Thus, the annealing schedule of Theorem 4.2 leads to an improvement of order $O(\log(C/\Delta)/\lambda)$ in the exponent. Originally, we stated the time complexity of Theorem 4.2 using $\delta$ rather than $\Delta$ to simplify the presentation while having in mind that $\delta$ and $\Delta$ are equivalent up to logarithmic factors. We agree with the reviewer that the time complexity should be stated explicitly in terms of $\Delta$ for easier comparison with the baseline annealing procedure.
> - **On the difference between using squared and un-squared TV regularizer**: Concerning the lifting formulation, our negative results can be generalized to the case of un-squared TV regularizer, and we will add a remark on this in the final version. For the bilevel formulation, using a similar trick as for the squared TV norm, the unsquared TV norm can be expressed as $\Vert \mu\Vert_{TV} = \int_{\eta \in \mathcal{M}_+(X)} \frac12 \int \frac{\vert \mu\vert^2}{\eta} + \frac12 \eta(X)$, but here $\eta$ is an un-normalized measure so the resulting bilevel problem is not amenable to MFLD.
> - **On the generalization of Section 5 to multi-index models**: For Theorem 5.2, we rely on the rotational symmetry (orthogonal to the target direction) induced by the single-index model to simplify the Gibbs potential. It is an interesting open question to find examples of multi-index models with similar symmetries that lead to a polynomial LSI constant.

---

> > ### Comment · Reviewer_3p9w · 2024-08-07
> >
> > Thank you for the reply. I will maintain my high rating of the work.

---

### Official Review · Reviewer_akPi · 2024-07-09

**Soundness:** 3
**Presentation:** 3
**Contribution:** 3
**Rating:** 7
**Confidence:** 3

**Summary:**

This paper extends the well-known and recently extensively studied mean-field Langevin dynamics (MFLD) to optimization problems over signed measures (instead of probability measures). This has applications and relevance to the training of NNs or other problems in data science, such as sparse deconvolution, which are intrinsically of such form.
As an example, the learning of one neuron is presented at the end of the paper.

In order to fit the setting of signed measures into the classical probabilistic framework of the MFLD, the Authors leverage and explore two classical strategies. Namely, the lifting and bilevel reduction.
They observe that the bilevel reduction behaves more favorably compared to the lifting reduction, allowing to obtain stronger and faster convergence.

**Strengths:**

- Extending the MFLD via a bilevel approach to optimization problems over signed measures seems to be an interesting and relevant direction given that this captures several problems of interest in machine learning and data science.
- After discussing two potential directions to approach the problem (lifting and bilevel), the paper adapts and employs the convergence analysis of [Chi22b] to the new setting of the bilevel MFLD, which transpired to be the way to go due to the assumptions holding in more realistic scenarios.
- Despite the paper being of purely theoretical nature and in parts quite technical, it is overall well-written and good to follow. The organization and structure of a general introduction, revisiting the classical MFLD as well as the two reduction strategies in Section 2 and 3, respectively, helps in this regard.
- Overall, the technical contributions seem to be rigorous and convincing.

**Weaknesses:**

The experimental exploration of the proposed approach is a bit limited. The experiment in Figure 1 seems very academic.
I was wondering, if the Authors could comment on whether there are experiments already conducted in the literature that could prove the practicability of the approach. A remark on that instead of the experiments in Figure 1, which could be presented in the Appendix, would be appreciated.
Yet, as the paper's contribution is predominantly of theoretical nature, I _do not_ see the limited experimental investigation as a reason for a lower score.

**Questions:**

- The description of the lifting and bilevel reduction approaches in lines 50--55 feels a bit loose and vague and not much insightful. I would recommend to the Authors to either skip this part here or be more elaborate by giving more insights into how these approaches allow to transfer from the signed measure to a probability measure. At the same place, some citations of the papers proposing these ideas would be welcome (i.p. for the lifting).
- What is $J$ in line 44?
- uniform log Sobolev inequality: At first sight, it seems unclear, whether this assumption is reasonable or quite strong, as it has to hold for $\mu_t$ for all times $t$. Could the Authors comment on that? Are there sufficient assumptions on $\mu_0$ and $G$ that ensure such assumption at all times? Moreover, does it hold in interesting settings such as the training of one-hidden layer NNs? As far as I know, the cited reference [Chi22b] addresses parts of this. It would be nice to include a comment.
- Is there any intuitive insight that could explain, why the bilevel approach works much better than the lifting strategy?
- And I have a clarifying question concerning the bilevel reduction: The central observation allowing this reduction is the fact that the TV-norm can be expressed as in line 197? In addition to the change of infima in line 198, which allows to recast the optimization problem over the signed measures into an optimization over probability measures? (If so, maybe it would be nice to put this as an equation (in addition to the Proposition) to make it better visible as the central "trick".)

**Limitations:**

I do not see any unadressed limitation.

Moreover, I like how the lifting approach is discussed, despite turning out not to be the way to go.

---

> ### Author Rebuttal · Authors · 2024-08-05
>
> We thank the reviewer for their detailed evaluation, and address their questions and comments below.
> - Thank you for the feedback on this part of the introduction, it will be adapted.
> - "$J$" on line 44 should be "$F$".
> - Thank you for the feedback on this part of Section 2, we will add more discussions on the reasonableness of the uniform log-Sobolev inequality (LSI) assumption to the final version. The cited reference [Chi22b] indeed contains sufficient conditions on $F$ ensuring uniform LSI, which include the training of two-layer neural networks under some technical assumptions such as bounded activation function. Part of our motivation for the present work was to remove this boundedness assumption.
> - On why "bilevel" is better than "lifting" intuitively: For optimization over probability measures $\mathcal{P}_2(\mathcal{W})$, MFLD differs from Wasserstein gradient flow (WGF) by adding noise, which encourages exploration of $\mathcal{W}$. For optimization over signed measures $\mathcal{M}(\mathcal{W})$, which can be formulated as optimization over $\mathcal{P}_2(\mathbb{R} \times \mathcal{W})$ by lifting, there is no need to explore in the direction of the "weight" variables in $\mathbb{R}$, as the landscape is already convex in these directions. On the contrary, adding noise to both the weight ($\mathbb{R})$ and position variables ($\mathcal{W}$) turns out to be detrimental for convergence, by our negative result on "lifting" in Section 3.1.
>
>     This suggests following an intermediary dynamics: adding noise to the WGF on the lifting objective, but only on the position variables. The bilevel approach is a two-timescale limit of this intermediary dynamics (see Eq. (3.4)), which is amenable to analysis as it is itself an instance of MFLD.
> - Indeed, lines 197 and 198 are the central trick leading to the bilevel formulation. We will emphasize more on the role of this trick in the final version.

---

> > ### Comment · Reviewer_akPi · 2024-08-07
> >
> > Thanks for your reply and for addressing my questions, in particular also for providing some intuition about the two approaches, which does sound very reasonable.
> >
> > After reading also the other rebuttals and the Authors' replies, I remain very positive about the paper and maintain my positive score.

---

### Official Review · Reviewer_hSwQ · 2024-07-12

**Soundness:** 3
**Presentation:** 3
**Contribution:** 2
**Rating:** 6
**Confidence:** 4

**Summary:**

Mean-field Langevin dynamics (MFLD) has been developed for optimizing convex functionals over the space of probability measures. This work extends MFLD to convex problems defined over the space of signed measures. The authors consider two approaches: lifting and bilevel approaches, and prove the superiority of the bilevel approach. The lifting approach cannot satisfy the two required conditions for MFLD convergence under weak regularization, whereas the bilevel approach satisfies both conditions simultaneously. Additionally, an improved convergence rate is given for the enhanced annealing schedule of the temperature.

**Strengths:**

- This work successfully extends MFLD to convex optimization problems over the space of signed measures. Although recent work [TS24] also considered a similar approach, their method is a special case of the MFLD-bilevel proposed in the paper.
- This work proves an improved convergence rate for MFLD-bilevel with annealing temperatures. Although the rate still exponentially depends on the regularization strength, it may be difficult to avoid such dependence.

**Weaknesses:**

- Since this work considers (compact) Riemannian manifolds as the particle space, time discretization is essentially challenging. I’m curious about how to discretize the dynamics in time and guarantee the convergence rate.
- I guess the lifting approach satisfies both (P1) and (P2) once we limit the range of $r$ to be bounded, like $r∈[−R,R]$. If this is true, it is worth considering whether the bilevel approach is truly superior to the lifting approach.
- As the authors commented in the paper, [TS24] studies a quite similar method. I would like to see any additional technical challenges or difficulties compared to [TS24] if possible.

**Questions:**

- In Proposition 5.1, the iteration complexity for entering the sublevel where MFLD converges faster is not specified. Is it possible to explicitly describe the time required to enter this level?

**Limitations:**

Limitations are addressed in the paper.

---

> ### Author Rebuttal · Authors · 2024-08-05
>
> We thank the reviewer for their detailed and encouraging comments. We address each point in "Weaknesses" and "Questions" separately.
> - **On time-discretization over Riemannian manifolds**: So far, the theory for time-discretization of MFLD is established when $\Omega = \mathbb{R}^d$ [SWN23], or when the objective functional $F+\beta^{-1} H$ is a relative entropy and $\Omega$ is a Hessian manifold [GV22] or a product of spheres [LE23]. In all cases, a crucial ingredient is the assumption that $F$ is smooth in the sense of (P1), hence our focus on this property. It would indeed be an interesting direction for future research to see how the analysis techniques of the aforementioned works could be combined, to show convergence guarantees for time-discretizations of MFLD on manifolds.
> - **On constraining $r$ to $[-R, R]$ in the lifting approach**: Indeed, artificially limiting the range of $r$ would ensure (P1) and (P2). This is an interesting suggestion to make the lifting approach work. We did not find it natural to investigate this direction mainly for the following reasons:
>     - Practically, this introduces a hyperparameter $R$, the tuning of which might be quite tricky: taking $R$ too small will exclude the true solution ($\mathrm{argmin}~ G_\lambda$) from ever being reached, and taking $R$ too large will significantly affect the smoothness and LSI constants.
>     - Theoretically, the manifold $[-R, R] \times \mathcal{W}$ possesses boundaries, which adds significant technical difficulties to the analysis (in particular for time discretization).
>
>     We note that a related modification of the lifting approach is discussed in our answer to Reviewer akPi (4th bullet point).
> - **On the technical difficulties compared with [TS24]**: The dynamics they analyzed is (a variant of) MFLD-Bilevel applied to an objective $G$ of a specific form, as explained in Appendix A. The additional difficulties for us were $a)$ to establish (P1) and (P2) for the bilevel objective $J\_\lambda$ with no structural assumptions on $G$; $b)$ to analyze the convergence of MFLD-Bilevel beyond the crude upper bound provided by Theorem 2.1, corresponding to [TS24, Theorem 3.7]. We also note that the angle of that paper is quite different as it does not consider the lifting approach nor mentions signed measures. (We developed our results independently and concurrently with [TS24] which was announced on arXiv in March 2024.)
> - **On the time $t_0$ required until Proposition 5.1 applies**: $t\_0$ can indeed be explicitly described by inspecting the proof. Inspection reveals that the localness requirement is $J\_{\lambda,\beta}(\eta_{t_0}) - \inf J\_{\lambda,\beta} \leq \delta$ for $\delta = \left( 2L_0 G(0) (1+\frac{L_0}{\lambda}) \frac{L_0}{\lambda} \beta \sqrt{\frac{2\beta}{\alpha^*}} \right)^{-2} \left( \log(1-\varepsilon/\alpha_\beta^*) \right)^2$, and that it is guaranteed for $t_0 = \frac{\beta}{2 \alpha\_\tau} \exp(\frac{L\_0}{\lambda} G(0) \beta) \log\frac{J\_{\lambda,\beta}(\eta_0) - \inf J\_{\lambda,\beta}}{\delta}$.
>
> [GV22] Khashayar Gatmiry and Santosh S Vempala. “Convergence of the Riemannian Langevin algorithm” (2022).
>
> [LE23] Mufan Li and Murat A Erdogdu. “Riemannian Langevin algorithm for solving semidefinite programs” (2023).
>
> [SWN23] Taiji Suzuki, Denny Wu, and Atsushi Nitanda. “Mean-field Langevin dynamics: Time-space discretization, stochastic gradient, and variance reduction” (2023).
>
> [TS24] Shokichi Takakura and Taiji Suzuki. “Mean-field Analysis on Two-layer Neural Networks from a Kernel Perspective” (2024).

---

> > ### Comment · Reviewer_hSwQ · 2024-08-09
> >
> > Thank you for the reply. The authors have addressed my concerns well. I will increase the score. I would encourage the authors to add discussion on the lifting case with the constraint $r \in [-r,r]$.

---

### Decision · Program_Chairs · 2024-09-25

**Decision:**

Accept (spotlight)

**Comment:**

**Summary** This paper extends mean-field langevin dynamics MFLD to optimization problems over the space of signed measures by two procedures: a lifting approach and a bilevel approach.  The bilevel approach is shown to be superior in various forms.

The reviewers were unanimous in their appreciation of the paper.  We have summarized some of the comments below.

**Strengths**

The bilevel approach is exciting, and the problem setup appears timely and broadly interesting.  Quoting from the reviewers:

>Extending the MFLD via a bilevel approach to optimization problems over signed measures seems to be an interesting and relevant direction given that this captures several problems of interest in machine learning and data science.

The authors have considered two approaches (bilevel and lifted), and they have shown how bilevel outperforms the lifted approach.  This is especially significant given the lifted formulation is more standard

>The paper rigorously compares the lifting and bilevel approaches, and the result that the latter leads to stronger guarantees (at least in the low temperature regime) is quite surprising to me as many influential works on shallow NNs have built on the lifted formulation.


The paper is well, written, despite being technical:
>Despite the paper being of purely theoretical nature and in parts quite technical, it is overall well-written and good to follow. The organization and structure of a general introduction, revisiting the classical MFLD as well as the two reduction strategies in Section 2 and 3, respectively, helps in this regard.

Finally, the paper contains additional mathematical novelties which may have extra impact.
> The analysis utilizes novel techniques (namely local LSI bounds) to go beyond the standard results in MFLD.

**Recommendation**

The consensus is that this is a strong acceptance.